# The role of aerosol layer height in quantifying aerosol absorption from ultraviolet satellite observations

Jiyunting Sun[1,2], Pepijn Veefkind[1,2], Swadhin Nanda[1,2], Peter van Velthoven[3], Pieternel Levelt[1,2]

[1] Department of Satellite Observations, Royal Netherlands Meteorological Institute, De Bilt, 3731 GA, the Netherlands

[2] Department of Geoscience and Remote Sensing (GRS), Civil Engineering and Geosciences,
Delft University of Technology, Delft, 2628 CD, the Netherlands

[3] Department of Weather & Climate Models, Royal Netherlands Meteorological Institute, De Bilt, 3731 GA, the Netherlands

Correspondence to: Jiyunting Sun (jiyunting.sun@knmi.nl)

**Abstract.** The purpose of this study is to demonstrate the role of aerosol layer height (ALH) in quantifying the single scattering albedo (SSA) from ultraviolet satellite observations for biomass burning aerosols. In the first experiment, we retrieve SSA by minimizing the UVAI difference between observed ones and that simulated by a radiative transfer model. With the recently released ALH product of S-5P TROPOMI constraining forward simulations, a significant gap in the retrieved SSA (0.25) is found between radiative transfer simulations with spectral flat aerosols and strong spectral dependent aerosols, implying that inappropriate assumptions on aerosol absorption spectral dependence may cause severe misinterpretations of aerosol absorption. In the second part of this paper, we propose an alternative method to retrieve SSA based on long-term record of collocated satellite and ground-based measurements using the support vector regression (SVR). This empirical method is free from the uncertainties due to the imperfection of a priori assumptions on aerosol micro-physics as that in the first experiment. We present the potential capabilities of the SVR by several fire events happened in recent years. For all cases, the difference between SVR-retrieved SSA and AERONET are generally within ±0.05, with over half of samples are within ±0.03. The results are encouraging, though at the current phase, the model tends to overestimate the SSA for relatively absorbing cases and fails to predict SSA for some extreme situations. The spatial contrast in SSA retrieved by radiative transfer simulations is significantly higher than that of SVR, and the latter better agrees with SSA from MERRA-2 reanalysis. In the future, more sophisticated feature selection procedure and kernel functions should be taken into consideration to improve the SVR model accuracy. Moreover, the high resolution TROPOMI UVAI and collocated ALH products will guide us to more reliable training data set and more powerful algorithms in quantify aerosol absorption from UVAI records.

## 1 Introduction

The concept of the near-ultraviolet (near-UV) absorbing aerosol index (UVAI) initially came along with the ozone product of the Nimbus 7/Total Ozone Mapping Spectrometers (TOMS). It detects elevated UV-absorbing aerosol layers by measuring the spectral contrast difference between a satellite observed radiance in a real atmosphere and a model simulated one in a Rayleigh atmosphere (Herman et al., 1997):

$$UVAI = -100 \left( log_{10} \left( \frac{I_\lambda}{I_{\lambda 0}} \right)^{obs} - log_{10} \left( \frac{I_\lambda}{I_{\lambda 0}} \right)^{Ray} \right) \tag{1}$$

, where the superscript $obs$ and $Ray$ denote the radiance from observations and that from simulations, respectively. $I_\lambda$ and $I_{\lambda 0}$ are the radiance at wavelength $\lambda$ and $\lambda_0$. $\lambda$ is the wavelength where the radiance difference between a Rayleigh

and a measured scene is calculated, and $\lambda_0$ is the longer wavelength where a spectrally constant scene reflectivity is assumed for the calculation of $I_\lambda^{Ray}$. Positive UVAI indicates the presence of absorbing aerosols, whereas negative or near zero values imply non-absorbing aerosols or clouds (Herman et al., 1997). The over four-decade UVAI observations (1978 to present) has been widely used for aerosol research. It would be beneficial to derive aerosol absorption properties from the long-term global UVAI records, e.g. the single scattering albedo (SSA), which is the

ratio of aerosol scattering to aerosol extinction. Aerosols are considered as the largest error source in radiative forcing assessments (IPCC, 2014), and SSA is one of the key parameters to reduce this uncertainty (Haywood and Shine, 1995).

The magnitude of UVAI depends on many factors (Herman et al., 1997; Torres et al., 1998; Hsu et al., 1999; de Graaf et al., 2005). Although non-aerosol factors exist, such as spectral dependence of the surface, ocean color, sun-glint and

cloud contamination, the most dominant are aerosol concentration, aerosol vertical distribution and aerosol optical properties (Wang et al., 2012; Buchard et al., 2017). To derive SSA from UVAI, the information on other two parameters are necessary. The aerosol concentration is usually provided in terms of the aerosol optical depth (AOD). There are plentiful AOD products with wide spatial-temporal coverage. By contrast, there is much less information on the aerosol vertical distribution. The most well-known aerosol vertical distribution product is provided by the Cloud-

Aerosol Lidar with Orthogonal Polarization (CALIOP), but the number of measurements is limited because of its narrow tracks (Winker et al., 2009). Passive sensors make efforts on retrieving the aerosol layer height (ALH) from columnar measurements. For example, Chimot et al. (2017) present the feasibility of ALH retrieval using the OMI oxygen band at 447 nm. Tilstra et al. (2018) developed algorithm to derive absorbing aerosol layer height from GOME-2 FRESCO cloud layer height products. Xu et al. (2017; 2019) attend to retrieve ALH from EPIC oxygen

absorption bands for dust and carbonaceous layers over both land and ocean surfaces.

Recently a new ALH product has been run operationally, based on the measurements at near-Infrared (NIR) oxygen A-band of the TROPOspheric Monitoring Instrument (TROPOMI) on board the Copernicus Sentinel-5 Precursor (S-5P) (Sanders et al., 2015). TROPOMI has a wide swath of 2600 km, providing daily global coverage with a spatial resolution of 7×3.5 km$^2$ in nadir. The instrument is equipped with both the UV-visible (270–500 nm) and the near-

infrared (NIR) (675–775 nm) channels, which can simultaneously provide UVAI and the collocated ALH product (Veefkind et al., 2015).

The purpose of this paper is to demonstrate the role of the ALH in quantifying aerosol absorption from UVAI using newly released TROPOMI level 2 ALH product. At current phase we only focus on the biomass burning aerosols. Two experiments are conducted. First, following previous studies (Colarco et al., 2002; Hu et al., 2007; Jeong and Hsu,

2008; Sun et al., 2018), we build up lookup tables (LUTs) of simulated UVAI for various aerosol optical properties by radiative transfer models (RTMs). Then SSA is derived by minimizing the difference between pre-calculated UVAI and satellite observed ones. The major uncertainties in the retrieved SSA are caused by assumptions on the wavelength-dependent refractive index and the availability of reliable aerosol vertical distribution information (Sun et al., 2018). Now with the operational TROPOMI ALH constraining forward simulations, it is expected to partly reduce

the SSA retrieval uncertainty meanwhile quantifying the influence of assumed aerosol properties on the retrieved SSA. Although the availability of ALH in radiative transfer calculations can improve the SSA retrieval, assumptions on aerosol micro-physics remain inevitable. In the second experiment, we therefore propose an empirical method to predict aerosol absorption, based on the long-term records of collocated UVAI, ALH, AOD and absorbing aerosol optical depth (AAOD) using machine learning (ML) techniques. ML algorithms learn the underlying behavior of a

system from a given training data set. They are particularly useful to address ill-defined inversion problems in the field

of geosciences and remote sensing, where theoretical understanding is incomplete but there is a significant amount of observations (Lary et al., 2015). We employ ML techniques in order to avoid explicit assumptions on aerosol microphysics as made in the first experiment. By now, the ALH observations are not abundant and we will use the ALH accompanied in the AOD retrieval from the OMAERUV product in the training procedure (Torres et al., 2013).

Nevertheless, the recent TROPOMI ALH and other future ALH products make such empirical methods of great potentials. Various ML algorithms have been developed to deal with classification or regression problems. In this paper we choose the support vector regression (SVR), a regression variant form of the support vector machines (SVM) (Drucker et al., 1997). Compared with other algorithms (e.g. the Artificial Neural Network), SVR is less sensitive to training data size and can successfully work with limited quantity of data (Mountrakis et al., 2011; Shin et al., 2005).

We will present the capability to retrieve SSA from UVAI of using this empirical method with multiple case studies. This paper is organized as follows: the first experiment is outlined in section 2, with description on setting radiative transfer simulations, and the analysis of the uncertainty trigger by the assumption on aerosol absorption spectral dependence; Section 3 starts with introduction of SVR, followed by training data set preparation, SVR model hyperparameter tuning, error analysis and case applications. Finally, the major conclusions and implications for future

research are summarized in section 4.

**2 Experiment 1: SSA retrieval using radiative transfer simulations**

In this section, we present the first experiment that retrieves SSA by radiative transfer calculations as done in previous studies (Colarco et al., 2002; Hu et al., 2007; Jeong and Hsu, 2008; Sun et al., 2018). Forward radiative transfer simulations are realized by the KNMI developed radiative transfer model DISAMAR (Determining Instrument

Specifications and Analyzing Methods for Atmospheric Retrieval) (de Haan, 2011). Fig.1 illustrates the model inputs and the procedure. For each pixel, first, aerosol optical properties are computed by Mie theory for various pre-defined aerosol models. Then DISAMAR calculates UVAI using the corresponding satellite information: AOD, ALH, the solar zenith angle ($\theta_0$), the viewing zenith angle ($\theta_v$), the solar azimuth angle ($\varphi_0$), the viewing azimuth angle ($\varphi_v$), surface albedo ($A_s$) and surface pressure ($P_s$) of the target pixel. The output of the forward simulations is a LUT of UVAI as a

function of the input SSA (determined by the pre-defined aerosol models), which is fit by a second order polynomial function. Finally, by specifying the corresponding satellite observed UVAI, the SSA of the target pixel is estimated from the UVAI-SSA relationship. The retrieved SSA is reported at 500 nm in order to compare with the results of the SVR method. Section 2.1 will introduce the input parameters in radiative transfer simulations, followed by retrieval results in section 2.2.

**2.1 Radiative transfer simulation setup**

**2.1.1 Aerosol models**

The aerosol models used for the Mie calculations are a combination of the aerosol models in ESA Aerosol_cci project (Holzer-Popp et al., 2013) and that in the OMAERUV algorithm (Torres et al., 2007; Torres et al., 2013). We assume a fine mode smoke aerosol type and further divide it into 7 subtypes as listed Table 1. We use the particle size

distribution of the fine mode strongly absorbing aerosol of ESA Aerosol_cci project. The geometric radius ($r_g$) is 0.07 μm (effective radius $r_{eff}$ of 0.14 μm) and the geometric standard deviation ($\sigma_g$) is 1.7 (logarithm variance $ln\sigma_g$ of 0.53). The real part of the refractive index ($n$) uses the same value as in the OMAERUV algorithm, which is set to be 1.5 for all subtypes and spectrally flat. We adopt the imaginary part of the refractive index at 388 nm ($\kappa_{388}$) of the

OMAERUV smoke subtypes (except for BIO-1 whose $\kappa_{388}$ is 0) in our study and add a subtype with $\kappa_{388}$ equaling to

115    0.06.

Many studies have shown evidence that absorption by biomass burning aerosols in the near-UV band has a strong spectral dependence (Kirchstetter et al., 2004; Bergstrom et al., 2007; Russell et al., 2010). Accordingly, a constant 20% $\Delta\kappa$ has been applied to all smoke subtypes in the recent OMAERUV algorithm (Jethva and Torres, 2011), where $\Delta\kappa$ is defined as the relative difference between $\kappa_{354}$ and $\kappa_{388}$ (i.e. $\Delta\kappa = (\kappa_{354} - \kappa_{388})/\kappa_{388}$). In this experiment, we

will investigate how the retrieved SSA responds to the assumed spectral dependence by considering 9 different $\Delta\kappa$ values from 0% (i.e. 'grey' aerosols) to 40% (very strong spectral dependence). This corresponds to an Absorbing Ångström exponent ($\alpha_{abs}$) from 1 to 3.4 and from 1.3 to 4.7, depending on aerosol subtype. Note that the $\Delta\kappa$ is only applied between $\kappa_{354}$ and $\kappa_{388}$. As we only investigate the influence due to aerosol absorption spectral dependence in near-UV range in this study, aerosol absorption at wavelengths larger than 388 nm is set equal to that at 388 nm.

To summarize, the first experiment consists of 9 cases represented by different $\Delta\kappa$. Within each case, there are 7 pre-defined aerosol subtypes with varying $\kappa_{388}$. Thus, 63 forward simulations are performed for each individual pixel.

### 2.1.2 Inputs from satellite

Fig.1 presents the parameters input for the radiative transfer simulations of UVAI. Satellite measurement geometries ($\theta_0$, $\theta_v$, $\varphi_0$ and $\varphi_v$) and the surface pressure ($P_s$) accompanied with the TROPOMI UVAI reprocessed product

(https://scihub.copernicus.eu last access: 8 June 2018) are input for the forward simulations. The TROPOMI UVAI is calculated for two different wavelength pairs. One uses the conventional 340 and 380 nm to continue the heritage of UVAI records from multiple sensors, and the other uses 354 and 388 nm in order to allow comparison with OMI measurements (D.C. Stein Zweers, 2016). In this study we employ the 354 and 388 nm pair.

TROPOMI ALH is retrieved at oxygen A-band (759-770 nm), where the strong absorption of oxygen causes the highly

structed spectrum (https://scihub.copernicus.eu last access: 22 June 2018). This feature is particularly suitable for elevated optically dense aerosol layers (Sanders et al., 2015; Sanders and de Haan, 2016). The ALH is reported in both altitude and pressure. For the forward radiative transfer calculations, the input aerosol profile is parameterized according to the settings in ALH retrieval algorithm: a one-layered box shape profile, with central layer height derived from TROPOMI and an assumed constant pressure thickness of 50 hPa (Sanders and de Haan, 2016). At the same

band, there is TROPOMI FRESCO cloud support product providing cloud fraction (CF) for mitigating cloud effects as will be explained later (https://scihub.copernicus.eu last access: 19 Sept 2018) (Apituley et al., 2017).

The TROPOMI AOD product has not been operational, thus we use AOD from the Level 2 product MYD04 (Collection 6) of Aqua MODIS (http://dx.doi.org/10.5067/MODIS/MYD04_L2.006 , last access: 17 July 2019). Aqua has an overpass time similar to S-5P (13:30 local time). The AOD at 550 nm used in the RTM-based method is a

combination of the Deep_Blue_Aerosol_Optical_Depth_550_Land and the Effective_Optical_Depth_Op55um_Ocean (Levy et al., 2013).

The surface albedo that used to retrieve TROPOMI UVAI is currently not available in the product. Instead, we use the Aura/OMI Level 3 Lambertian equivalent reflectance (LER) monthly climatology calculated from measurements between 2005 and 2009 (Kleipool et al., 2008) (Kleipool, 2010) (http://dx.doi.org/10.5067/Aura/OMI/DATA3006, last

access: 26 September 2018). TROPOMI on S-5P and OMI on Aura have similar overpass times (13:30 local time) and measuring geometries (Levelt and Noordhoek, 2002) (Veefkind et al., 2015).

Due to different spatial resolutions, TROPOMI ALH, OMI LER climatology and MODIS AOD are resampled onto the TROPOMI UVAI grid. Before implementing radiative transfer calculations, pre-processing excludes pixels with large

solar zenith angle ($\theta_0 > 70°$), weak aerosol absorption ($UVAI_{354,388} < 1$), insignificant aerosol amount ($AOD_{550} < 0.5$) or cloud contamination ($CF > 0.3$).

## 2.2 SSA retrieved by radiative transfer simulations

In the first experiment, we focus on one of the largest fire events that happened in southern California in 2017, i.e. the Thomas Fire (http://www.fire.ca.gov/current_incidents/incidentdetails/Index/1922 ). Fig.A1 in Appendix shows the RGB plume captured by MODIS on 12 December 2017. A brown smoke plume produced by the Thomas Fire was blown away from the continent and transported northwards. The major part of the plume was over the ocean and under cloud free condition, which is favorable for space-borne aerosol observations. There are totally 5217 pixels in this case. Fig.2 presents the UVAI, ALH and AOD after pre-processing. The highest UVAI appeared at the south part of the plume, where both aerosol loading and aerosol layering are relatively high (AOD > 2 and ALH is over 2.5 km). Fig.3a displays the mean SSA of all plume pixels retrieved by the RTM-based method as a function of $\Delta\kappa$. The retrieved aerosol absorption decreases with $\Delta\kappa$. This finding is in good agreement with Jethva and Torres (2011). 'Gray' aerosols require stronger absorption to reach the same level of UVAI than 'colored' aerosols. This also explains the high SSA standard deviation (filled area) in the cases with little or no spectral dependence in aerosol absorption. The large variability in retrieved SSA (from 0.69±0.13 to 0.94 ±0.03) demonstrates that inappropriate assumptions on the spectral dependence of near-UV aerosol absorption may significantly bias interpretations of smoke aerosol absorption and should be carefully handled in forward radiative transfer calculations.

The retrieved aerosol absorption is compared with the nearby the version 3 level 1.5 AERONET inversion product (https://aeronet.gsfc.nasa.gov last access: 4 June 2019). Only one site is within 50 km from TROPOMI plume pixels (UCSB, (119.845°W,34.415°N)) with only one record for this case. The SSA at 500 nm at 18:54:47 UTC is 0.98 (sky radiance error 15.8%), which is nearly 3 hours ahead of TROPOMI overpass. There are 15 TROPOMI collocated pixels to UCSB with distance within 50 km and time difference within 3 hours. Hereafter we call them AERONET-collocated pixels. As illustrated in Fig.3b, the mean SSA of the collocated pixels also increases with $\Delta\kappa$ and eventually levels off at around 0.96. The extremely low SSA and high variation (0.57±0.25) retrieved for 'gray' aerosols prove that the spectral independence assumption is not recommended for smoke aerosols.

The differences between the mean SSA of the collocated pixels and the AERONET measurement are shown in Fig.3c. The retrieved SSA starts falling inside the uncertainty range of AERONET (±0.03) (Holben et al., 2006) when $\Delta\kappa$ is 25%, where the plume SSA is 0.90±0.05 and the AERONET-collocated SSA is 0.96±0.02 (Table 2). Table 2 also presents the SSA accompanied in AOD retrieval from OMAERUV version 3 product (http://dx.doi.org/10.5067/Aura/OMI/DATA2004 last access: 17 October 2018). OMI pixels are collocated to the AERONET site in the same way as TROPOMI. The SSA of the OMAERUV-AERONET collocated pixels is 0.06 lower than that of AERONET, which indicates a 20% spectral dependence of aerosol absorption in OMAERUV algorithm may be not sufficient for this case. Although our retrieved SSA seems closer to AERONET retrieved SSA than that provided in OMAERUV, one should keep in mind that there is only one record for this event, the meteorological conditions, combustion phases and even the aerosol compositions may change during the 3-hour time difference.

Fig.4 presents the spatial distribution of retrieved AAOD and SSA when $\Delta\kappa$ is 25%, which shows a strong heterogeneity in the horizontal direction. The plume center is most absorbing, where SSA is even less than 0.70. The SSA gradually increases when the plume transported northwards. SSA is expected to be low near source flaming regions (Eck et al., 1998; Eck et al., 2003; Eck et al., 2013) while SSA may become higher when aerosols age during

transport (Reid et al., 2005; Lewis et al., 2009).The strong spatial variability in SSA is mainly controlled by the heterogeneity of the UVAI (Fig.3a) through the one-to-one numerical relationship. This relationship may differ from one pixel to another as the algorithm focuses on one-pixel retrieval each time. Depending on the combustion phase and meteorological conditions, heterogeneity in aerosol properties is expected for plume of this size. Nevertheless, whether such a large SSA difference 0f 0.38 (maximum SSA – minimum SSA, Table 2) is reasonable needs further investigations (discussed in Section 3.6.3).

## 3 Experiment 2: SSA retrieval using support vector regression

In this section, we propose an empirical method to derive SSA as an alternative of the radiative transfer simulations presented in the first experiment. The motivation is that the assumptions on aerosol micro-physics in forward simulations are inevitable, whereas our knowledge to them is inadequate (particularly the aerosol absorption spectral dependence). An inappropriate assumption may lead to significant bias in retrieved SSA (Fig.3). On the other hand, SVR (and other ML algorithms) is applicable to solve ill-posed inversion problems by learning the underlying behavior of a system from a given data sets without such a priori knowledge on aerosol micro-physics. In this paper, we construct an SVR model with UVAI, AOD and ALH as input features and AAOD as the output, then derive the SSA by the following relationship:

$$SSA = 1 - \frac{AAOD}{AOD} \tag{2}$$

The procedure of SVR prediction is presented in Fig.5. We start with a brief introduction of the SVR algorithm, followed by input feature selection (section 3.2), training and testing data set preparation (section 3.3), SVR model hyper-parameters tuning (section 3.4), error analysis (section 3.5) and case applications (section 3.6).

### 3.1 Support vector regression

SVR (Drucker et al., 1997) is the regression variant of SVM, a supervised non-parametric statistical algorithm initially devised by Cortes and Vapnik (1995). SVM algorithm is suitable to solve problems of small training data sets with a high-dimensional feature space and can provide excellent generalization performance (Durbha et al., 2007; Yao et al., 2008), which has been applied extensively to solve remote sensing problems (Lary et al., 2009; Mountrakis et al., 2011; Noia and Hasekamp, 2018). The basic ideal of SVM in classification problems is finding an optimal hyperplane in a high-dimensional feature space maximizing the margin between the two classes to minimize misclassifications (Durbha et al., 2007). The same principle is applied to regression problems, SVR attends to find an optimal hyperplane that maximizes the margin of tolerance in order to minimize the prediction error. The error within the margin does not contribute to the total loss function, while samples on the margin are called support vectors.

For the detailed mathematical formulation of SVR algorithm one can refer to Smola and Scholkopf (2004). Briefly, given the training data with $n$ observations $\{(x_1, y_1), (x_2, y_2), \ldots, (x_n, y_n)\}$, assuming the statistical model as the following:

$$y = r(x) + \delta \tag{3}$$

, where $x$ is a multivariate input and $y$ is a scalar output with length $n$. $\delta$ is the independent zero mean random noise. The input $x$ is first mapped onto a feature space with dimension of $m$ by a non-linear transformation, then a linear model $f(x)$ is constructed based on it:

$$f(x) = \sum_{j=1}^{m} \omega_j\, g_j(x) + b \tag{4}$$

, where the $g_j(x)$ is the non-linear transformation, $\omega_j$ is the model parameter vector and $b$ is the bias. SVR tries to find the optical model from a set of approximate functions $f(x)$. An approximate function is assessed by the loss function.

In SVR, the loss function is defined as $\varepsilon$-insensitive loss:

$$L\big(y, f(x)\big) = \begin{cases} 0 & if\ |y - f(x)| \le \varepsilon \\ |y - f(x)| - \varepsilon & otherwise \end{cases} \tag{5}$$

Then the total empirical risk is:

$$R(\omega) = \frac{1}{n} \sum_{i=1}^{n} L\big(y_i, f(x_i)\big) \tag{6}$$

SVR performs linear regression in high-dimension feature space using $\varepsilon$-insensitive loss, meanwhile reduce the model complexity by minimizing the norm $\|\omega\|^2$. By introducing non-negative slack variables ($\xi_i$ and $\xi_i^*$) to measure the deviations of errors outside $\varepsilon$, SVR problems can be formulated as following:

$$minimize\ \frac{1}{2}\|\omega\|^2 + C \sum_{i=1}^{n} (\xi_i + \xi_i^*) \tag{7}$$

$$s.t. \begin{cases} y_i - f(x_i) \le \varepsilon + \xi_i^* \\ f(x_i) - y_i \le \varepsilon + \xi_i \\ \xi_i, \xi_i^* \ge 0 \end{cases}$$

, where $C$ is a positive regularization constant determining the trade-off between model complexity and the degree to which deviations larger than $\varepsilon$ are penalized. The optimization problem can be transferred into the dual problem by introducing Lagrange multipliers ($\alpha_i$ and $\alpha_i^*$) and the solution becomes:

$$f(x) = \sum_{i=1}^{n} (\alpha_i - \alpha_i^*)\, K(x_i, x) + b \tag{8}$$

$$s.t.\ 0 \le \alpha_i, \alpha_i^* \le C$$

, where $K(x_i, x)$ is the kernel function that is positive semi-definite in order to satisfy Mercer's theorem. The kernel function makes the SVR able to solve non-linear problems.

According to the description above, we know that SVR generalization performance and estimation accuracy depend on the regularization constant $C$, the width of the tolerance margin $\varepsilon$ and the kernel function $K(x_i, x)$. We will discuss how to determine the three hyper-parameters in section 3.3.

**3.2 Feature selection based on OMI and AERONET observations**

Although SVR is able to deal with high-dimensional input features, feature selection is still important for

generalization performance, computational efficiency and interpretational issues (Weston et al., 2001). Many sophisticated approaches have been devised for feature selection (Guyon and Elisseeff, 2003). In this study we choose features based on our empirical knowledge of UVAI and the Spearman's rank correlation coefficients ($\rho$).

**3.2.1 Collocating OMI and AERONET observations**

The feature selection is based on the collocated OMAERUV version 3 product

(http://dx.doi.org/10.5067/Aura/OMI/DATA2004 last access: 17 October 2018) and AERONET version 3 level 1.5 inversion product (https://aeronet.gsfc.nasa.gov, last access: 4 June 2019). The OMAERUV is currently the only

satellite product containing a long-term UVAI, AOD, SSA and corresponding ALH (Torres et al., 2007; 2013). Its AOD was validated by the multi-year AERONET record (Ahn et al., 2014) and its SSA was evaluated by AERONET Almucantar retrievals (Jethva et al., 2014). The ALH is the best-guessed either from CALIOP climatology or assumed ALH in the retrieval (if the CALIOP climatology is not available) (Torres et al., 2013). As a result, one should keep in mind that the ALH from OMAERUV may suffer from the uncertainties of CALIOP climatology and a priori assumptions, and collocation error between OMI pixels and CALIOP footprint. It is also noted that there are two official OMI aerosol level 2 products though, the OMI measurements in this paper only refers to the OMAERUV product.

We collect the measurements of OMAERUV and AERONET from 2005-01-01 to 2017-12-31. OMI pixels with $\theta_0$ larger than 70° or cloud fraction larger than 0.1 are excluded. Then OMI observations are considered as collocated with an AERONET site if their spatial distance is within 50 km and their temporal difference is within 3 hours. To ensure consistency between the different measurement techniques (ground-based and space-borne), we also exclude samples if the SSA difference between OMAERUV and AERONET is larger than 0.03, or the AOD difference between OMAERUV and AERONET is larger than 5%. The AERONET SSA and AAOD are linearly interpolated to 500 nm as OMAERUV reports them at this wavelength. In total 5679 samples are obtained. Fig.B1 in the Appendix shows the global distribution of the collocated OMAERUV-AERONET samples. Note that these samples are not restricted to biomass burning areas, but may also contain other aerosol types.

### 3.2.2 Feature selection

The OMAERUV-AERONET joint data set consists of following parameters: UVAI calculated by 354 and 388 nm wavelength pair, satellite geometries, surface albedo, surface pressure and ALH from OMAERUV, and SSA, AOD and AAOD from AERONET. Note that the UVAI used here is the 'residue' field in the original OMAERUV product, where the simulated radiance ($I_\lambda^{Ray}$ in Eq.(1)) is calculated by a simple Lambertian approximation that is consistent with TROPOMI UVAI (Torres et al., 2018). Fig.6 presents the Spearman's rank correlation coefficients matrix ($\rho$) of those parameters. It is clear that except for AAOD, SSA is barely associated with other parameters. The correlation between UVAI and SSA is rather low ($\rho = -0.25$). On the other hand, AAOD is highly associated with UVAI ($\rho = 0.66$) as well as AOD ($\rho = 0.66$) as it carries information on both aerosol absorption and aerosol loading. Therefore, it is preferred to predict AAOD from given UVAI and derive SSA via in Eq. (2) afterwards rather than to directly predict SSA from UVAI. Besides, as mentioned previously, AOD and ALH are the major factors influencing UVAI, which is also reflected by the relatively stronger correlation ($\rho = 0.4$). Consequently, we construct an SVR model with UVAI, ALH and AOD as the input features, and AAOD as the output. The UVAI also has a dependence on $\theta_0$, but in this study we only focus on the aerosol related features.

### 3.3 Preparing training and testing data sets

The SVR model is trained and tested base on the OMAERUV-AERONET joint data set containing 8616 samples as described in the last section (consisting of UVAI, ALH from OMAERUV, and AOD, AAOD from AERONET). We further partition it into a training and a testing data set, respectively. The testing data set is used to evaluate the generalization performance of an SVR model trained by training data set, in order to avoid high bias (underfitting) or high variance (overfitting) problems. The empirical ratio between a training and testing data set is 70% versus 30%, thus there are 3975 samples in the training data set and 1704 samples in the testing data set.

### 3.4 SVR hyper-parameters tuning

As described in section 3.1, the generalization performance and model accuracy of the SVR depends on the following hyper-parameters: (1) the width of insensitive zone $\varepsilon$. The cost function does not consider errors in the training data as long as their deviation to the truth is smaller than $\varepsilon$; (2) the regularization constant $C$ that determines the trade-off between model complexity and the degree to which deviations larger than $\varepsilon$ are penalized; (3) choice of the kernel and its parameters. We adopt the methodology from (Cherkassky and Ma, 2004), where SVR parameter $C$ and $\varepsilon$ can be directly determined from the statistics of training data set:

$$C = max(|\overline{y} + 3\sigma_y|, |\overline{y} - 3\sigma_y|) \tag{9}$$

$$\varepsilon = 3\sigma \sqrt{\frac{ln(n)}{n}} \tag{10}$$

, where $\overline{y}$ and $\sigma_y$ are the mean and standard deviation of the output parameter in the training data set, $\sigma$ is the input noise level (we set it to 0.001) and $n$ is the number of training samples. The determined values for $C$ and $\varepsilon$ are in Table 3. We employ the widely used radial basis function (RBF) kernel function to solve the non-linearity in the SVR model. Compared with other kernel functions, RBF is relatively less complex and more efficient. The RBF kernel is defined as:

$$K(x_i, x) = exp\left(-\frac{\|x_i - x\|^2}{2p^2}\right) \tag{11}$$

, where $p$ is the kernel width parameter that reflect the influencing area of support vectors. This parameter is determined by hyper-tuning on the testing data set (Durbha et al., 2007) (explained below).

The RMSE of training process may overestimate the accuracy of an SVR model, because the training and predicting process are based on the same data set. Instead, an independent testing data set is used to represent the accuracy of the SVR model. The difference of model accuracy between training and testing process reflects the generalization performance of the SVR model. An ideal SVR model should output a low level RMSE meanwhile the discrepancy between training and testing process is also small. If the RMSE of testing process is much larger than that of training process, then the SVR may suffer from overfitting problems. Fig.7 shows the hyper-parameter tuning process. The first row is the RMSE of training process as a function of $C$ and $\varepsilon$. The second row is the RMSE relative difference between testing process and training process. Column indicate different values of $p$. The cross marker indicates values of the $C$ and $\varepsilon$ determined by the Eq. (9) and (10). It is clear that when $p^2$=1.67, the RMSE of training process is relatively small, meanwhile the model accuracy difference between training process and testing process is also small. The final value of $C$, $\varepsilon$ and $p$ that will be applied in case studies are listed in Table 3. The corresponding RMSE of AAOD predicted by the training process and testing process are at level of 0.01 (Fig.8a).

### 3.5 Error analysis

The error sources of SSA retrieval using SVR model depends on the model accuracy as well as the quality of input data. The model accuracy can be represented by the RMSE of the testing process (0.01). As shown in Fig.8a, the SVR model has difficult predicting AAOD larger than 0.05, where most significant biases appear at this range. The uncertainty in AAOD is passed to the SSA by Eq. (2). Fig.8b shows the retrieved SSA in training and testing process. It is noted that the predicted SSA is overall positively biased, particularly in relatively stronger absorption cases (SSA <0.90). The bias is possibly due to that in the feature domain, the UVAI is relatively strongly correlated to others (i.e.

AOD and ALH), which may contain redundant information that adversely impact model performance (Weston et al., 2001; Durbha et al., 2007). More sophisticated feature selection scheme is suggested to reduce the redundancy, e.g. Minimum Redundancy Maximum Relevance (mRMR, Peng et al., 2005). Moreover, the RBF kernel function may not capable enough to solve the non-linearity among the training data sets. The accuracy of SSA predict by testing data set

is $\pm 0.02$, where 82% samples falling the uncertainty range ($\pm 0.03$) of the true SSA (AERONET) and their accuracy is even higher ($\pm 0.01$).

The error the retrieved SSA due to the input features may come from the observational or retrieval uncertainties in each parameter. In our case, the typical UVAI bias requirement is at magnitude of 1 (Lambert et al., 2019). It is reported TROPOMI UVAI suffers from the long-term downward wavelength-dependent trend in irradiance

(Rozemeijer and Kleipool, 2018). The detected degradation in $UVAI_{354,388}$ is around 0.2 since August 2018 (Lambert et al., 2019). The typical accuracy of TROPOMI ALH is 50 hPa, though in some situations the bias may over this value (e.g. low aerosol loading over bright surface) (Sanders et al., 2016). Depending on the retrieval algorithm the uncertainty of MODIS AOD is $\pm 0.05 + 15\% AOD_{AERONET}$ (Dark Target algorithm) (Levy et al., 2010) or $\pm 0.03 + 0.2 AOD_{MODIS}$ (Deep Blue algorithm) (Sayer et al., 2013). The SSA sensitivity to input features is presented in

Fig.9. We use the mean value of each parameter in the OMAERUV-AERONET data set as reference values (Fig.B2, UVAI = 1.59, ALH = 2.96 km, AOD = 0.39), the corresponding SSA value is 0.94. The positive bias of UVAI always leads to underestimation in SSA, unless the aerosol layer is located at a relatively high altitude or aerosol loading is low. Conversely, the insufficient UVAI causes the overestimation in SSA, except for cases where ALH is low or AOD is high. The sensitivity of SSA to UVAI is weaker when the aerosol layer is close to surface or at a very high altitude.

The sensitivity of SSA to UVAI always increases with AOD.

### 3.6 Case applications

Once the hyper-parameters are determined (Section 3.4), the trained SVR model is ready to predict aerosol absorption. The first application is the California fire event in 2017 December (Section 3.6.2), the same as that in the first experiment. To demonstrate the generalization capability of the SVR model, we also apply it to other fire events as

long as there are collocated TROPOMI and MODIS measurements and AERONET-retrieved SSA to compare with (Section 3.6.2).

For all applications, the input parameters in the SVR model are TROPOMI UVAI (calculated by 354 and 388 nm wavelength pair), TROPOMI ALH and MODIS AOD, respectively. The MODIS AOD at 550 nm is converted to 500 nm using the Ångström exponent ($\alpha$) provided by the collocated AERONET site. Note that the data includes pixels

with CF larger than 0.1 in order to ensure there is satellite measurements collocated with the AERONET sites (though CF is no larger than 0.3).

### 3.6.1 California fire event on 12 December 2017

Fig.10 presents the retrieved AAOD and corresponding SSA. It is noted that in the center of the plume, where UVAI and AOD are higher while ALH is relatively lower (Fig.2). The SSA should be smaller to compensate the low altitude

of the aerosol layer according to Fig.9. However, the SVR retrieved SSA is even higher than its surroundings. It is because that at this region, the UVAI and AOD are outside of the distribution of corresponding parameters shown in Fig.B2. The 13-year OMAERUV-AERONET joint data cannot cover some extreme situations. The reason could be the size of the joint data is relatively small as a result of data availability and collocation criteria, or the quality of the joint

data suffers from observational or retrieval uncertainties. As a result, the SVR model fails to handle the input values outside the range of training data set.

The SSA of the all plume pixels is 0.94±0.01 (including the failed-predicted pixels) and that for the AERONET-collocated pixels (pixels within 50 km from UCSB) is 0.97±0.01 (Table 4). These values may be overestimated while the standard deviation may be underestimated because of the SVR prediction failures of some samples. The SSA difference relative to the AERONET retrieval is only 0.01, which is within the uncertainty range of AERONET (±0.03).

### 3.6.2 Other case applications

To present the generalization performance of SVR, we apply it to other fire events as long as there is collocated information from TROPOMI, MODIS and AERONET. The same pre-processing as the previous case is applied to exclude pixels with UVAI smaller than 1, AOD smaller than 0.5 or CF larger than 0.3.

Fig.11-13 present California fire events during 9-11 November 2018. The plumes were over ocean but partly contaminated by the underlying clouds (Fig.A2-A4 present the Aqua MODIS RGB images). Fig.14 shows the Canada fire event on 29 May 2019. The case was over land (Fig.A5 present the Aqua MODIS RGB image), which means the brighter surface may cause higher bias in the input AOD and ALH than cases over dark surfaces (Remer 2005; Sanders and de Haan, 2016).

The retrieved SSA for above events is listed in Table 4. Similar to the California 2017-12-12 case, The SVR fails to retrieve reasonable SSA for pixels if input features outside their corresponding histogram in the OMAERUV-AERONET data (Fig.2B), which may cause overestimations in plume mean SSA. The plume SSA of two California fire events are similar, with values around 0.94-0.95. The retrieved SSA for the Canada fire is relatively higher (0.97) We further plot the SSA retrieved by SVR against collocated AERONET records (black crosses in Fig.15). Including the first case (California fire on 2017-12-12), there are 9 collocated records obtained. The difference between SVR-retrieved SSA and AERONET are alomost within difference of ±0.05, among which over half (5 out of 9) fall in AERONET SSA uncertainty range (±0.03). We also provide SSA from OMAERUV for these cases (Table 4 and blue circles in Fig.15). Compared with OMAERUV, the SSA retrieved by SVR shows a better consistency with AERONET, though one should keep in mind that the accuracy of SVR-retrieved SSA is ±0.02 and the model tends to overestimate the SSA for relatively absorbing cases.

### 3.6.3 Spatial variability of retrieved SSA

Compared with Fig.4b, the spatial variability of SSA retrieved by SVR is less strong (Fig.10-14), whose difference between maximum and minimum SSA falls in range from 0.09 to 0.10 (Table 4). In the first experiment, SSA is determined by UVAI for each pixel individually. In the SVR model, the spatial variability of the intermediate output AAOD depends on the three input features. Furthermore, SVR predicts SSA for each pixel based on the common relationship between UVAI, AOD and ALH in the training data set.

Heterogeneity in aerosol properties is expected for plume of this size, but to what extend needs further investigations. Here we assess the SSA spatial variability of by an independent data set. We employ the SSA calculated by AOD and scattering AOD from MERRA-2 aerosol reanalysis hourly single-level product (https://disc.gsfc.nasa.gov/datacollection/M2T1NXAER_5.12.4.htm last access: 16 July 2019). The AOD and aerosol properties of MERRA-2 are proved to be in good agreement with independent measurements (Buchard et al., 2017;

Randles et al., 2017). The MERRA-2 AOD and SSA for these cases are shown in Appendix C. The plume can be detected by the high AOD against its surrounding. Although the plume presented by the satellite observations significantly differs from that of model simulations, the SSA spatial difference within the plume is approximately at magnitude of 0.1. From this aspect, the spatial variability of SSA retrieved by the SVR model is in better agreement with MERRA-2.

**4 Conclusions**

The long-term record of global UVAI data is a treasure to derive aerosol optical properties such as SSA, which is important for aerosol radiative forcing assessments. To quantify aerosol absorption from UVAI, the information of AOD and ALH is necessary. There are various AOD products while ALH products are much less accessible. Recently, the TROPOMI oxygen A-band ALH product has been run operationally, using which we demonstrate the role of ALH in quantifying SSA from satellite retrieved UVAI for biomass burning aerosols.

In the first experiment, we derive the SSA by forward radiative transfer simulation of UVAI for a fire event in California on 2017-12-12. With the TROPOMI ALH, we are able to quantify the influence of assumed spectral dependence of near-UV aerosol absorption (represented by the relative difference between $\kappa_{354}$ and $\kappa_{388}$) on the retrieved SSA. A significant gap in plume mean SSA (0.25) between 'gray' and strong spectral dependent aerosols ($\Delta\kappa$=0% and 40%, respectively) implies that inappropriate assumptions on spectral dependence may significantly bias the retrieved aerosol absorption. The SSA difference between AERONET and collocated pixels becomes smaller than the uncertainty of AERONET ($\pm$0.03) when $\Delta\kappa$=25%. The corresponding plume SSA of 0.90$\pm$0.05 and the AERONET-collocated pixels SSA of 0.96$\pm$0.02.

In the second part of this paper, we propose a statistical method based on the long-term records of UVAI, AOD, ALH and AAOD using an SVR algorithm, in order to avoid making assumptions on aerosol absorption spectral dependence over near-UV band. The SVR model is trained by 5679 collocated global observations from OMAERUV and AERONET during the period from 2005 to 2017. The SVR-retrieved SSA for the California fire event on 2017-12-12 is 0.97$\pm$0.01, which is 0.01 lower than that of AERONET. The SVR algorithm is also applied to other cases. Consider all case applications, the results are encouraging: the SSA discrepancy between retrieval and AERONET for almost all collocated samples are within $\pm$0.05 difference and over half of them fall in the AERONET uncertainty range ($\pm$0.03). One should keep in mind the SVR model tends to overestimate the SSA for relatively absorbing cases (e.g. SSA<0.90), and sometimes fails to predict reasonable SSA when the input values fall outside the range of the corresponding parameters in the training data set.

In terms of spatial variability, the SSA retrieved by radiative transfer simulations significantly differs from that retrieved by SVR. Spatial heterogeneity in SSA is expected, but to what extent needs further investigations. We employ the SSA provided by MERRA-2 aerosol reanalysis as a reference, whose spatial difference within smoke plume is approximately at magnitude of 0.1. The spatial pattern of SSA retrieved by SVR agrees better with this finding.

In this study, we present the potential to retrieve SSA based on the long-term data records of UVAI, ALH, AOD and AAOD using a statistical method. The motivation is to avoid a priori assumptions on aerosol micro-physics as we made in radiative transfer simulations. At the current phase, the algorithm we choose is SVR as the size of the training data set is relatively small. The input features are selected by the Spearman's rank correlation coefficients and a priori knowledge on relationship between UVAI and only aerosol related features. The model hyper-parameters are analytically determined. The accuracy of SVR-predicted SSA is $\pm$0.02, with higher tendency to overestimate the SSA

for relatively absorbing cases. The OMAERUV-AERONET data set cannot cover some extreme situations, as a result, the SVR fails to predict reasonable SSA when the input values fall outside the range of the corresponding parameters in the training data set. In the future, more sophisticated feature selection techniques and kernel functions should be

considered to improve the accuracy the algorithm. Other non-aerosol features affecting UVAI should be also taken into consideration. Moreover, the high-resolution TROPOMI level 2 UVAI and ALH products are expected to significantly increase the size of training data set and improve the quality of the training data set, which will reduce the computational failures of the SVR model and even guide use to more powerful algorithms (e.g. ANN) to retrieve SSA.

**Acknowledgements.** This work was performed in the framework of the KNMI Multi-Annual Strategic Research (MSO). The authors thank NASA's GES-DISC and LAADS DAAC for free online access of OMI and MODIS data. The authors thank NASA Goddard Space Flight Center AERONET Project for providing the data from the AERONET. The authors thank Swadhin Nanda for providing TROPOMI aerosol layer height product. The authors thank Deborah Stein Zweers for providing TROPOMI UVAI degradation plot.

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

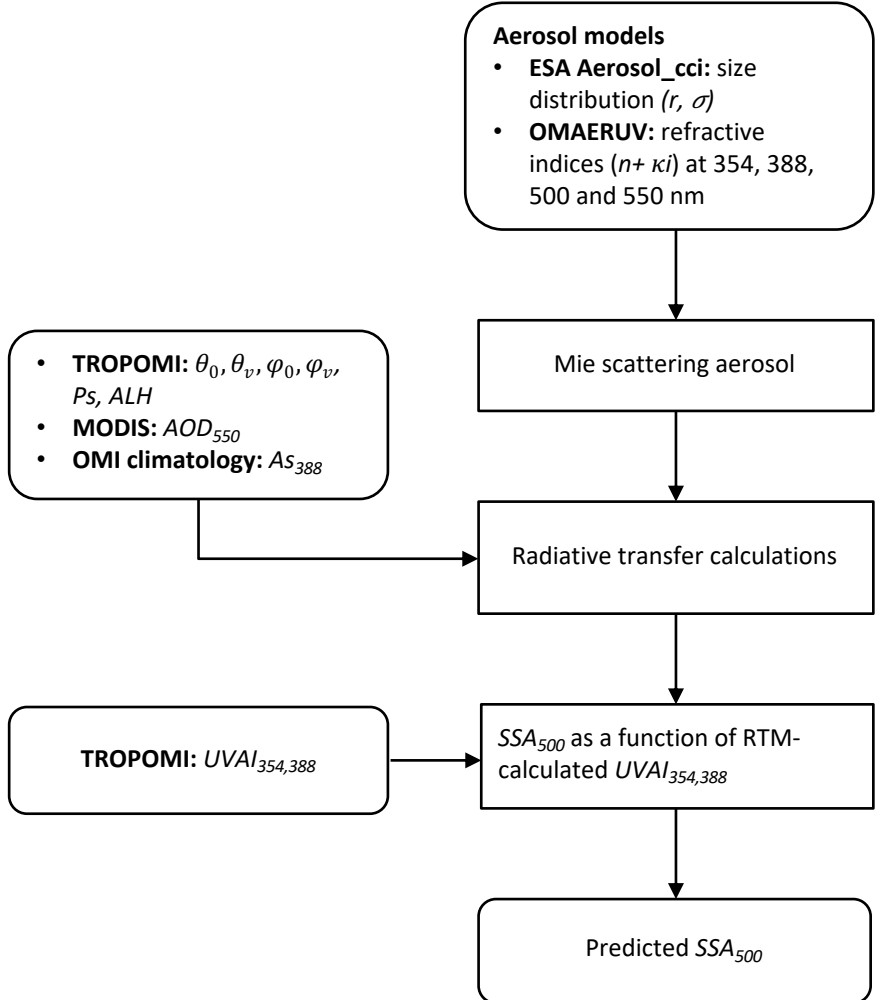


**Figure 1: Procedure of the radiative transfer simulation of UVAI. The aerosol models come from that of ESA Aerosol_cci (Holzer-Popp et al., 2013) and that of OMAERUV algorithm (Torres et al., 2007; Torres et al., 2013). The satellite inputs are the TROPOMI measurement geometry and ALH, the MODIS AOD and the OMI surface climatology. The aerosol profile is parameterized as a one-layered box shape profile, with the central layer height set to be the TROPOMI ALH and an assumed**

**constant pressure thickness of 50 hPa.**

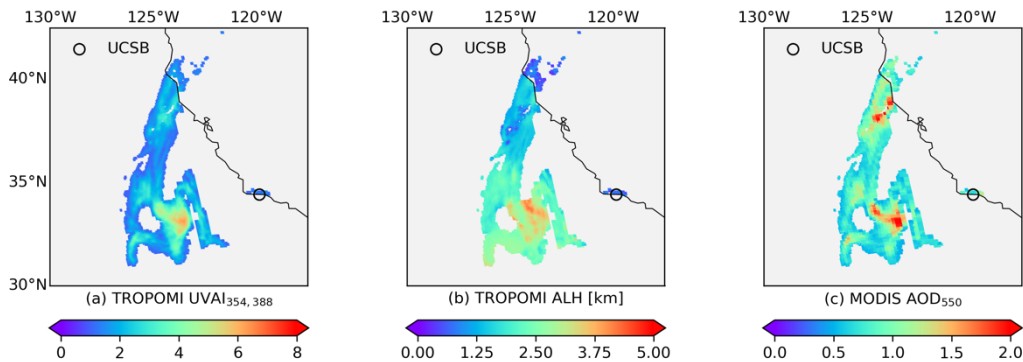

**Figure 2: Satellite data from California fire event on 2017-12-12: (a) TROPOMI UVAI calculated by reflectance at 354 and 388 nm; (b) TROPOMI ALH (unit: km); (c) MODIS AOD at 550 nm.**

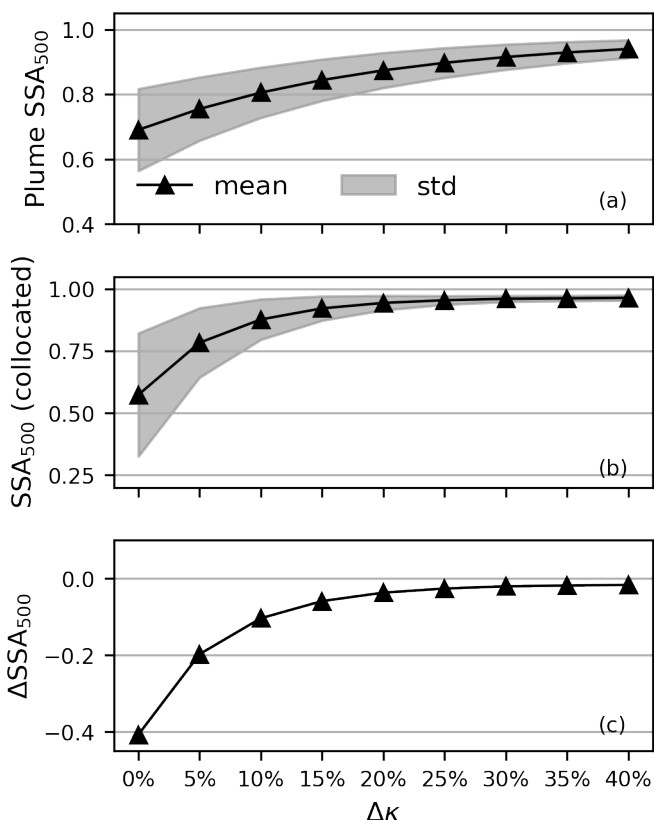

**Figure 3: SSA retrieved by radiative transfer simulations as a function of $\Delta\kappa$ ($\Delta\kappa = (\kappa_{354} - \kappa_{388})/\kappa_{388}$): (a) SSA mean and standard deviation (filled region) of plume pixels; (b) SSA mean and standard deviation (filled region) of the 15 AERONET-collocated pixels; (c) absolute difference between the mean SSA of the 15 collocated pixels and the AERONET retrieval.**

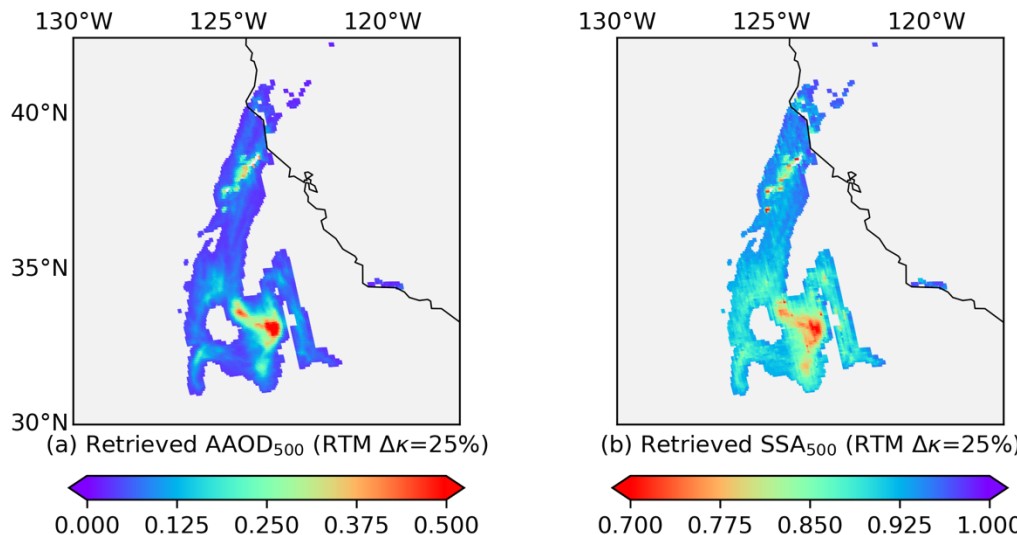

**Figure 4: Retrievals of radiative transfer simulations for California fire event on 2017-12-12 when Δκ=25% ($\Delta\kappa = (\kappa_{354} - \kappa_{388})/\kappa_{388}$): (a) retrieved AAOD at 500 nm; (b) retrieved SSA at 500nm.**


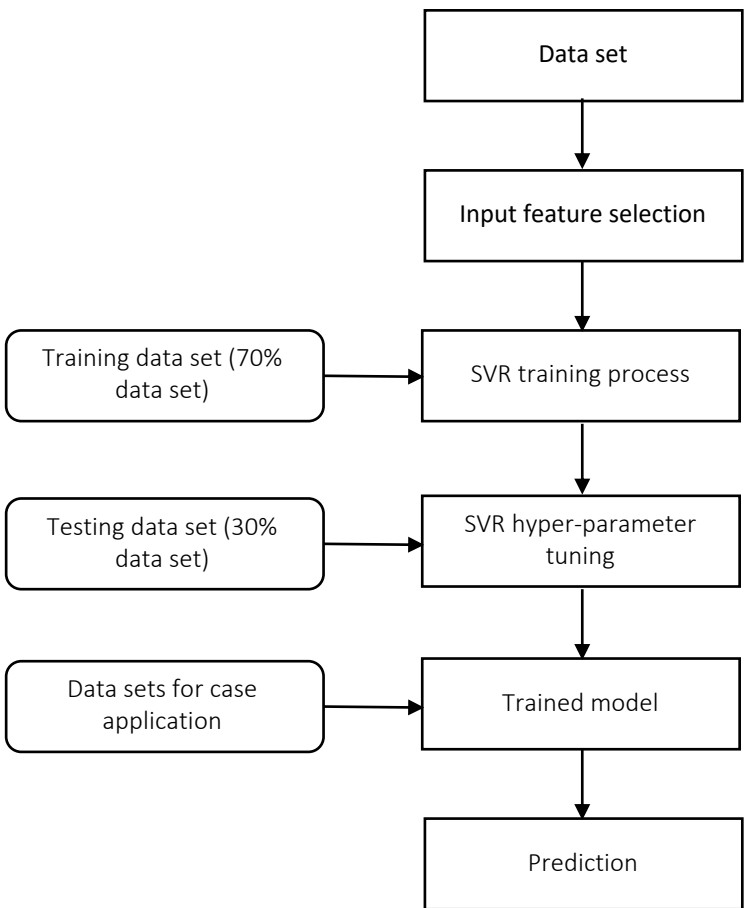

**Figure 5: Procedure of the support vector regression (SVR).**


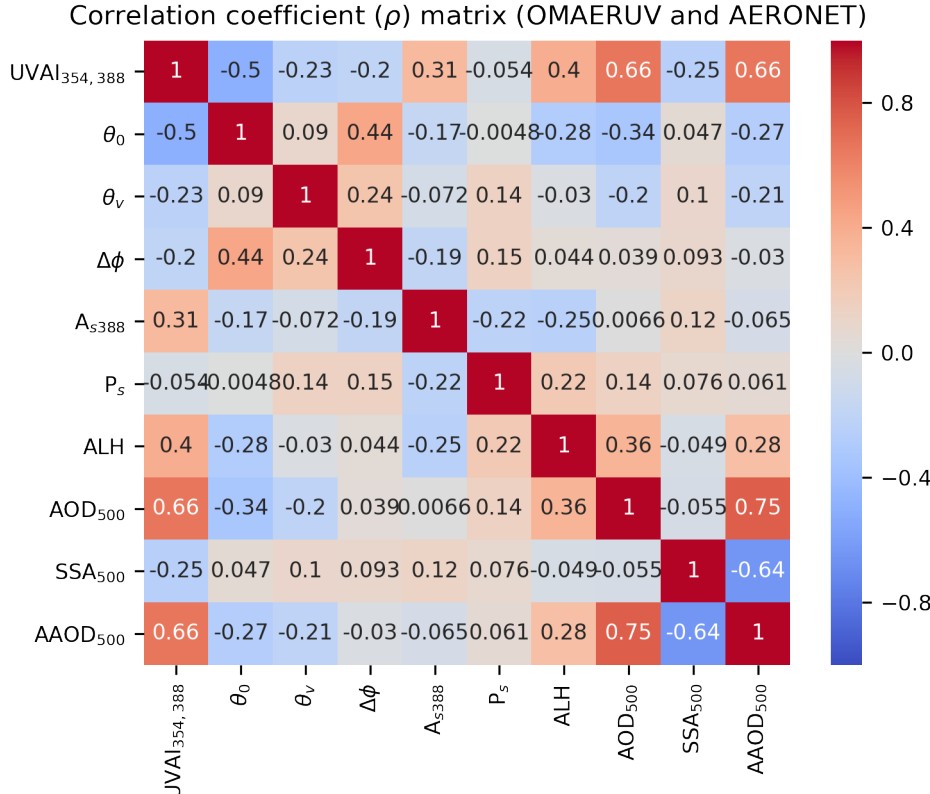

**Figure 6: Spearman's rank correlation coefficient matrix ($\rho$) of parameters in the OMAERUV-AERONET joint data set.**





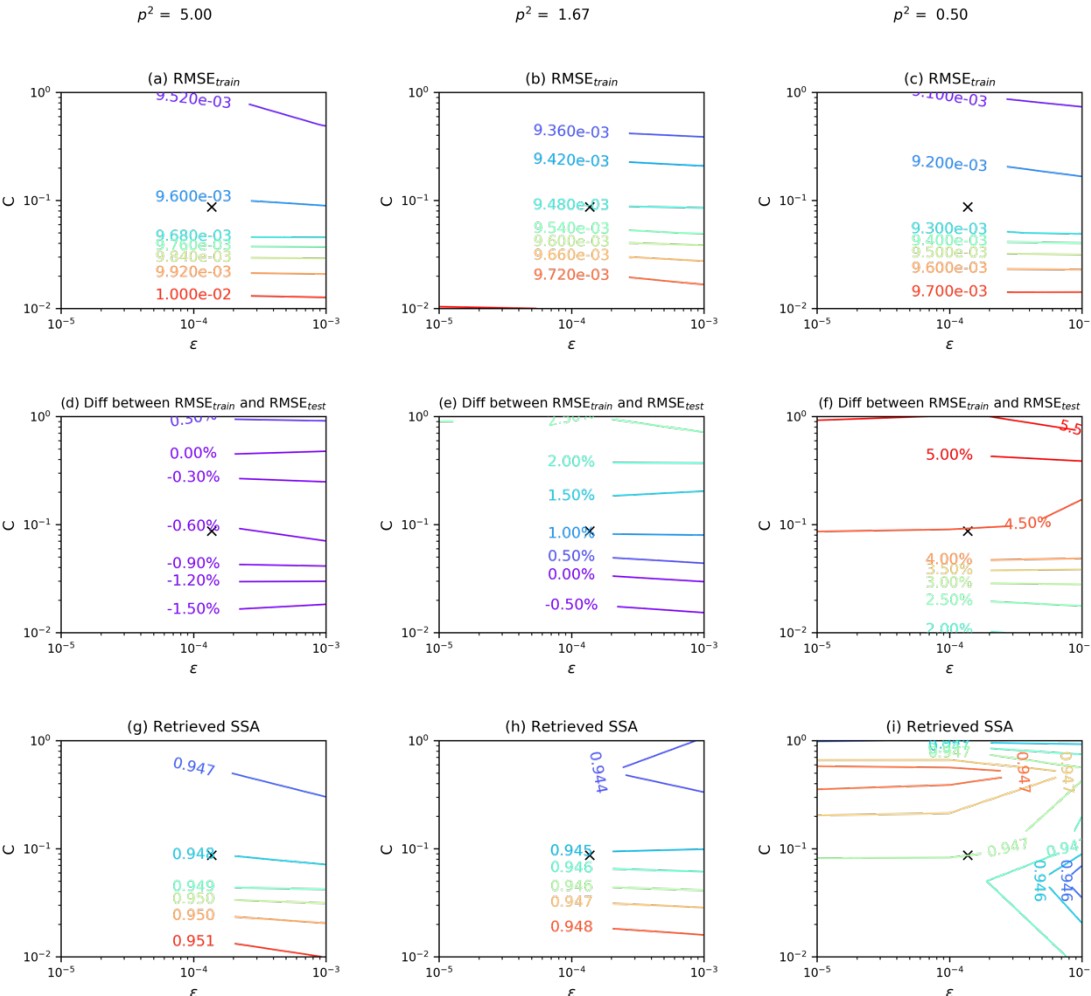

**Figure 7: The performance of SVR model as a function to hyper-parameters (C, $\varepsilon$ and $p$). The cross marker represents the values of C and $\varepsilon$ according to Cherkassky and Ma (2004). $p^2$ equaling 1.67 is sufficient to obtain a relatively high accuracy, meanwhile prevents overfitting on the training data set.**

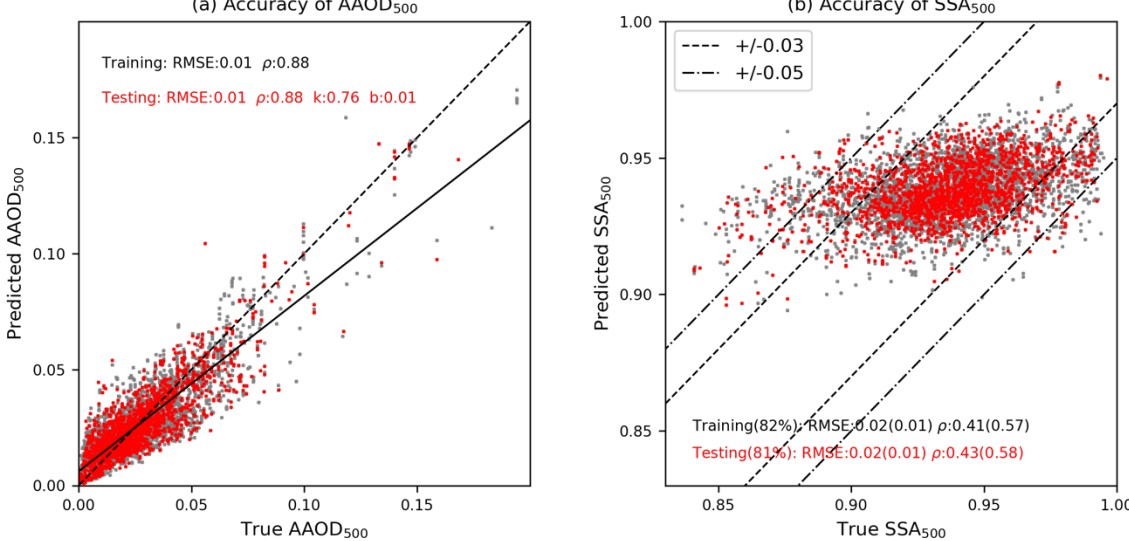

**Figure 8: The accuracy of the trained SVR model: (a) the predicted AAOD at 500 nm against true AAOD at 500 nm. The dashed line is the 1:1 line and the solid line is the linear fitting for the testing data set; (b) the predicted SSA at 500 nm against true SSA at 500 nm. The grey and red color indicates samples in training and testing data set, respectively. The values inside parenthesis is the statistics for samples fall in AERONET uncertainty of 0.03.**

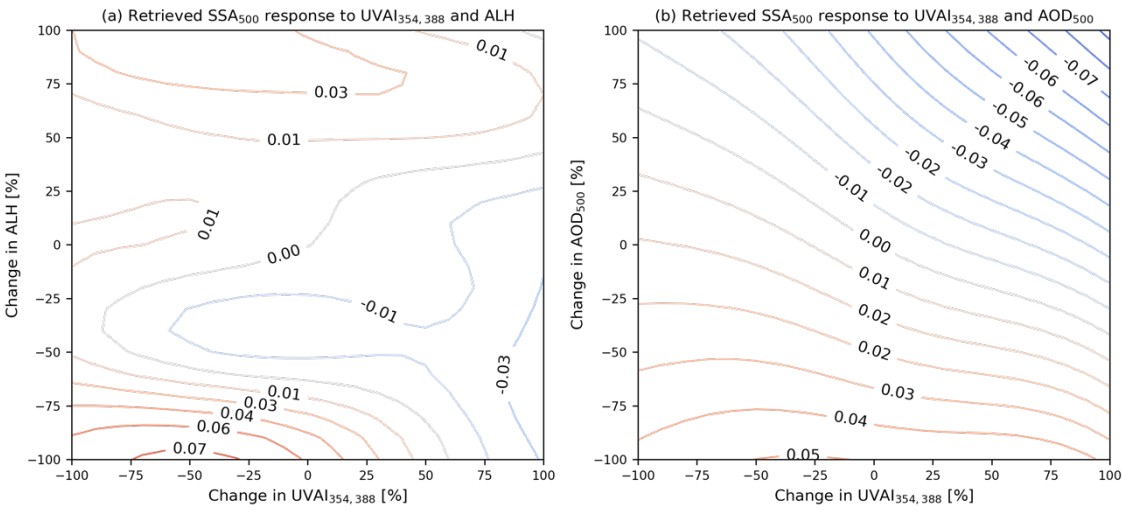

**Figure 9: The sensitivity of the SVR-retrieved SSA: (a) the response of predicted SSA at 500 nm as a function of changes in UVAI and ALH; (b) the response of predicted SSA at 500 nm as a function of changes in UVAI and AOD.**

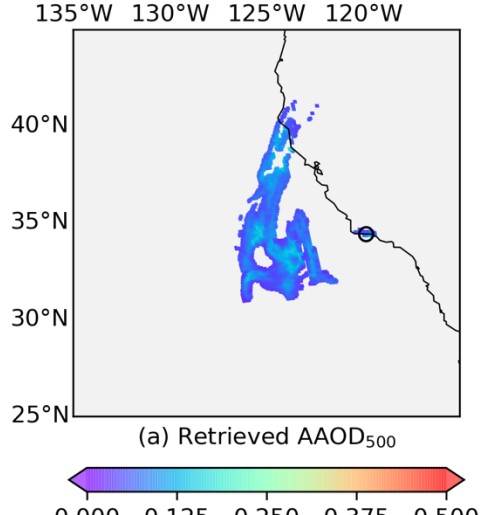
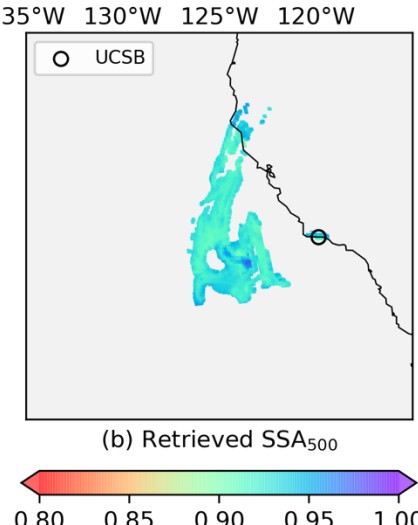

**Figure10: SVR retrievals for California fire event on 2017-12-12: (a) retrieved AAOD at 500 nm; (b) retrieved SSA at 500 nm.**


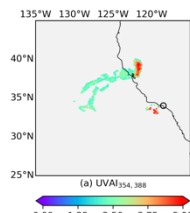
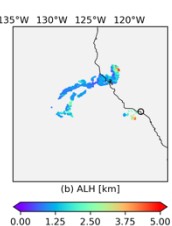
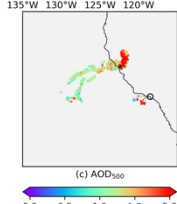
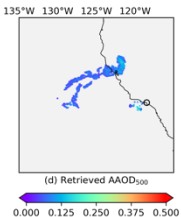
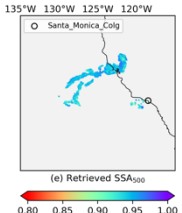

**Figure11: SVR retrievals for California fire event on 2018-11-09: (a) TROPOMI UVAI calculated by reflectance at 354 and 388 nm; (b) TROPOMI ALH; (c) MODIS AOD at 550 nm; (d) retrieved AAOD at 500 nm; (e) retrieved SSA at 500 nm.**

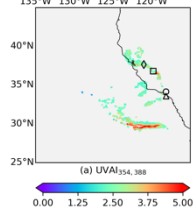
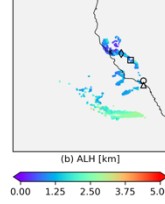
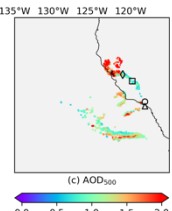
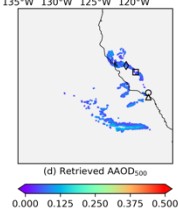
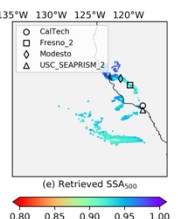


**Figure12: SVR retrievals for California fire event on 2018-11-10: (a) TROPOMI UVAI calculated by reflectance at 354 and 388 nm; (b) TROPOMI ALH; (c) MODIS AOD at 550 nm; (d) retrieved AAOD at 500 nm; (e) retrieved SSA at 500 nm.**

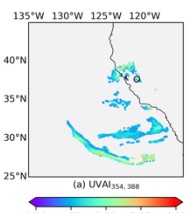
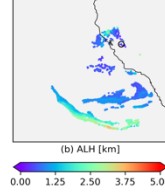
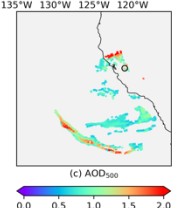
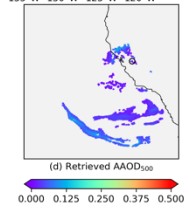
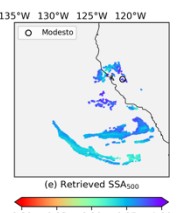

**Figure13: SVR retrievals for California fire event on 2018-11-11: (a) TROPOMI UVAI calculated by reflectance at 354 and 388 nm; (b) TROPOMI ALH; (c) MODIS AOD at 550 nm; (d) retrieved AAOD at 500 nm; (e) retrieved SSA at 500 nm.**

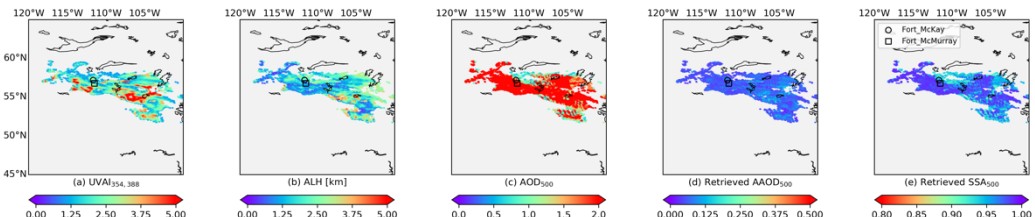

**Figure14: SVR retrievals for Canada fire event on 2019-05-29: (a) TROPOMI UVAI calculated by reflectance at 354 and 388 nm; (b) TROPOMI ALH; (c) MODIS AOD at 550 nm; (d) retrieved AAOD at 500 nm; (e) retrieved SSA at 500 nm.**

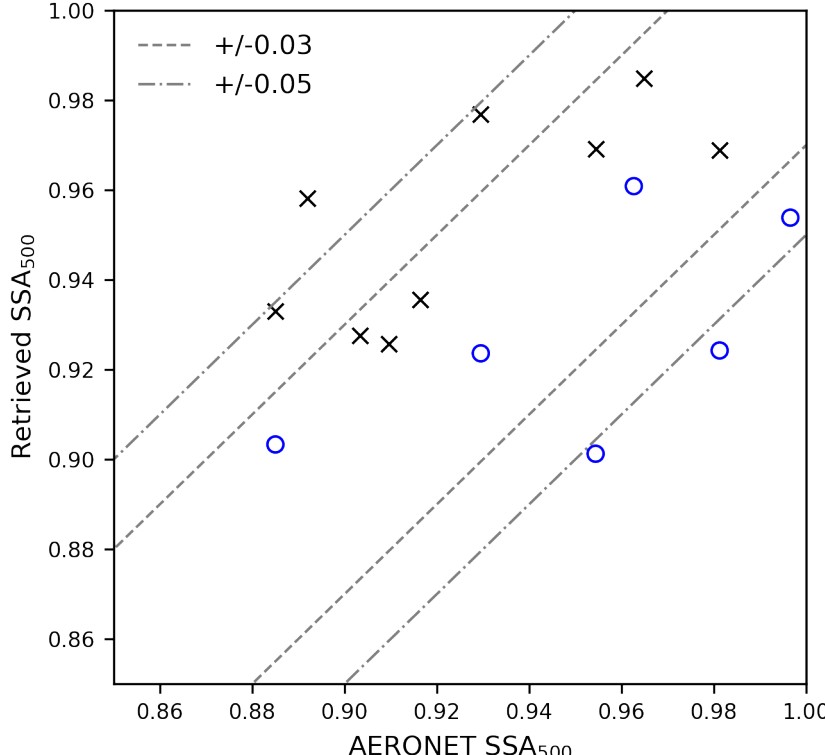

**Figure15: SVR-retrieved SSA (black cross) and OMAERUV-retrieved SSA (blue circle) against AERONET SSA at 500 nm for all 5 cases in this study.**

**Table 1 Aerosol models used in forward radiative transfer calculations. $\Delta\kappa$ is the relative difference between $\kappa_{354}$ and $\kappa_{388}$, defined as $\Delta\kappa = (\kappa_{354} - \kappa_{388})/\kappa_{388}$.**

| Geometric radius $(r_g)$ | Effective radius $(r_{eff})$ | Geometry standard deviation $(\sigma_g)$ | Variance $(ln\sigma_{eff})$ | Refractive index real part $(n)$ | Spectral dependence $(\Delta\kappa)$ | Refractive index imaginary part at 354 nm $(\kappa_{354})$ | Refractive index imaginary part of other wavelengths $(\geq 388\ nm)$ |
|---|---|---|---|---|---|---|---|
| 0.07 μm | 0.14 μm | 1.7 | 0.53 | 1.5 | 0%, 5%, 10%, 15%, 20%, 25%, 30%, 35% and 40% | $(1 + \Delta\kappa)\times\kappa_{388}$ | 0.005 |
| | | | | | | | 0.010 |
| | | | | | | | 0.020 |
| | | | | | | | 0.030 |
| | | | | | | | 0.040 |
| | | | | | | | 0.048 |
| | | | | | | | 0.060 |

**Table 2 Retrieved SSA by the radiative transfer simulations for the California fire on 2017-12-12.**

| Retrieval methods | Number of plume pixels | Retrieved SSA (plume pixels) | $SSA_{max} - SSA_{min}$ | Retrieved SSA (collocated-pixels) | AERONET SSA | OMAERUV SSA |
|---|---|---|---|---|---|---|
| RTM with $\Delta\kappa$=25% | 5217 | 0.90±0.05 | 0.38 | 0.95±0.02 | 0.98 | 0.92±0.01 |

**Table 3 Values for hyper-tuning decided regularization constant C, the width of the insensitive zone $\varepsilon$ and the BRF kernel parameter $p^2$.**

| | SVR hyper-parameters | | |
|---|---|---|---|
| Parameters | C | $\varepsilon$ | $p^2$ |
| Values | 0.09 | 0.0001 | 1.67 |

**Table 4 SVR-retrieved SSA. If there is no standard deviation followed, then it indicates there is only one record.**

| Case | Num. of Plume pixels | Retrieved SSA (plume pixels) | $SSA_{max} - SSA_{min}$ | Collocated AERONET | SSA (collocated-pixels) | AERONET SSA | OMAERUV SSA |
|---|---|---|---|---|---|---|---|
| California 2017-12-12 | 5217 | 0.94±0.01 | 0.09 | UCSB | 0.97±0.01 | 0.98 | 0.92±0.01 |
| California 2018-11-09 | 1944 | 0.94±0.01 | 0.10 | Santa_Monica_Colg | 0.93±0.01 | 0.89±0.06 | 0.89±0.06 |
| California 2018-11-10 | 2184 | 0.94±0.02 | 0.10 | CalTech | 0.96±0.01 | 0.89±0.07 | - |
| | | | | Fresno_2 | 0.93±0.02 | 0.91±0.01 | - |
| | | | | Modesto | 0.94±0.01 | 0.92±0.01 | 0.96±0.01 |
| | | | | USC_SEAPRISM_2 | 0.93±0.00 | 0.90 | - |
| California 2018-11-11 | 2815 | 0.95±0.02 | 0.09 | Modesto | 0.98±0.00 | 0.96±0.01 | 0.95±0.00 |
| Canada 2019-05-29 | 8013 | 0.97±0.02 | 0.10 | Fort_McKay | 0.97±0.02 | 0.95±0.00 | 0.93 |
| | | | | Fort_McMurray | 0.98±0.01 | 0.93 | 1.00 |

**Appendix**

**Part A: Case information**

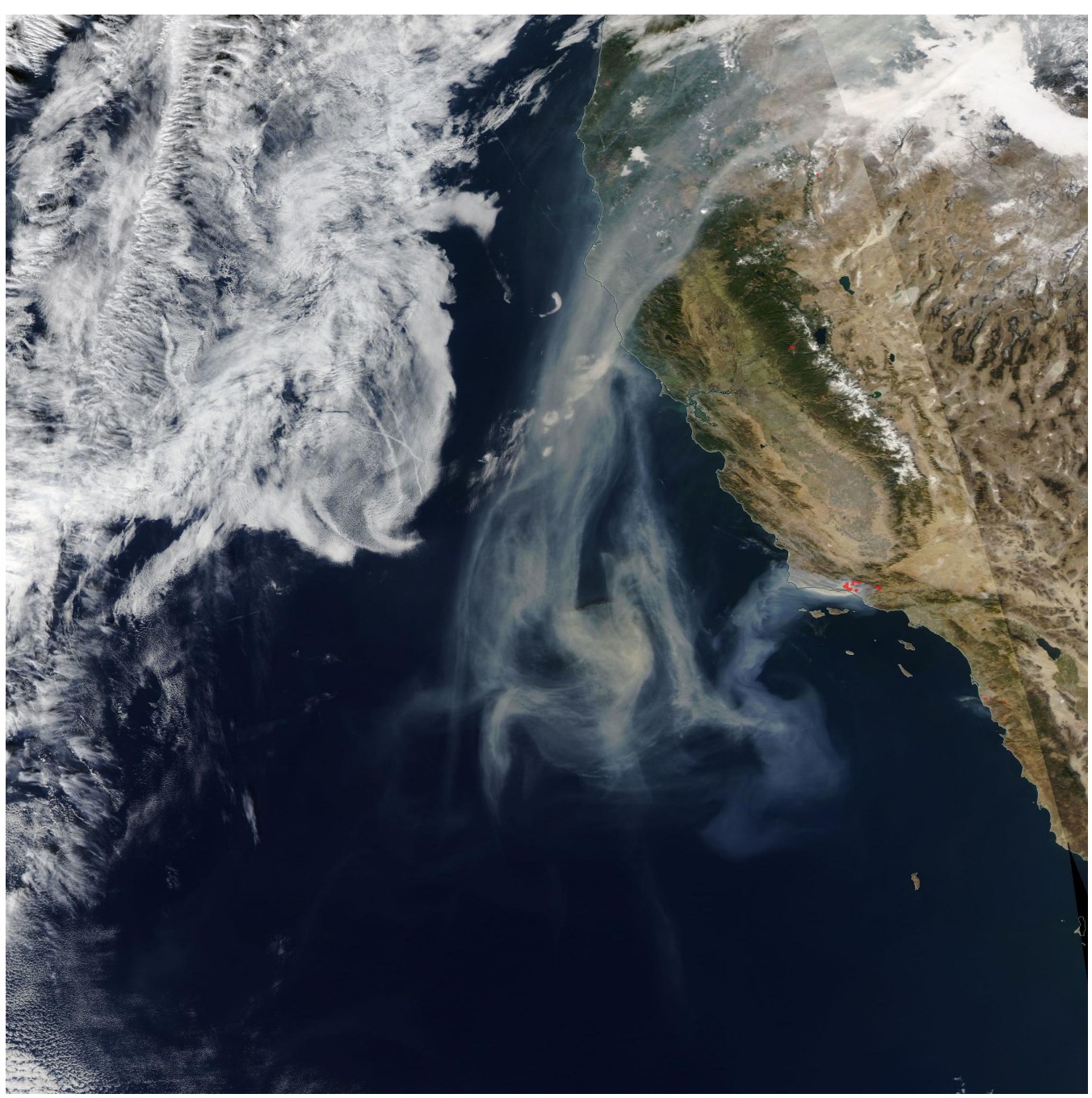

**Figure A1: Smoke plume captured by Aqua MODIS for California fire event on 2017-12-12**
**(source:https://gibs.earthdata.nasa.gov). The red regions indicate fires and thermal anomalies.**

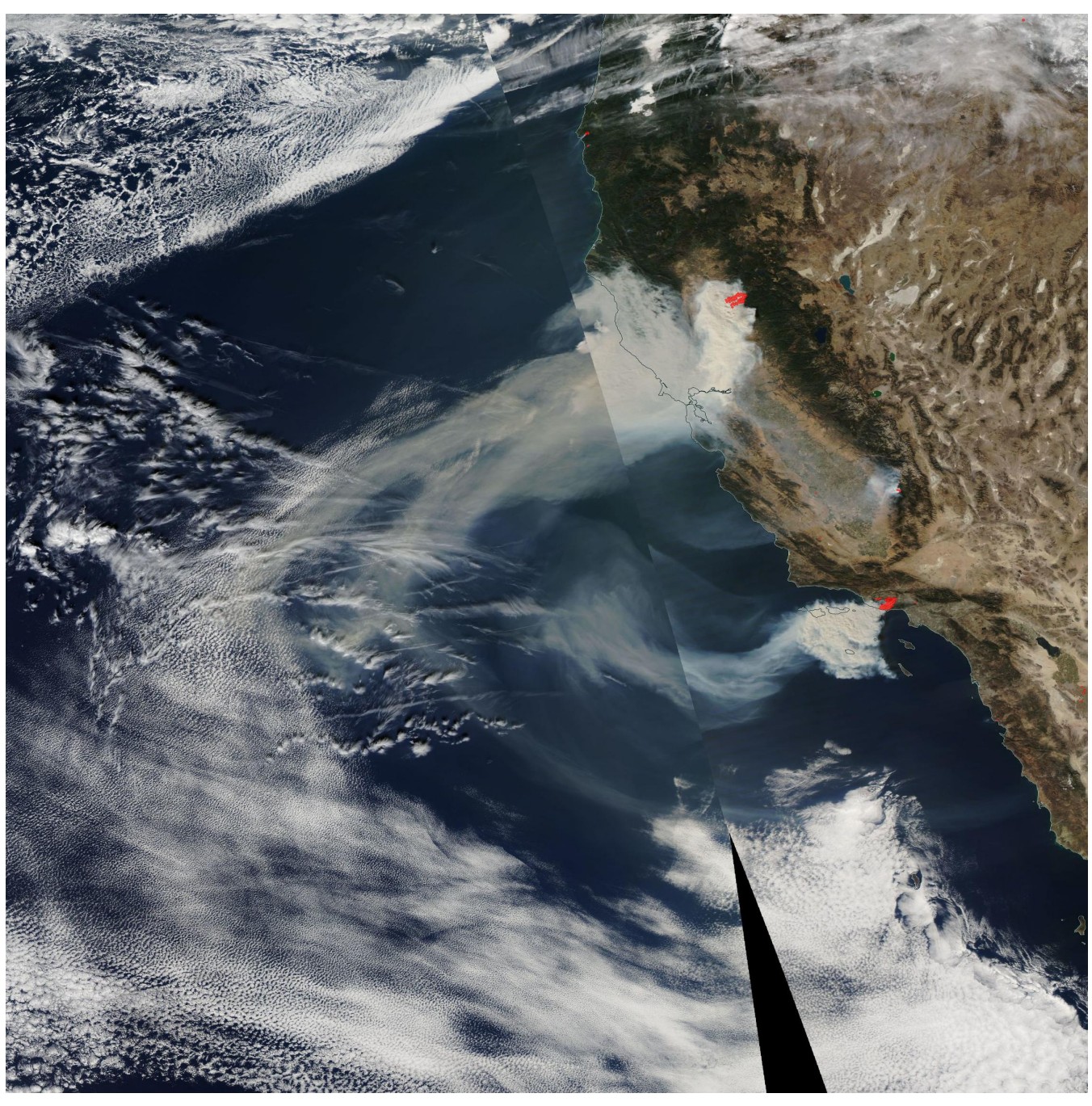

**Figure A2: Smoke plume captured by Aqua MODIS for California fire event on 2018-11-09
(source: https://gibs.earthdata.nasa.gov). The red regions indicate fires and thermal anomalies.**

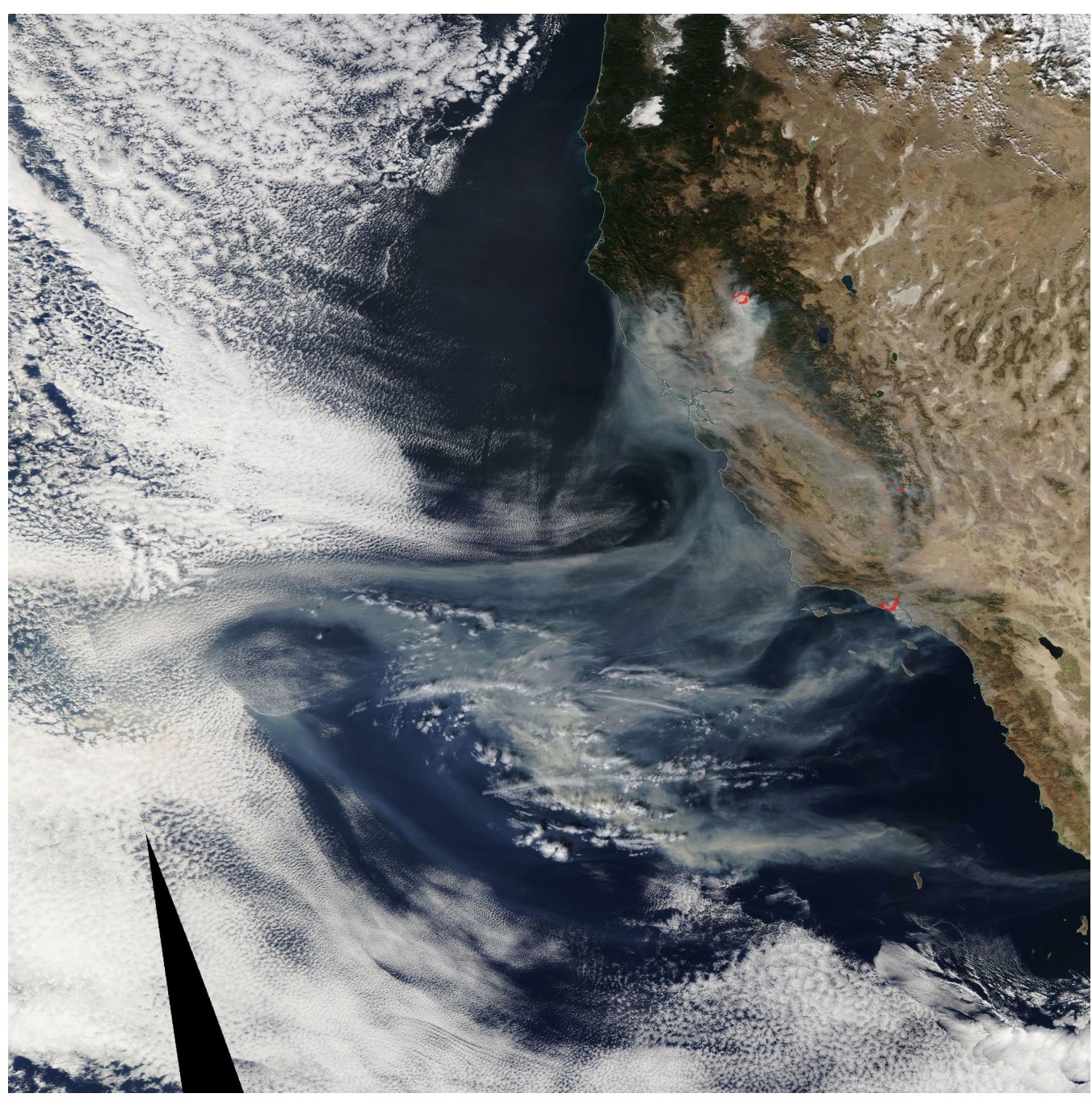

**Figure A3: Smoke plume captured by Aqua MODIS for California fire event on 2018-11-10 (source: https://gibs.earthdata.nasa.gov). The red regions indicate fires and thermal anomalies.**

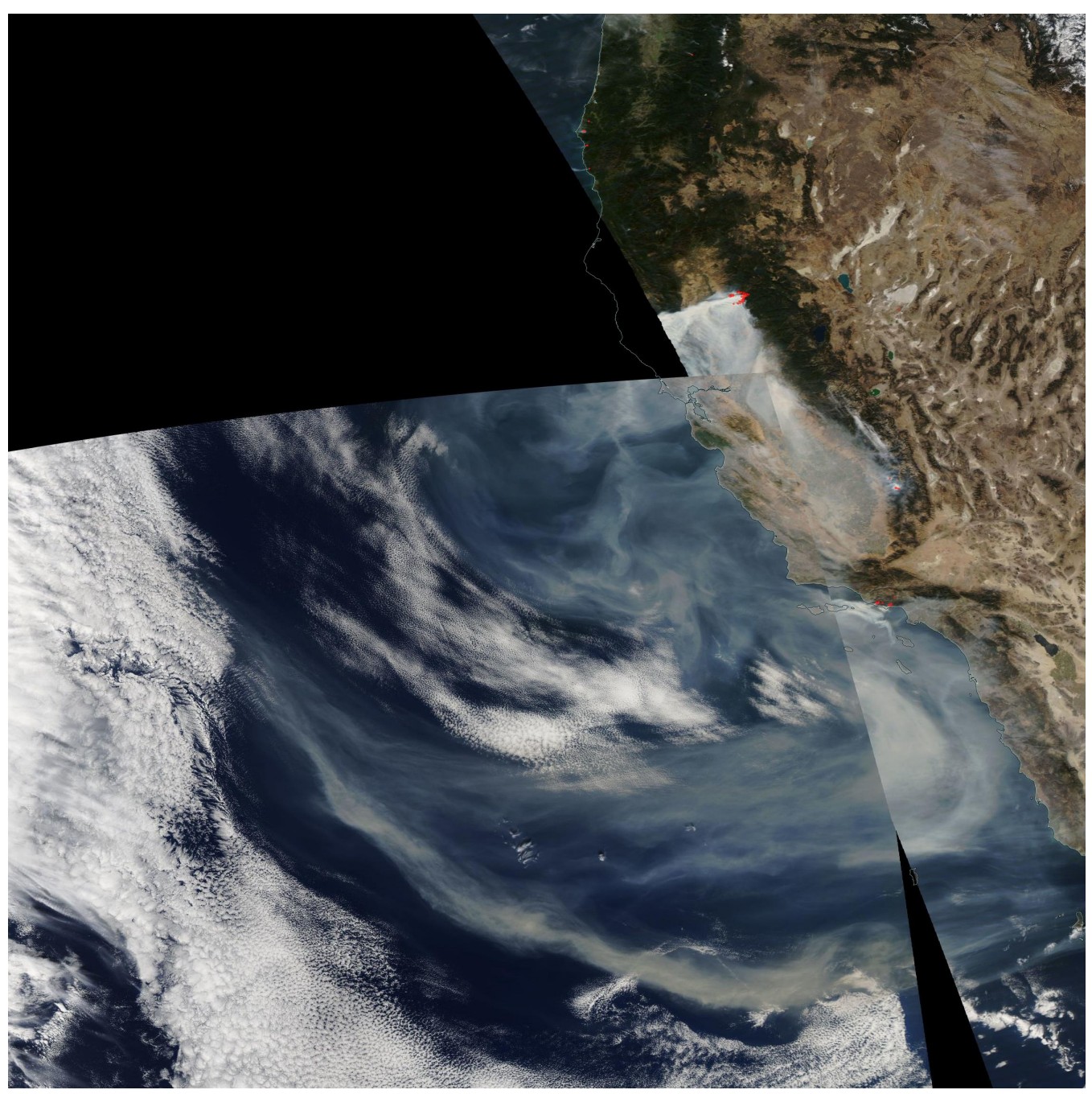

**Figure A4: Smoke plume captured by Aqua MODIS for California fire event on 2018-11-11**
**(source: https://gibs.earthdata.nasa.gov). The red regions indicate fires and thermal anomalies.**

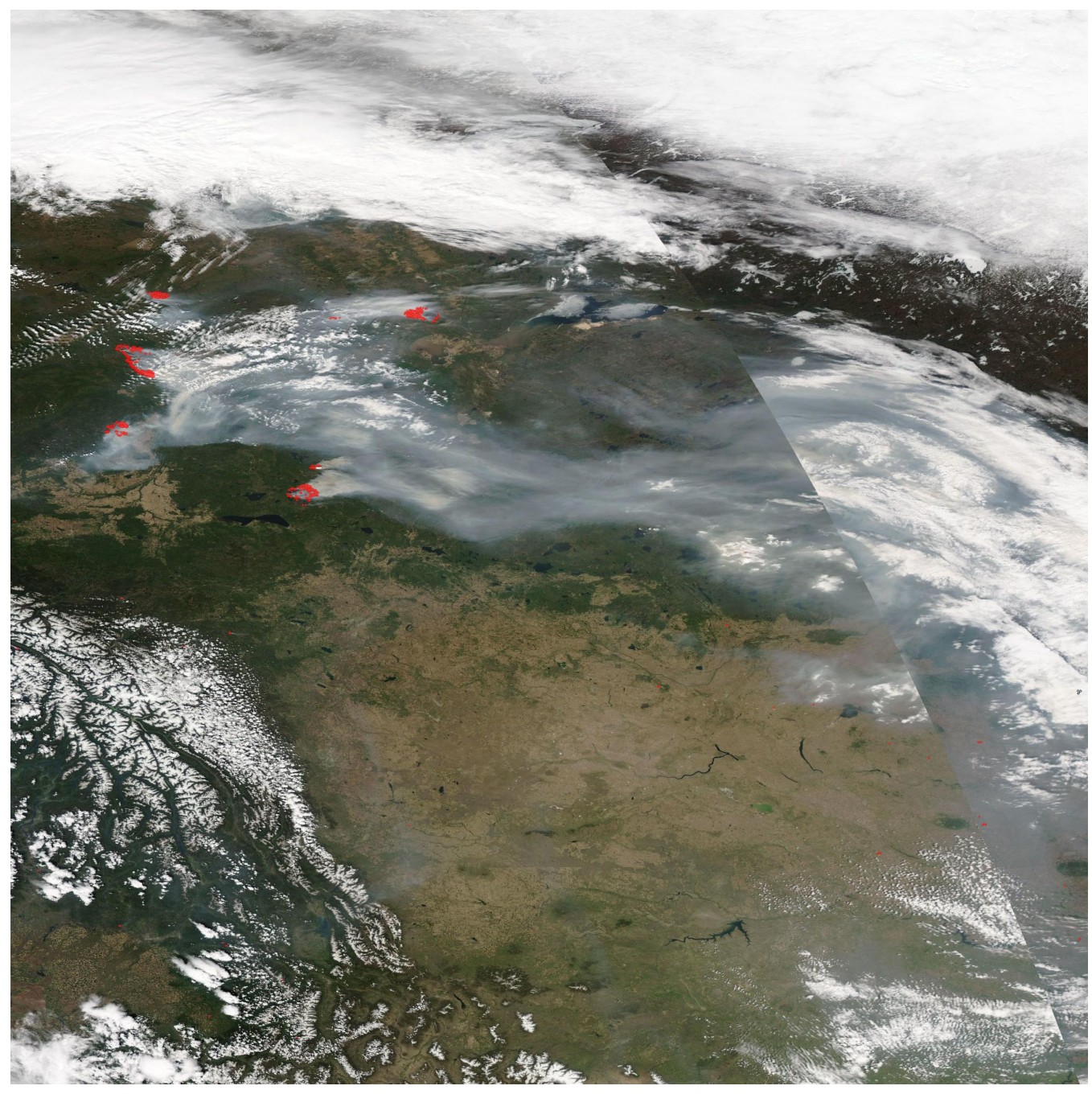

**Figure A5: Smoke plume captured by Aqua MODIS in for Canada fire event on 2019-05-29 (source: https://gibs.earthdata.nasa.gov). The red regions indicate fires and thermal anomalies.**

**Part B: OMI-AERONET joint data set (based on global data from 1 January 2005 to 31 December 2017).**

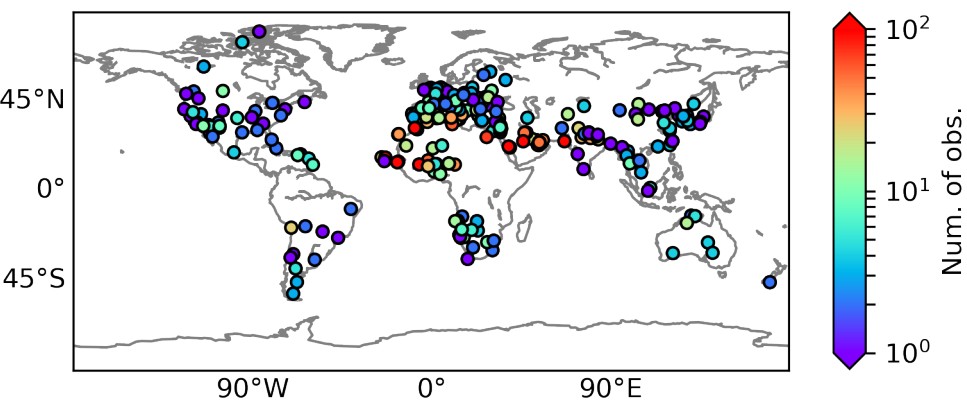

**Figure B1: Global distribution of OMAERUV-AERONET joint data set. The color indicates the number of observations. Note that all aerosol types are included.**

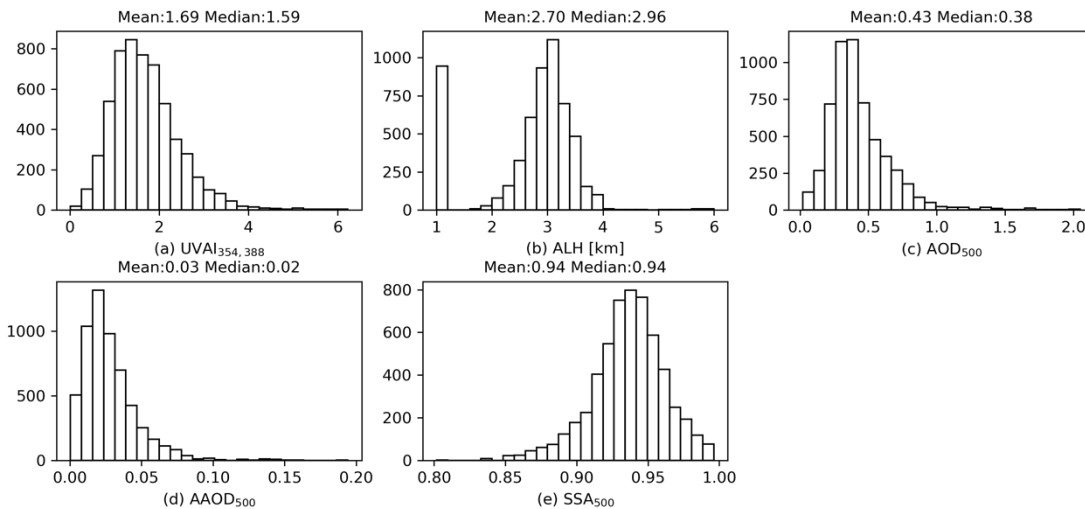

**Figure B2: Statistics of the OMAERUV-AERONET joint data set: (a) OMAERUV UVAI calculated from reflectance at 354 and 388 nm; (b) OMAERUV ALH; (c) AERONET AOD at 500 nm; (d) AERONET AAOD at 500 nm; (e) AERONET SSA at 500 nm.**

**Part C: MERRA-2 aerosol reanalysis.**

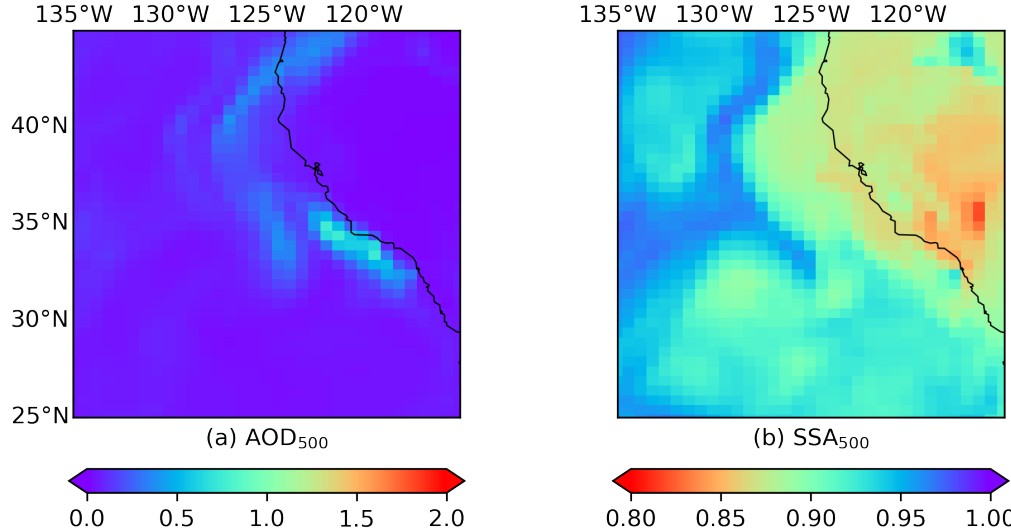

**Figure C1: MERRA-2 M2T1NXAER averaged between 12:00 and 15:00 local time for California fire event on 2017-12-12: (a) AOD at 500 nm; (b) SSA at 500 nm.**

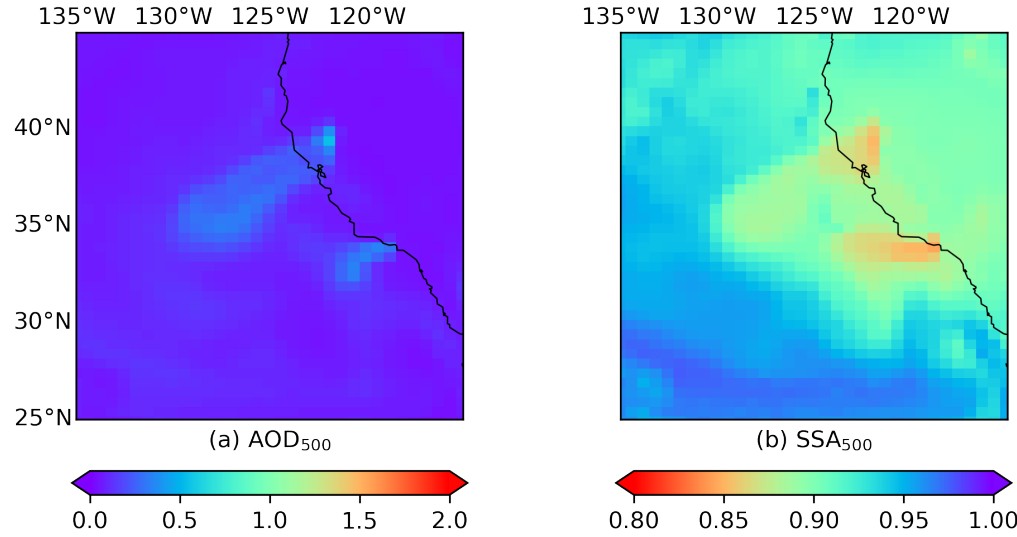

**Figure C2: MERRA-2 M2T1NXAER averaged between 12:00 and 15:00 local time for California fire event on 2018-11-09: (a) AOD at 500 nm; (b) SSA at 500 nm.**

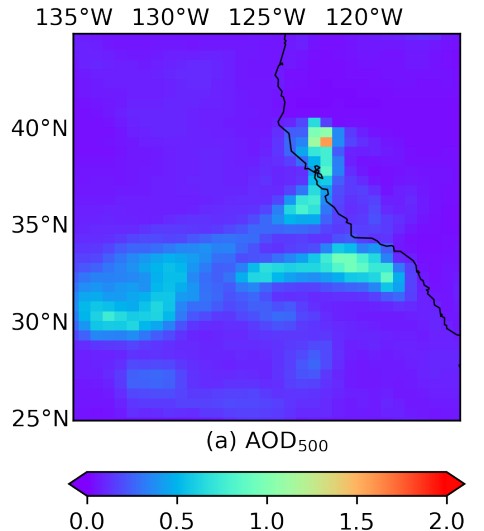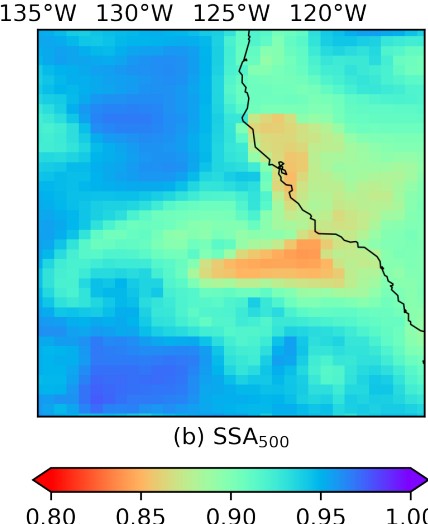


**Figure C3: MERRA-2 M2T1NXAER averaged between 12:00 and 15:00 local time for California fire event on 2018-11-10: (a) AOD at 500 nm; (b) SSA at 500 nm.**

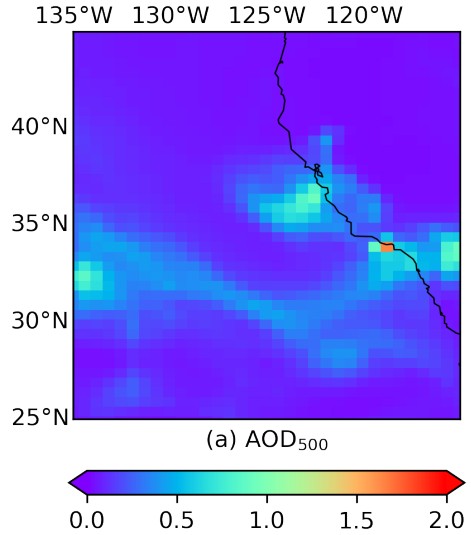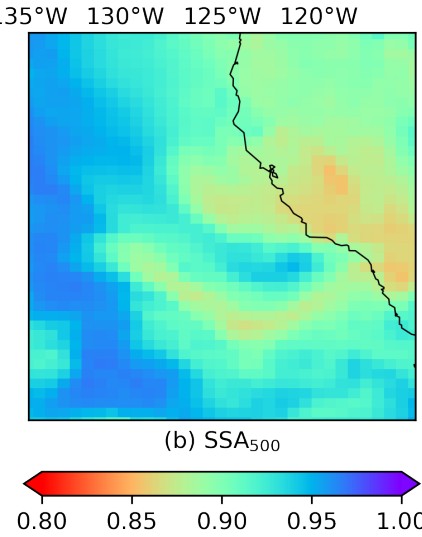

**Figure C4: MERRA-2 M2T1NXAER averaged between 12:00 and 15:00 local time for California fire event on 2018-11-11: (a) AOD at 500 nm; (b) SSA at 500 nm.**


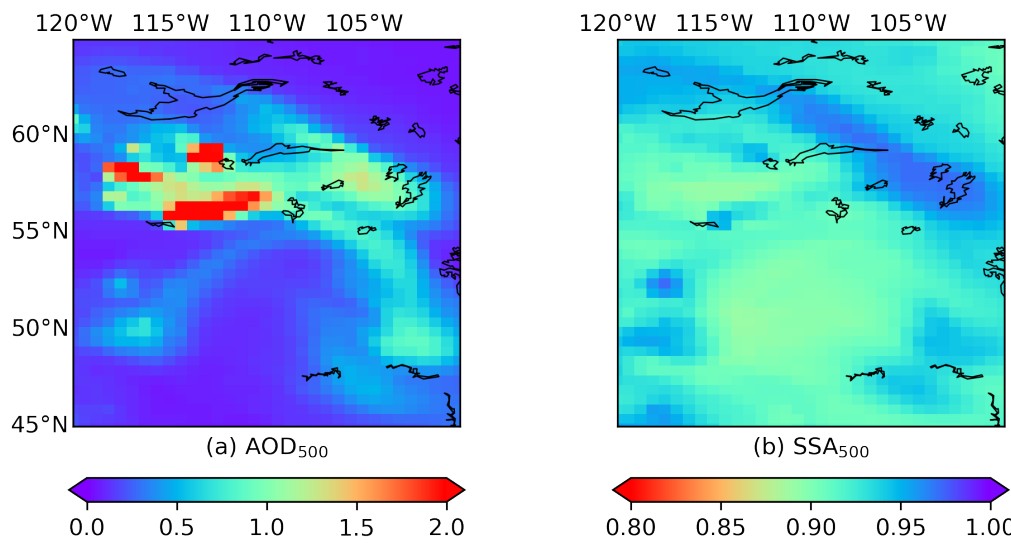

**Figure C5: MERRA-2 M2T1NXAER averaged between 12:00 and 15:00 local time for Canada fire event on 2019-05-29: (a) AOD at 500 nm; (b) SSA at 500 nm.**