# Peer review of "The role of aerosol layer height in quantifying aerosol absorption from ultraviolet satellite observations"

_Atmospheric Measurement Techniques, 2019_

## Editor Comment (EC1) · Ben Veihelmann (Editor) · 20 May 2019

OVERALL

The manuscript deals with a hot topic, i.e. the constraining the aerosol Single Scattering Albedo (SSA) using satellite observations. This quantity is difficult to capture by observations and at the same time very important regarding the radiative forcing of aerosol. The proposed scheme to infer SSA information is new and of interest to the AMT community. The manuscript is suitable for publication in AMT after the issues below are addressed.

[Figure]

**GENERAL COMMENTS**

The manuscript does not read smoothly. Formulations need to be improved in many instances. The specific issues listed below cover a number of these instances but the list is not exhaustive.

The study is dealing with a single case (a plume of one specific emission event). It needs to be discussed how robust are findings.

The study approach needs to be explained upfront more clearly. The choices regarding the source of data (UVAI, ALH, AOD, SSA) used for training the SVM-based scheme, for evaluating the SVM-based algorithms, and for evaluating the RTM-based algorithms, needs to be clarified (at a high level upfront, in detail in the specific sections).

**SPECIFIC COMMENTS**

Line 12: It is not clear how SSA is retrieved, which algorithm is employed. Reference to "conventional radiative transfer simulations" is not sufficient.

Line 13: The approach to constraining the SSA retrieval is not clear. Is the ALH fixed in the forward model used in the SSA retrieval?

Line 17: The sentence "In the second part of this paper, we propose..." is not clear. Clarify that the method relies on an empirical relation that has been established based on long-term datasets of UVAI, ALH and AOD based on the SVR concept. The term "data-driven" is misleading.

Line 20 (also caption Figure 8): AERONET does not "measure" SSA directly but retrieves it. Reformulate.

Eq. 1 is unclear and the variables are not introduced. Without more information the reader cannot guess how to interpret the superscribed labels "obs" and "Ray". It is recommended to explain the UVAI concept and highlight that the obtained index is

sensitive to elevated absorbing aerosol.

Line 41: What is meant by "with various spectral choices"?

Line 68: "quantitatively determine" –> quantify.

Line 72: data-driven –> empirical. Proposed to reformulate "We propose an empirical [. . .]. ML algorithms learn [. . .]".

Line 77: one piece of information is missing: does the training of an SVM requires less training data than ANN?

Line 77: ML and SVM seem to be used interchangeably, which not entirely correct: also ANN can be seen as ML tools.

Line 79, 81: inconsistent use of singular and plural

Line 84: The term kernel functions is used as if it had been introduced already. Are these related to the support vectors?

Line 107: reformulate "TROPOMI ALH retrieval is based on the pattern . . ."

Line 102 "For the forward radiative transfer calculations, the input aerosol profile is parameterized as . . ." is this choice consistent with the assumptions made in the ALH algorithm?

Line 129: The relevance of the reference to Herman & Celarier is not clear. Does the statement "A spectrally flat As is assumed . . ." apply to the OMI LER product? Or do you need to make this assumption?

Section 3.1 falls short on an explicit and upfront specification of the source of the input data (such as AOD, ALH, UVAI) used for the RTM-based method. A discussion of temporal mis-registration between MODIS and TROPOMI data acquisitions is missing.

Line 156/157: it is not clear what are the implication of using surface reflectance data from the OMI LER for reproducing the UVAI using the RTM method. Please discuss.

It is assumed that the surface reflectance generated within a UVAI product can be reproduced in a straight forward fashion if needed.

Line 161: The justification of the reporting wavelength of the retrieved SSA is not understood; in the end it is determined by the OMAERUV reporting wavelength?

Line 173: Aerosol models cannot be a combination of a project and an algorithm. Rephrase.

Line 175: the phrase "The particle size distribution . . ." needs grammatical/syntactic fixing

Line 179: subtype "BIO-1" is referred to without explanation/reference

Line 177: real PART OF THE refractive index

Line 178: imaginary PART OF THE refractive index

Line 181: Sentence incomplete

Line 182 (also caption Figure 8): The specification of $\Delta$ðÌIJĚ in % in not clear. Clarify.

Table 1: The specification of the imaginary part of the refractive indices ðÌIJĚ is unclear. For which reference wavelength are the numbers in the rightmost column valid? In the column "Refractive index imaginary part at 354 nm (k354) one expects an explicit list of values rather than a formula. Clarify.

Line 186: Absorbing Ångström Exponent –> Ångström exponent

Line 207: "13-year measurement OMAERUV and AERONET measurements" rectify formulation.

Line 210: an OMI pixel is collocated –> OMI observations are considered as collocated

Section 3.2.2 (Preparing training and testing data sets) is quite confusing. Please rewrite. Some terminology is used inconsistently (e.g. the terms "extra SVR", "adjusted ALH", "predicted ALH", or "ALH from OMAERUV" and "ALH from OMI"). Maybe

introduce Table 2 and Flow charts (Figure 5) already at the beginning of the section.

Line 240, 259, 326: It is referred to "ALH from the OMAERUV" suggesting that ALH is generated by the OMAERUV algorithm. It is stated in the manuscript that these ALH values are actually taken from CALIOP. Please refer to "ALH from CALIOP" for clarity.

Line 240: It is stated that the ALH from the OMAERUV product (actually from CALIOP) may not have sufficient quality. Clarify what is the concern. Co-location?

Line 242: What is meant with the OMI ALH? Is it the same as the ALH from the OMAERUV (CALIOP)?

Line 245: Is the TROPOMI ALH the one retrieved from the O2-A band?

Line 249: It is stated that the "extra" SVR is trained on the Thomas fire case. A training sets should cover more than one case. Please discuss the validity of the approach.

Line 253: it is noted that this "extra" SVR is a temporary intermediate step to obtain a better ALH". Please explain upfront the approach.

Line 255: It is stated that "there is no necessity to do this anymore once a reliable ALH product is accessible to build up training data sets, e.g. the TROPOMI ALH product that will be released in the near future". Clarify in which sense the training set using TROPOMI ALH is expected to outperform the training set using CALIOP ALH.

Line 259: What is meant with "The rule of thumb ratio is 70% versus 30%"?

Eq. 3: Introduce the variable n.

Eq. 3: What is the dimensionality of omega? What is meant with ||omega||? Some kind of norm? What is the dimensionality of omega?

Eq. 4: Introduce the variable x.

Eq. 4: Why introduce the kernel function K? What is done with it?

Section 3.2.4 Data for case application: Please report the number of validation samples

Figure 5: The figure shows at the same time SVR based ALH prediction and SVR based AAOD prediction, this is confusing. It would help to depict the two schemes for AAOD prediction in one flow chart, and the ALH prediction in a separate one.

Figure 6 (also Line 206): Why is the sign of the correlation coefficient not reported? Report the sign or justify and clarify in caption that |rho| is reported.

Figure 6: For which ALH parameter are the correlations reported? For the predicted one or for the one from CALIOP?

Figure 7: The 3D plot is hard to interpret. Recommended to replace it with 2D scatter-plots (AOD versus ALH) where the UVAI is only color-coded.

---

## Referee Comment (RC1) · Anonymous Referee #1 · 24 Jun 2019

The paper tackles the important issue of the impact of assumptions about aerosol layer height and spectral dependency of the aerosol refractive index on the quantification of aerosol SSA in the ultraviolet. With this aim in mind, the Authors compare the results of the "standard" KNMI retrieval scheme to those of a novel retrieval based on support vector machines (SVM), trained with real observations, on a particular scene of an aerosol smoke plume observed by TROPOMI. The comparison, which uses AERONET SSA as a benchmark, reveals that some assumptions made in the KNMI standard retrieval look problematic, and that the SVM based method is able to circumvent the problem and return more realistic values for the SSA.

[Figure]

[Figure]

**GENERAL COMMENTS**

While the scientific result of this paper is certainly interesting, I think there are a number of issues that need to be addressed before the paper can be published. First of all, I agree with the Editor's opinion that the manuscript does not read smoothly. The explanation of the SVM algorithm is difficult to follow, fails to mention important information (what's a support vector, what's a kernel) and makes it difficult for a reader to understand what is going on. In the description of the pre-processing it is not always easy to understand which quantity comes from which product (e.g., surface reflectance). The actual description of what was done to train the SVR for the retrieval of the AAOD is also confusing. Till Section 3.2.3 I was convinced that only a SVR is trained for the retrieval of AAOD, but at the end of Section 3.2.3 I get to know that there are two, and I don't fully understand why. In general, I think that the description of the entire process flow and of the logic behind it needs to be made more intelligible.

Finally, I have some concerns on validation. Testing the proposed method on a single scene basically means that the validation of the method is done against only one measurement. While the agreement between the SVM-based retrieval and AERONET looks excellent for the case shown, it would be important to see if this result is confirmed by looking at some more high aerosol loading events, which I guess should be possible to find, with ∼1.5 years of TROPOMI observations now available. Below are some point-by-point comments.

**SPECIFIC COMMENTS**

- Abstract, L16. Do you mean inappropriate assumptions on the spectral dependency of the SSA?

- L29. After Eq. 1 it would be useful to recap what are typical values of the UVAI for absorbing and non-absorbing aerosols.

- L37 and L46. Jeong and Su (2008) and Chimot et al. (2017) cannot be found in the

references.

- L72, "Another advantage". "Another" with respect to what?

- L81. Format reference correctly.

- L83. Yao et al. (2008) cannot be found in the references.

- L83, ". . . as it only depends on a subset of training data". WHAT exactly depends on a subset of training data? Also, here you mention the term "epsilon-insensitive loss" but don't say what it is, thus after this sentence the reader is really none the wiser about what you mean.

- L84. Again the same problem. You mention "kernel functions", but if you don't say what they are and what they have to do with SVMs, then this sentence is of no use at this point.

- L86. Mountrakis et al. (2011), Noia and Hasekamp (2018) cannot be found in the references.

- L86, "consist" -> "consisting"?

- L90, "expresses" -> "discusses"

- L99 and L110. What is the point of indicating the date of last access for a dataset that is only internally available?

- L109. Sanders and de Haan (2016) is not in the references.

- L125. Earlier you said that the TROPOMI product has a "scene albedo" A_sc. What is the difference between A_sc and A_s? Then later, at L168, you say that you filter your data for A_sc. Does this come from TROPOMI or from OMI then? I don't get it, I think all this is confusing.

- L142, Dubovik et al. (2000), Dubovik and King (2000) are not in the references.

- L165-166. While the reason for excluding large SZAs looks clear, why are the other

two criteria introduced? Please discuss.

- L181, "a strong spectral dependence . . . aerosols" -> "absorption by biomass burning aerosols in the near-UV has a strong spectral dependence".

- L199, "by the testing data" -> "on the testing data"

- Feature selection. It looks to me like you decided to train the SVR using only quantities that have a strong linear correlation to the SSA. In this way, though, you may be discarding some quantities that have some nonlinear relationship to the SSA which does not show up in the linear correlation coefficient. Please discuss.

- L209-L210. Please explain the reasons behind these filters for UVAI and ALH.

- L246-248, sentence "This is realized . . . predicted". You want to replace the OMI ALH with a value that is closer to the one that would have been retrieved by TROPOMI. But then why is OMI the target and TROPOMI the input? I was expecting it to be the other way around.

- L248-249, sentence "It should be noted . . . SVR". Please discuss why have you chosen to train this ALH-adjusting SVR on the Thomas fire and not on the dataset for the AAOD retrieval SVR.

- L260. I don't get what you mean by "We fit the SVR for AAOD prediction to both data sets".

- L262-264. I am lost here. Up to this point I was convinced that you trained two SVMs: one to adjust OMI ALH to the TROPOMI value and one to predict AAOD from UVAI, ALH and AOD, and that the goal of the ALH-adjusting SVM was to allow the use of OMI data to train the SVM for TROPOMI. Now I learn that there is a third SVM. It looks to me like this sentence contains new information, so it does not just "summarize the section". Please make sure that this is better explained in the paper, because it makes it really difficult to follow the discussion.

- L273, "the nonlinear transformation" -> "a nonlinear transformation"

- L275. Either shed some light on the connection between the concept of kernel and the training of SVMs, or avoid mentioning kernels at all.

- L275. You should make it clear that the Mercer theorem sets the conditions for a function to be admissible as a kernel in a SVM (basically, it says that the function should give rise to a positive-definite kernel matrix).

- L280. At line 276 you start the paragraph with "It is clear that", but actually point 3 is not clear at all from what you say. Nowhere before this line have you introduced the concept of support vector, nor have you explained what you mean by its "influencing area".

- L282. It would be better to move Section B of the supplement to an appendix in the main paper. Supplement should be used for additional figures and data, not for theoretical explanations.

- L282-283. Before saying that you are using radial basis function kernels, it may be useful to say that these are among the functions that satisfy Mercer's theorem. You can do this at the end of the previous paragraph (L276). Also, I would advise to write down the expression of the RBF kernel, so that the reader can better appreciate what is the parameter sigma that you mentioned earlier.

- L328. I get a bit confused by the distinction between the validation pixels and the rest of the plume. Are the validation pixels those in the small horizontal strip near the AERONET site in Fig. 9? You may want to indicate that in the paper.

- L352, "trained by the adjusted ALH" -> "trained using the adjusted ALH".

- L353, "to quantify" -> "of quantifying"

- L366, "representative" -> "well known"

- P10, References. The first reference looks incorrectly formatted.

---

## Referee Comment (RC2) · Omar Torres (Referee) · 12 Jul 2019

**Review of manuscript amt-2019-96: The role of aerosol layer height in quantifying aerosol absorption from ultraviolet satellite observations by Jiyunting Sun et al.**

**Summary**

This manuscript documents a statistics-based approach referred to as SVR (support vector regression) to retrieve single scattering albedo using MODIS retrieved aerosol optical depth (AOD) and TROPOMI UV Aerosol Index (UVAI) and aerosol layer height (ALH) from TROPOMI radiance measurements in the Oxygen-A band.

AERONET ground-based aerosol observations and the 13-year satellite OMI aerosol record (OMAERUV product) are used to build a training data set. The OMAERUV component of the training data set consists of a sub-set of ancillary parameters as well as UVAI and ALH values assumed in OMAERUV for the simultaneous retrieval of AOD and SSA. The resulting training data set includes only UVAI and ALH values associated with high accuracy OMAERUV AOD/SSA retrievals as measured by the difference between collocated AERONET and OMAERUV reported parameters (not larger than 0.03 for SSA and better than 5% for AOD).

Two versions of the trained SVR algorithm were used to retrieve the SSA of an aerosol plume over the Pacific Ocean off the coast of Southern California on December 12, 2017. Retrievals were also carried out using a conventional radiative-transfer-based algorithm, referred to as RTM by the authors. Comparison of the three satellite-based retrievals to AERONET Version 2 retrieved SSA at 500 nm (University of California Santa Barbara site) shows that the three space-based inversions agree with the only AERONET ground-based measurement available within AERONET's stated uncertainty (±0.03). On the other hand, the spread of the three satellite based SSA retrievals over the AERONET site is 0.01.

The authors examined the resulting SSA spatial variability over the extent of the plume, and conclude that the results of the SVR retrievals that show higher homogeneity are more convincing than the RTM approach that shows more spatial variability.

**Comments**

The authors have not demonstrated that the proposed SVR algorithm performs better than conventional RTM-based approaches. Deriving conclusions on the suitability of a retrieval method based on just one independent measurement is scientifically dubious. The author's accompanying argument that the lower spatial variability of the SVR approach makes the result more convincing is purely subjective and lacks the backing of a rigorous error analysis. It also ignores the radiative and dynamic interaction between the aerosol plume and the atmosphere that could generate SSA heterogeneity over a plume stretching over hundreds of kilometers.

The authors should carefully build an evaluation dataset using as many AERONET observations as possible, to judiciously examine the SVR algorithm performance. The interpretation of spatial variability is certainly not easy. Perhaps, CTM-generated data could also be used for this purpose.

I see a problem with the use of the AERONET as both training and evaluation tool. Unlike the AOD, AERONET SSA is not regarded a 'ground truth' measurement. The SSA is the result of an inversion procedure that yields non-unique solutions, and can produce different answers as the inversion algorithm evolves. For instance, for the case study in this paper the AERONET V2 500 nm SSA value used for evaluation of the satellite retrieval was 0.960. In the recently released AERONET V3 data, the reported SSA for the same event is now 0.982. If a SVR operational algorithm is in place, does the algorithm needs to be re-trained every time a new version of the AERONET data becomes available?

Based on the above consideration I do not think this work is publishable in its current form. Additional specific comments follow.

**Specific comments**

Line 29: Equation (1) is not consistent with equation given in Herman et al [1997]. What is the meaning of $\lambda$ and $\lambda_0$?

Line 31: Such a SSA global long-term record derived from the information content of the UVAI already exists [Torres et al., 2007]. It is produced by inverting OMI observations at 354 and 388 nm (same wavelengths used in the UVAI definition) to simultaneously retrieve aerosol optical depth (AOD) and single scattering albedo (SSA) at 388 nm. The AOD/SSA retrieval approach by the OMAERUV algorithm is fully documented [Torres et al., 2007; Torres et al., 2013] and SSA retrieval results have been systematically evaluated by comparison to the global AERONET SSA record [Torres et al., 2013; Jethva et al, 2014] and to SKYNET [Jethva and Torres, 2019] and MFRSR [Mok et al., 2016] SSA retrievals. The author's disregard of the 15-year near UV SSA record in the literature review is rather puzzling.

Line 38: The label 'RTM-based method' is not appropriate. All atmospheric retrievals methods are one way or another based on radiative transfer calculations. The authors are referring to SSA inversions in the UVAI space that infer SSA by 'matching' calculated to observed UVAI. The listed references on this approach are mostly academic exercises, none of which led to algorithm development. While the UVAI parameter contains information on aerosol properties, it is also affected by land surface effects, ocean color, sub-pixel size clouds, gas absorption, etc. Thus, the direct UVAI to SSA conversion techniques is not an optimal way to extract aerosol absorption from the near UV measurements. It is best to use actual radiances.

Line 47: Please mention the recently developed ALH retrieval capability from EPIC oxygen absorption bands to retrieve ALH of dust layers and carbonaceous aerosol layers over both ocean and land surfaces Xu et al., 2017, 2019]

Line 70: The discussion of SSA retrieval for this event should also include OMAERUV SSA results if available.

Line 111: An UVAI threshold of 1.0 also excludes low altitude absorbing aerosol layers, and low AOD elevated layers.

Line 114: Sensitivity of results of the assumed 50 hPA pressure thickness assumption should be discussed.

Line 130: The Herman et al [1997] assumption (spectrally flat As in the near UV) has been shown to be not generally valid. Is there a reason why the authors do not use the OMI-based Kleipool et al [2008, 2010] databases?

Line 131: In the description of the OMAERUV record, the authors list only the UVAI and ALH and omit the fact that both AOD and SSA are reported retrieved parameters. After all, the reason why the ALH is included in the OMAERUV product is that the inversion requires information on ALH. A more candid description of the OMAERUV product should read '….long-term UVAI, AOD and SSA with corresponding ALH….'

Line 142: Discuss the error bars and whiskers on Fig 2, particularly for the SSA. What are the implications of the expected diurnal variability?

Line 144: The time difference between TROPOMI and AERONET observations on December 12, 2017 is about 2.5 hours. Discuss the implication of that time difference in the context of the AERONET results in Fig 2.

Line 145: Both AERONET SSA and TROPOMI are results of inversions in which multiple solutions are possible. Thus, an inversion cannot be validated with another inversion. Use 'compare' instead of 'validate'. Use 'comparison' instead of 'validation' in all instances in the paper where the word 'validation' is used.

Line 146: Use AERONET version 3 data in the construction of the training data set. There are significant differences between version 2 and 3 of the AERONET inversion product.

Line 170. Provide the reasoning to conclude that the southern part of the plume is the most absorbing region. All it can be said, is that the largest UVAI is observed in that region, but AOD, ALH and spectral aerosol absorption exponent affect the magnitude of the resulting UVAI.

Line 187. Assuming constant refractive index for wavelengths longer than 388 nm is not a reasonable assumption. At longer wavelengths, the role of black carbon is more important. Discuss the implication of this assumption on the reported results.

Line 193: The 'existing' MODIS AOD and TROPOMI ALH retrievals involve assumptions on particle size distribution (PSD) and aerosol single scattering albedo. Are the assumed PSD's in the two algorithms consistent? How about the complex refractive indices assumed in the AOD and ALH retrieval? Please list the values of those parameters and discuss the implication of inconsistencies if any.

Line 206: Please describe in more detail the implied statistical analysis of 13 years of data involving the OMAERUV and AERONET data sets. What are the parameters being examined?

Line 208: The 13-year OMAERUV global dataset includes AOD and SSA, a record that the authors claim back on page 31, does not exist.

Line 212: I am totally lost here. The statement *'samples are excluded if the SSA difference between OMI and AERONET are larger than 0.03 or the AOD difference between OMI and AERONET is larger than 5%'* is incomprehensible. What OMI SSA/AOD are the authors talking about? Are these OMAERUV-retrieved values? Up to this point in the manuscript, the authors have not acknowledged the existence of such products. If these are indeed the OMAERUV SSA/AOD, then the authors have created a dataset consisting of the best quality OMAERUV AOD and SSA retrievals (as measured by the level of agreement with AERONET) to train the SVR algorithm. It is suggest that the description OMAERUV-SSA and OMAERUV-AOD be used (instead of the generic OMI-SSA or OMI-AOD) to avoid confusing the reader since there is a second OMI aerosol algorithm (OMAERO).

Line 228: Add Torres et al 2013 reference to the CALIOP ALH climatology.

Line 232: The UVAI height dependence was first documented [Herman at al., 1997; Torres et al., 1998] based on analysis of TOMS data.

Line 235: If spectrally dependent AOD (354 and 388 nm) and ALH are indeed independently know, one should be able to retrieve the SSA at 354 and 388 nm via a direct RTM inversion of the 354 and 388 nm radiances (not the UVAI). This is the simplest RTM approach that would fully characterize the aerosol plume.

Line 241: Fig 7(c) is not mentioned in the discussion. Remove it if not needed. Otherwise, explain, or eliminate, the difference between UVAI OMI and UVAI OMAERUV in the z-axis label of figures 7(b) and 7(c).

Line 245: As described the ALH adjustment sounds arbitrary. It looks to me the authors are just conveniently making up a convenient dataset. Please provide an understandable rationale for the creation of this ALH *dataset.*

Line 250 There is no mention in this work of the Oxygen-A band AOD that is simultaneously retrieved with ALH from TROPOPMI observations. Shouldn't it be better to use the TROPOMI AOD rather than the MODISD AOD? That would eliminate possible implicit inconsistencies in aerosol microphysics.

Line 259: What does 'rule of thumb ratio 70% versus 30%' really mean? This all sounds arbitrary.

Line 290: Figures should be described sequentially. From the description of figures 7(a) and (7b), the authors jump to Figure 5, and then back to Figure 7(c).

Line 301: The MODIS AOD uncertainty needs to be taking into account and propagated in a sensitivity analysis of the SVR application. Over the US west coast, in particular, the AOD is subject to large uncertainty due to surface effects [Jethva et al., 2019].

Line 323: The difference of 0.01 between the two SVR applications has not statistical meaning, as they are both within the stated AERONET uncertainty of ± 0.03 for a single measurement. Any over-interpretation is just splitting hairs.

Line 326: I disagree with this statement. No measurable improvement in performance is apparent from this comparison. The use of the adjusted ALH instead of the original OMAERUV ALH makes no statistically quantifiable difference whatsoever. A more systematic analysis using a large number of independently measured SSA values is required.

Line 328: In section 4.2, the authors try interpreting the lower spatial variability over the entire plume of the SVR retrieved SSA with respect to the SSA spatial variability resulting from the RTM-based approach, as an indication of better SVR accuracy. The north-south extent of the plume over the ocean is about 1000 km whereas the east-west dimension varies from about 200 to 700 km. For an aerosol plume this large, it is not unreasonable to expect spatial variability in SSA. The SSA of carbonaceous aerosols from biomass burning or wild fires is lowest near the source areas in the flaming phase of the fires. As the resulting smoke layer is transported downwind, it interacts with the surrounding air. Aerosol SSA may increase due to water uptake by hydrophilic particles. The resulting SSA homogeneity over the plume may therefore depend on the homogeneity of meteorological fields.

I do not think this study conclusively demonstrates that the described SVR technique yields more accurate retrieval than standard well thought out RTM approaches. Undoubtedly, however, the availability of TROPOMI ALH observations will improve the accuracy of retrieved aerosol absorption.

Line 345-347: The statement '*In cloud-free cases, it is expected that micro-physical properties of smoke particles within the plume should be similar over short time periods as they were originated from the same source and generated under the same conditions..*' is not always correct. The variability over a large smoke plume like the one used in this analysis may be important.

Line 364: The statement that the proposed method based on the correlation between between UVAI, AOD and ALH requires no a priori assumptions on aerosol micro-physics is incorrect. Implicit microphysics assumptions are involved in the use of MODIS AOD as well as TROPOMI ALH. The authors have ignored this fact, and treat the AOD and ALH as 'given true values', ignoring the fact that these parameters come out as the result of RTM-based inversions that assume particle size distribution, and complex refractive index over an extended spectral range. The results of a sensitivity analysis that propagates AOD and ALH retrieval uncertainties into the SVR method should be included in the paper.

Line 365: The statement '*a priori assumptions on aerosol microphysics is considered one of the major error sources in RTM-based method*' is misleading. I should read instead ' wrong *a priori assumptions* ..'

Line 368: '*Convincing..*' is not an objective characterization. I was not convinced as stated in this review.

 **Additional References**

Xu, X., Wang, J., Wang, Y., Zeng, J., Torres, O., Reid, J. S., Miller, S. D., Martins, J. V., and Remer, L. A.:
    Detecting layer height of smoke aerosols over vegetated land and water surfaces via oxygen

absorption bands: hourly results from EPIC/DSCOVR in deep space, Atmos. Meas. Tech., 12, 3269-3288, https://doi.org/10.5194/amt-12-3269-2019, 2019.

Xu, X., J. Wang, Y. Wang, J. Zeng, O. Torres, Y. Yang, A. Marshak, J. Reid, *and* S. Miller *(*2017*)*, Passive remote sensing of altitude and optical depth of dust plumes using the oxygen A and B bands: First results from EPIC/DSCOVR at Lagrange-1 point, Geophys. Res. Lett., 44, *doi:*10.1002/2017GL073939

Jethva, H., Torres, O., and Yoshida, Y.: Accuracy Assessment of MODIS Land Aerosol Optical Thickness Algorithms using AERONET Measurements, Atmos. Meas. Tech. Discuss., https://doi.org/10.5194/amt-2019-77, in review, 2019.

Jethva, H. and Torres, O.: A Comparative Evaluation of Aura-OMI and SKYNET Near-UV Single-scattering Albedo Products, Atmos. Meas. Tech. Discuss., https://doi.org/10.5194/amt-2019-174, in review, 2019.

---

## Author Comment (AC2) · 7 Aug 2019

Please read the .pdf file in the supplement. Thank you very much!

Please also note the supplement to this comment:
https://www.atmos-meas-tech-discuss.net/amt-2019-96/amt-2019-96-AC2-supplement.pdf

---

## Editor Comment (EC2) · Ben Veihelmann (Editor) · 27 Aug 2019

**GENERAL COMMENTS**

The manuscript (v3) has improved significantly. It is recommended to be published after the remaining issues listed below have been addressed.

Main comment

The choice of data used for training, testing, and evaluating the SVR algorithm needs to be explained more clearly and upfront e.g already in the introduction, or in a dedicated subsection in section 3 (similarly to Section 2.1.2 for Experiment I). Is the scheme

trained and tested on OMEARUV (augmented with CALIOP) and AERONET data? Is the scheme applied in case studies to UVAI and ALH data from S5P/Tropomi observations and AOD from MODIS?

Minor comments

Line 10-11: sentence incomplete

Line 18-19: the sentence "This empirical method is free from the uncertainties triggered by a priori assumptions ..." need to be reformulated. A) uncertainties are not "triggered by a priori assumptions". You might consider a formulation like "uncertainties due to imperfection a priori assumptions"; B) You can state that a priori assumptions do not appear explicitly in this empirical method, but it needs to be acknowledged that unknown variability in micro-physics contributes to the uncertainty of the results obtained using the empirical method.

Line 24: sentence incorrect "is better agrees"

Line 35/36: add that the index is sensitive to ELEVATED absorbing aerosol. This feature is important in the present study.

Line 75-76 the sentence "From our perspective, ML techniques can avoid making assumptions on poorly-understand aerosol micro-physics as that in the first experiment." ... " need to be reformulated. Proposed: "We employ ML techniques in order to avoid explicit assumptions on aerosol micro-physics as made in the first experiment."

Line 83-84: We will present the capability to retrieve SSA from UVAI of USING this empirical method with multiple case studies.

Line 85 replace "implemented" by e.g. "outlined"

Line 91 replace "implement" by e.g. "outline" or "report results from"

Line 197: sentence incorrect "from as an alternative"
Line 199: replace "understanding to" by e.g. "knowledge of"

Section 3.4 introduces SVR hyper-parameters tuning and explains theoretically the potential need for different values of the parameter 'p' to account for statistical differences between training and test data sets. The text remains vague about whether this is actually needed in the present case. Table 3 lists only one single value for p, which suggests is it no needed here. Please clarify, and streamline.

Line 421: remove "d" in "potential to retrieved SSA"

It is repeatedly mentioned that "The input features are selected by the Spearman's rank correlation coefficients", which raises more questions than it answers. The rationale should be explained more clearly. (Near Line 265 it is explained that AAOD is chosen as output parameter of the SVR scheme rather than SSA in view of the difference in the correlation coefficients. Is that what is meant with selection of input features?)

AMTD

---

## Author Comment (AC4) · 5 Sep 2019

Please check the .pdf file in the attachment. Thank you for your cooperation!

Please also note the supplement to this comment:
https://www.atmos-meas-tech-discuss.net/amt-2019-96/amt-2019-96-AC4-supplement.pdf

---

## Author Response (AR1)

This document is structured as follows:
- Response to Editor's comment (EC1)
- Response to Anonymous Referee #1 (RC1)
- Response to reviewer Omar Torres (RC2)
5 - Revised manuscript with markups

The editor's / reviewers' comment is in black, the author's response is in blue.

According to editor's and reviewers' comment, the structure of the manuscript has to be changed for better reading. As a guidance, we describe here the major changes:
10 - We used to describe SSA retrieval using radiative transfer simulations and SVR in parallel, which caused troubles in reading. Now in the revised manuscript, we separate the two methods thoroughly. Section 2 includes everything about the SSA retrieved by radiative transfer simulations, and Section 3 contains all information on SVR retrieval.
- There used to be 2 SVR models: one uses the OMAERUV-AERONET joint data set (UVAI, ALH from
15 OMAERUV and AOD, AAOD from AERONET) to train the SVR model, we call it as the SVR trained by the original training data set; another uses the same training data but with adjusted ALH to replace the ALH in OMAERUV, we called it the SVR trained by the adjusted training data set. The adjusted ALH is using an intermediate SVR trained by TROPOMI ALH. Thus, there used to be 3 SVR models in the previous version manuscript. The SSA retrieved by the adjusted training data set is slightly better than that retrieved from the
20 original training data set (OMAERUV-AERONET joint).

The original purpose to adjust the ALH is because the OMAERUV ALH is not retrieval but is guessed either from CALIOP climatology or a priori assumptions from AOD retrieval. We used to adjust it with TROPOMI ALH to make it more like observations. But the SSA retrieved by the SVR with the original training data set is acceptable, meanwhile the adjusted ALH causes many confusions. Thus, in the revised manuscript, we have
25 removed the process of adjusted ALH and the SVR trained by the adjusted training data set. There is only one SVR model in the revised manuscript, which is trained by the OMAERUV-AERONET joint data.

- We used to employ AEORNET version 2 inversion product to evaluate our SSA retrievals, and to construct the training data set for SVR method. According to Omar Torres's comment, we have replaced it with AERONET version 3 inversion product. The results and conclusions may change to some extent.
30 - We used to have only one case study in the manuscript as it was the only one available at that time. Now, we have searched through the recent half year since 2018 November and added cases as long as there are collocated TROPOMI UVAI and ALH, MODIS AOD and AERONET measurements available.
- We have included MERRA-2 aerosol reanalysis (Appendix C) as an independent reference to analyze the spatial variability of retrieved SSA in Section 3.6.3.

The structure of the revised manuscript is as follows:
Section 1 Introduction

Section 2 Experiment 1: SSA retrieval using radiative transfer simulations
40 Section 2.1 Radiative transfer simulation setup
Section 2.1.1 Aerosol models
Section 2.1.2 Inputs from satellite
Section 2.2 SSA retrieved by radiative transfer simulations

45 Section 3 Experiment 2: SSA retrieval using support vector regression
Section 3.1 Support vector regression
Section 3.2 Feature selection based on OMI and AERONET observations
Section 3.3 Preparing training and testing data sets
Section 3.4 SVR hyper-parameter tuning
50 Section 3.5 Error analysis
Section 3.6 Case applications
Section 3.6.1 California fire event on 12 December 2017
Section 3.6.2 Other case applications
Section 3.6.3 Spatial variability of retrieved SSA

Section 4 Conclusions
Appendix

**Response to Editor's comments**

60  OVERALL

The manuscript deals with a hot topic, i.e. the constraining the aerosol Single Scattering Albedo (SSA) using satellite observations. This quantity is difficult to capture by observations and at the same time very important regarding the radiative forcing of aerosol. The proposed scheme to infer SSA information is new and of interest to the AMT community. The manuscript is suitable for publication in AMT after the issues below are addressed.

65  GENERAL COMMENTS

The manuscript does not read smoothly. Formulations need to be improved in many instances. The specific issues listed below cover a number of these instances but the list is not exhaustive.

The study is dealing with a single case (a plume of one specific emission event). It needs to be discussed how robust are findings.

70  We have more case studies for the SVR algorithm to prove its capability of SSA retrieval (Section 3.6 in the revised manuscript).

The study approach needs to be explained upfront more clearly. The choices regarding the source of data (UVAI, ALH, AOD, SSA) used for training the SVM-based scheme, for evaluating the SVM-based algorithms, and for evaluating the RTM-based algorithms, needs to be clarified (at a high level upfront, in detail in the specific sections).

75  In the last version manuscript, the support vector regression (SVR) method was not well-demonstrated in the manuscript, as we planned to more focus on the implementation and results.

But in the revised manuscript, we have restructured the manuscript. We have separated the RTM part (Section 2) from SVR part (Section 3). The SVR section now includes: theory of SVR (Section 3.1), feature selection based on OMAERUV-AERONET joint data set (Section 3.2), the training and testing data set (Section 3.3), the hyper-
80  parameters tuning of SVR model (Section 3.4), the error analysis of SVR model (Section 3.5) and case applications (Section 3.6).

SPECIFIC COMMENTS

Line 12: It is not clear how SSA is retrieved, which algorithm is employed. Reference to "conventional radiative transfer simulations" is not sufficient.

85  It is not a specific algorithm. We fixed all other inputs in radiative transfer model except for the imaginary part of refractive index, then find the SSA by minimizing the difference between satellite retrieved UVAI and model simulated UVAI.

This sentence has been changed into: *In the first experiment, we retrieve SSA by minimizing the UVAI difference between observed ones and that simulated by a radiative transfer model. (line 11-13)*

90  Line 13: The approach to constraining the SSA retrieval is not clear. Is the ALH fixed in the forward model used in the SSA retrieval?

Yes, the ALH is taken from TROPOMI measurement, which is fixed. We used to try this method to retrieve SSA but the ALH is unknow for most cases, causing large uncertainties in SSA (Sun et al., 2018).

In the revised manuscript, we have rephrased the abstract: *With the recently released ALH product of S-5P TROPOMI*
95  *constraining forward simulations, a significant gap in the retrieved SSA (0.25) is found between radiative transfer simulations with spectral flat aerosols and strong spectral dependent aerosols, implying that inappropriate assumptions on aerosol absorption spectral dependence may cause severe misinterpretations of aerosol absorption. (line 13-16)*

Line 17: The sentence "In the second part of this paper, we propose. . ." is not clear. Clarify that the method relies on
100 an empirical relation that has been established based on long-term datasets of UVAI, ALH and AOD based on the SVR concept. The term "data-driven" is misleading.

This sentence has been changed into: *In the second part of this paper, we propose an alternative method to retrieve SSA based on long-term record of collocated satellite and ground-based measurements using the support vector regression (SVR). (line 16-18)*

105 Line 20 (also caption Figure 8): AERONET does not "measure" SSA directly but retrieves it. Reformulate.

We have reformulated the term through the manuscript.

Eq. 1 is unclear and the variables are not introduced. Without more information the reader cannot guess how to interpret the superscribed labels "obs" and "Ray". It is recommended to explain the UVAI concept and highlight that the obtained index is sensitive to elevated absorbing aerosol.

110 We have added descriptions for each symbol. (line 31-35)

Line 41: What is meant by "with various spectral choices"?

It means AOD product is available at many wavelengths. As we consider it has nothing to do with comparison of ALH data availability (i.e. ALH is not wavelength-dependent). This sentence has been changed into: *There are plentiful AOD products with wide spatial-temporal coverage. (line 46)*

115 Line 68: "quantitatively determine" –> quantify.

This sentence has been changed accordingly: *Now with the operational TROPOMI ALH constraining forward simulations, it is expected to partly reduce the SSA retrieval uncertainty meanwhile quantifying the influence of assumed aerosol properties on the retrieved SSA. (line 67-68)*

Line 72: data-driven –> empirical. Proposed to reformulate "We propose an empirical [. . .]. ML algorithms learn
120 [. . .]".

This sentence has been changed accordingly: *In the second experiment, we therefore propose an empirical method to predict aerosol absorption, based on the long-term records of collocated UVAI, ALH, AOD and absorbing aerosol optical depth (AAOD) using machine learning (ML) techniques. (line 70-72)*

Line 77: one piece of information is missing: does the training of an SVM requires less training data than ANN?

125 Yes, the SVM requires less training data than ANN method. We have added reference on this information: *Compared with other algorithms (e.g. the Artificial Neural Network), SVR is less sensitive to training data size and can successfully work with limited quantity of data (Mountrakis et al., 2011; Shin et al., 2005). (line 81-83)*

Line 77: ML and SVM seem to be used interchangeably, which not entirely correct: also ANN can be seen as ML tools.

130 This sentence is no more applicable. In the revised manuscript, we only use the term SVR in the parts related to SSA retrieval.

Line 79, 81: inconsistent use of singular and plural

We have uniformed the term into 'SVM' (support vector machines). SVR is a variant of SVM to solve regression problems. In the revised manuscript, we only use SVR.

135 Line 84: The term kernel functions is used as if it had been introduced already. Are these related to the support vectors?

Yes, the kernel functions are related to the SVR. The kernel function is an option for SVR method to solve either linear or nonlinear problems depending on whether the kernel function types. The parameters of a kernel function are determined during the training process. We have added introduction of SVR and its kernel in Section 3.1, the kernel hyper-parameter is determined in Section 3.4.

Line 107: reformulate "TROPOMI ALH retrieval is based on the pattern . . ."

This sentence has been changed into: *TROPOMI ALH is retrieved at oxygen A-band (759-770 nm), where the strong absorption of oxygen causes the highly structed spectrum. This feature is particularly suitable for elevated optically dense aerosol layers (Sanders et al., 2015; Sanders and de Haan, 2016) (line 133-135)*

Line 102 "For the forward radiative transfer calculations, the input aerosol profile is parameterized as . . ." is this choice consistent with the assumptions made in the ALH algorithm?

Yes, the same setting as ALH algorithm.

We have added reference on this information: *For the forward radiative transfer calculations, the input aerosol profile is parameterized according to the settings in ALH retrieval algorithm: a one-layered box shape profile, with central layer height derived from TROPOMI and an assumed constant pressure thickness of 50 hPa (Sanders and de Haan, 2016). (line 136-138)*

Line 129: The relevance of the reference to Herman & Celarier is not clear. Does the statement "A spectrally flat As is assumed . . ." apply to the OMI LER product? Or do you need to make this assumption?

In the previous version manuscript, I made this assumption as the wavelength dependence of the surface reflectivity between 340 and 380 nm is little (0.2%), but it is proved to be not generally true.

Thus in the revised version, I use the spectrally dependent surface albedo. Although in the radiative transfer calculation, due to the round-off, the results may not be significantly changed.

Section 3.1 falls short on an explicit and upfront specification of the source of the input data (such as AOD, ALH, UVAI) used for the RTM-based method. A discussion of temporal mis-registration between MODIS and TROPOMI data acquisitions is missing.

The content is now moved in Section 2.1.2. The UVAI and ALH come from the TROPOMI level 2 product and the AOD comes from the MODIS/AQUA level 2 collection 6. AQUA has a similar overpass time to that of S-5P (around 13:30 local time), which already has been included in the manuscript. The time difference in this case is only several minutes.

Line 156/157: it is not clear what are the implication of using surface reflectance data from the OMI LER for reproducing the UVAI using the RTM method. Please discuss. It is assumed that the surface reflectance generated within a UVAI product can be reproduced in a straight forward fashion if needed.

OMI LER surface reflectance is one of the inputs for radiative transfer calculation of UVAI. As currently surface albedo is not included in TROPOMI L2 UVAI product, we use OMI climatology instead (introduced in Section 2.1.2).

Line 161: The justification of the reporting wavelength of the retrieved SSA is not understood; in the end it is determined by the OMAERUV reporting wavelength?

The SSA retrieved by radiative transfer simulations can be reported at any wavelength you want only if you specify it in the configuration file used to run the radiative transfer model. We report SSA at 500 nm because the SSA retrieved by SVR is at this wavelength (both the OMAERUV SSA and AERONET SSA are available at 500 nm).

Line 173: Aerosol models cannot be a combination of a project and an algorithm. Rephrase.

This sentence has been changed into: *The aerosol models used for the Mie calculations are a combination of the aerosol models in ESA Aerosol_cci project (Holzer-Popp et al., 2013) and that in the OMAERUV algorithm (Torres et al., 2007; Torres et al., 2013). (line 106-107)*

Line 175: the phrase "The particle size distribution ..." needs grammatical/syntactic fixing

This sentence has been changed into: *We use the particle size distribution of the fine mode strongly absorbing aerosol of ESA Aerosol_cci project. The geometric radius ($r_g$) is 0.07 μm (effective radius $r_{eff}$ of 0.14 μm) and the geometric standard deviation ($\sigma_g$) is 1.7 (logarithm variance $\ln\sigma_g$ of 0.53). (line 108-111)*

Line 179: subtype "BIO-1" is referred to without explanation/reference Line 177: real PART OF THE refractive index

This sentence has been changed into: *The real part of the refractive index (n) uses the same value as in the OMAERUV algorithm, which is set to be 1.5 for all subtypes and spectrally flat. We adopt the imaginary part of the refractive index at 388 nm ($\kappa_{388}$) of the OMAERUV smoke subtypes (except for BIO-1 whose $\kappa_{388}$ is 0) in our study and add a subtype with $\kappa_{388}$ equaling to 0.06. (line 111-114)*

Line 178: imaginary PART OF THE refractive index

See previous response.

Line 181: Sentence incomplete

This sentence has been changed into: *Many studies have shown evidence that absorption by biomass burning aerosols in the near-UV band has a strong spectral dependence (Kirchstetter et al., 2004; Bergstrom et al., 2007; Russell et al., 2010). (line 115-116)*

Line 182 (also caption Figure 8): The specification of Δκ in not clear. Clarify.

We actually explain the Δκ in the latter part of this sentence: Δκ is defined as the relative difference between $\kappa_{354}$ and $\kappa_{388}$. We have added a formula in the revised manuscript: $\Delta\kappa = (\kappa_{354} - \kappa_{388})/\kappa_{388}$ *(line 118, Figure.4 caption and Table 1 title)*

Table 1: The specification of the imaginary part of the refractive indices Δκ is unclear. For which reference wavelength are the numbers in the rightmost column valid? In the column "Refractive index imaginary part at 354 nm (k354) one expects an explicit list of values rather than a formula. Clarify.

The reference wavelength is 388 nm. Δκ is the relative difference between $\kappa_{354}$ and $\kappa_{388}$ ($(\kappa_{354} - \kappa_{388})/\kappa_{388}$). Also see previous response.

Here we select 9 different Δκ values from 0% to 40% and 7 different $\kappa_{388}$ values from 0.005 to 0.060, thus overall 63 values of $\kappa_{354}$. It is trivial to list all values of the refractive index imaginary part at 354 nm.

Line 186: Absorbing Ångström Exponent –> Ångström exponent

It has been changed accordingly through the manuscript. (line 121)

Line 207: "13-year measurement OMAERUV and AERONET measurements" rectify formulation.

This sentence has been moved to Section 3.2: *To start with, we collect the measurements of OMAERUV version 3 product (http://dx.doi.org/10.5067/Aura/OMI/DATA2004 last access: 17 October 2018) and AERONET version 3 level 1.5 inversion product (https://aeronet.gsfc.nasa.gov, last access: 4 June 2019) from 2005-01-01 to 2017-12-31. (line 250-252)*

Line 210: an OMI pixel is collocated –> OMI observations are considered as collocated

It has been changed accordingly: *Then OMI observations are considered as collocated with an AERONET site if their spatial distance is within 50 km and their temporal difference is within 3 hours. (line 253-255)*

Section 3.2.2 (Preparing training and testing data sets) is quite confusing. Please rewrite. Some terminology is used inconsistently (e.g. the terms "extra SVR", "adjusted ALH", "predicted ALH", or "ALH from OMAERUV" and "ALH from OMI"). Maybe introduce Table 2 and Flow charts (Figure 5) already at the beginning of the section.

220   As declared at the beginning of this document, the structure of the manuscript is changed. The original preparing training and testing sets is moved to Section 3.2 and Section 3.3. The flow chart is still Figure.5.

Line 240, 259, 326: It is referred to "ALH from the OMAERUV" suggesting that ALH is generated by the OMAERUV algorithm. It is stated in the manuscript that these ALH values are actually taken from CALIOP. Please refer to "ALH from CALIOP" for clarity.

225   The ALH in OMAERUV product is not entirely from CALIOP climatology. Actually, we have explained that ALH in the OMAERUV product is a combination of CALIOP climatology and assumed ALH in the AOD retrieval (if the CALIOP climatology is not available): *The best-guessed ALH in the OMAERUV is either from CALIOP climatology or assumed ALH in the retrieval (if the CALIOP climatology is not available) (Torres et al., 2013). (line 246-247)*

The ALH from the OMAERUV just indicate the ALH provided in the OMAERUV. If use ALH from CALIOP, it
230   seems that we take ALH directly from the CALIOP product, which is not the case.

Line 240: It is stated that the ALH from the OMAERUV product (actually from CALIOP) may not have sufficient quality. Clarify what is the concern. Co-location?

The co-location is one of the concerns, considering the OMI measurements are significantly affected by the row anomaly and the limited swath of CALIOP. Moreover, as described in the precious response, the ALH provided in
235   OMAERUV product is not a real retrieval from measurements, but a best-guess based on CALIOP climatology and a priori assumptions.

We have added comment on the OMAERUV ALH: *As a result, one should keep in mind that the ALH from OMAERUV may suffer from the uncertainties of CALIOP climatology and a priori assumptions, and collocation error between OMI pixels and CALIOP footprint. (line 247-249)*

240   Line 242: What is meant with the OMI ALH? Is it the same as the ALH from the OMAERUV (CALIOP)?

It refers to the ALH from OMAERUV product. In the revised manuscript, we use the term OMAERUV ALH to make it clear.

Line 245: Is the TROPOMI ALH the one retrieved from the O2-A band?

Yes, TROPOMI ALH product is retrieved from the Oxygen A-band.

245   Line 249: It is stated that the "extra" SVR is trained on the Thomas fire case. A training sets should cover more than one case. Please discuss the validity of the approach.

As declared at the beginning of this document, the 'extra' SVR in the previous version manuscript is removed. We only use the OMAERUV-AERONET joint data source to train the SVR model and retrieve the SSA.

Line 253: it is noted that this "extra" SVR is a temporary intermediate step to obtain a better ALH". Please explain
250   upfront the approach.

This part is no longer applicable as we have removed it from the revised manuscript. See the previous response and the description at the beginning of this document.

Line 255: It is stated that "there is no necessity to do this anymore once a reliable ALH product is accessible to build up training data sets, e.g. the TROPOMI ALH product that will be released in the near future". Clarify in which sense
255   the training set using TROPOMI ALH is expected to outperform the training set using CALIOP ALH.

This part is no longer applicable as we have removed it from the revised manuscript. See the previous response and the description at the beginning of this document.

Line 259: What is meant with "The rule of thumb ratio is 70% versus 30%"? Eq. 3: Introduce the variable n.

260 This introduction on training/testing data set separation is now in Section 3.3. The empirical ratio to divide a data set into a training set (to training the SVR model) and a test set (to evaluate the generalization performance of the trained SVR model) is 70% to 30%.

n in the Eq.(3) is the number of samples. We have added this explanation in line 219.

Eq. 3: What is the dimensionality of omega? What is meant with ‖omega‖? Some kind of norm?

265 This content has been moved to Section 3.1, where we have added a new section briefly explaining the theory of SVR. ‖…‖ denotes the norm. (line 229)

Eq. 4: Introduce the variable x.

See Section 3.1. We have added this explanation in line 229.

Eq. 4: Why introduce the kernel function K? What is done with it?
270 Section 3.2.4 Data for case application: Please report the number of validation samples

The kernel function is aimed to solve either linear or nonlinear problems, depending on the kernel function types. This is introduced in Section 3.1 (line 232-235).

The number of samples in case applications are listed in Table 2 (California fire on 2017-12-12) and Table 4 (other case applications).

275 Figure 5: The figure shows at the same time SVR based ALH prediction and SVR based AAOD prediction, this is confusing. It would help to depict the two schemes for AAOD prediction in one flow chart, and the ALH prediction in a separate one.

As described at the beginning of this document, there is only one SVR model left, and Figure.5 has been changed accordingly.

280 Figure 6 (also Line 206): Why is the sign of the correlation coefficient not reported? Report the sign or justify and clarify in caption that |rho| is reported.

The priority is given to the magnitude of correlation rather than the sign, it is not a problem to show sign though. Besides, we use the Spearman's rank correlation coefficient instead of Pearson's correlation coefficient in order to deal with the non-linearity among different parameters.

285 Figure 6: For which ALH parameter are the correlations reported? For the predicted one or for the one from CALIOP?

The ALH in the training data set, i.e. the ALH reported in the OMAERUV product. The description on parameters in Figure.6 is in Section 3.2 line 261-263: *The parameters in OMAERUV-AERONET joint data set for feature selection consists of UVAI calculated by 354 and 388 nm wavelength pair, satellite geometries, surface conditions and ALH from OMAERUV, and SSA, AOD and AAOD from AERONET.*

290 Figure 7: The 3D plot is hard to interpret. Recommended to replace it with 2D scatter-plots (AOD versus ALH) where the UVAI is only color-coded.

As described at the beginning of this document, there is only one SVR model left, and this figure is no longer necessary to show. We have removed it from the revised manuscript.

295 **Response to anonymous referee #1's comments**

The paper tackles the important issue of the impact of assumptions about aerosol layer height and spectral dependency of the aerosol refractive index on the quantification of aerosol SSA in the ultraviolet. With this aim in mind, the Authors compare the results of the "standard" KNMI retrieval scheme to those of a novel retrieval based on support vector machines (SVM), trained with real observations, on a particular scene of an aerosol smoke plume observed by TROPOMI. The comparison, which uses AERONET SSA as a benchmark, reveals that some assumptions made in the KNMI standard retrieval look problematic, and that the SVM based method is able to circumvent the problem and return more realistic values for the SSA.

GENERAL COMMENTS

While the scientific result of this paper is certainly interesting, I think there are a number of issues that need to be addressed before the paper can be published. First of all, I agree with the Editor's opinion that the manuscript does not read smoothly. The explanation of the SVM algorithm is difficult to follow, fails to mention important information (what's a support vector, what's a kernel) and makes it difficult for a reader to understand what is going on. In the description of the pre-processing it is not always easy to understand which quantity comes from which product (e.g., surface reflectance). The actual description of what was done to train the SVR for the retrieval of the AAOD is also confusing. Till Section 3.2.3 I was convinced that only a SVR is trained for the retrieval of AAOD, but at the end of Section 3.2.3 I get to know that there are two, and I don't fully understand why. In general, I think that the description of the entire process flow and of the logic behind it needs to be made more intelligible.

Finally, I have some concerns on validation. Testing the proposed method on a single scene basically means that the validation of the method is done against only one measurement. While the agreement between the SVM-based retrieval and AERONET looks excellent for the case shown, it would be important to see if this result is con- firmed by looking at some more high aerosol loading events, which I guess should be possible to find, with ~1.5 years of TROPOMI observations now available. Below are some point-by-point comments.

As declared at the beginning of this document, we have restructured the manuscript, with a separated section on SVR (Section 3), and only keep one SVR model to avoid misunderstanding. For more information on the structure modifications and other changes, please see the overview at the beginning of this document.

There was only one case available when we were preparing this manuscript. In the revised manuscript, we have added other fire events happened recently, as long as there are collocated TROPOMI, MODIS and AERONET measurements available.

SPECIFIC COMMENTS

- Abstract, L16. Do you mean inappropriate assumptions on the spectral dependency of the SSA?

Actually, it is the inappropriate assumption on the spectral dependency of the imaginary part of the refractive index causes the disagreement between retrieved SSA and AERONET SSA.

The sentence has been changed into: *With the recently released ALH product of S-5P TROPOMI constraining forward simulations, a significant gap in the retrieved SSA (0.25) is found between radiative transfer simulations with spectral flat aerosols and strong spectral dependent aerosols, implying that inappropriate assumptions on aerosol absorption spectral dependence may cause severe misinterpretations of aerosol absorption. (line 13-16)*

- L29. After Eq. 1 it would be useful to recap what are typical values of the UVAI for absorbing and non-absorbing aerosols.

We have added the explanation: *Positive UVAI indicates the presence of absorbing aerosols, while the negative or near zero values imply non-absorbing aerosols or clouds (Herman et al., 1997). (line 35-37)*

– L37 and L46. Jeong and Su (2008) and Chimot et al. (2017) cannot be found in the references.

We have added the references accordingly.

- L72, "Another advantage". "Another" with respect to what?

Sorry for the misunderstanding. The sentence has been changed into: *From our perspective, ML techniques can avoid*
340 *making assumptions on poorly-understand aerosol micro-physics as that in the first experiment. (line 75-76)*

- L81. Format reference correctly.

The reference format has been changed accordingly.

- L83. Yao et al. (2008) cannot be found in the references.

345 We have added it in the reference.

- L83, ". . . as it only depends on a subset of training data". WHAT exactly depends on a subset of training data? Also, here you mention the term "epsilon-insensitive loss" but don't say what it is, thus after this sentence the reader is really none the wiser about what you mean.

350 SVR attends to find an optimal hyperplane that maximizes the margin of tolerance (i.e.$\varepsilon$) in order to minimize the error. The error within the margin does not contribute to the total loss function. Thus, we say SVR only depends on a subset of training data and its loss function is $\varepsilon$-insensitive.

More introduction on SVR is in the newly added Section 3.1.

- L84. Again the same problem. You mention "kernel functions", but if you don't say what they are and what they
355 have to do with SVMs, then this sentence is of no use at this point.

The kernel function is a property of SVR to solve linear or non-linear problems, depending on the kernel functions.

More introduction on SVR kernel is in the newly added Section 3.1.

- L86. Mountrakis et al. (2011), Noia and Hasekamp (2018) cannot be found in the references.

360 We have added them in the reference.

- L86, "consist" -> "consisting"?

We have changed it accordingly.

365 - L90, "expresses" -> "discusses"

We have changed it accordingly.

- L99 and L110. What is the point of indicating the date of last access for a dataset that is only internally available?

It is just on the command of the journal.

- L109. Sanders and de Haan (2016) is not in the references.

370 We have added it in the reference.

- L125. Earlier you said that the TROPOMI product has a "scene albedo" A_sc. What is the difference between A_sc and A_s? Then later, at L168, you say that you filter your data for A_sc. Does this come from TROPOMI or from OMI then? I don't get it, I think all this is confusing.

The scene albedo (A_sc) is the total albedo of the scene (contributed by clouds, aerosols, surface, etc.)  while the
surface albedo (A_s) is only the albedo of surface. A_sc comes with TROPOMI L2 UVAI product, while the A_s is
not provided in this product. Instead we use A_s from OMI climatology. For the radiative transfer simulation of UVAI,
A_s is required rather than A_sc.

We used to A_sc to filter our data in order to reduce impacts of clouds. Now in the revised manuscript, we use the
TROPOMI FRESCO cloud support product to filter the clouds (Section 2.1.2). The pre-processing criteria has been
changed into: *$\theta_0$ larger than 75°, $UVAI_{354,388}$ smaller than 1, $AOD_{550}$ smaller than 0.5 or CF larger than 0.3. (line 151-152)*

- L142, Dubovik et al. (2000), Dubovik and King (2000) are not in the references.

We have added them in the reference.

- L165-166. While the reason for excluding large SZAs looks clear, why are the other two criteria introduced? Please
discuss.

The other two criteria are to exclude effects due to non-absorbing compositions and lower measurement confidence
(smaller aerosol signal).

The criteria in the revised manuscript also includes the FRESCO cloud fraction <= 0.3 to reduce effects from clouds:
*Before implementing radiative transfer calculations, pre-processing excludes pixels meeting at least one of the
following criteria: $\theta_0$ larger than 75°, $UVAI_{354,388}$ smaller than 1, $AOD_{550}$ smaller than 0.5 or CF larger than 0.3. (line
150-152)*

- L181, "a strong spectral dependence . . . aerosols" -> "absorption by biomass burning aerosols in the near-UV has a
strong spectral dependence".

The sentence has been changed into: *Many studies have shown evidence that absorption by biomass burning aerosols
in the near-UV band has a strong spectral dependence (Kirchstetter et al., 2004; Bergstrom et al., 2007; Russell et al.,
2010). (line 115-116)*

- L199, "by the testing data" -> "on the testing data"

 The sentence has been changed accordingly.

- Feature selection. It looks to me like you decided to train the SVR using only quantities that have a strong linear
correlation to the SSA. In this way, though, you may be discarding some quantities that have some nonlinear
relationship to the SSA which does not show up in the linear correlation coefficient. Please discuss.

We have replaced the Pearson correlation coefficient with the Spearman's rank correlation coefficient in the revised
manuscript. The Pearson correlation assesses linear relationships, while the Spearman correlation assesses monotonic
relationships (whether linear or not). The feature selection is re-written in Section 3.2.

- L209-L210. Please explain the reasons behind these filters for UVAI and ALH.

We used to exclude samples with UVAI < 0.8 and pixels with extreme high ALH but low UVAI, in order to exclude
situations where strong absorbing aerosols layering at low altitude (because the California fire 2017-12-12 is elevated
plume). But in other cases (added in the revised manuscript), where aerosol layering are more close to the surface. As a
result, in the revised manuscript, no constraint on UVAI and ALH applied.

The criteria has been slightly changed in the revised manuscript, where only SZA and clouds are considered: *OMI
pixels with $\theta_0$ larger than 75° or cloud fraction larger than 0.3 are excluded.(line 252-253)*

- L246-248, sentence "This is realized . . . predicted". You want to replace the OMI ALH with a value that is closer to
the one that would have been retrieved by TROPOMI. But then why is OMI the target and TROPOMI the input? I was
expecting it to be the other way around.

415    This step is no longer applicable as we have deleted this part to avoid confusions. There is only one SVR model in the revised manuscript, which is the SVR trained by the OMAERUV-AERONET joint data set, with UVAI and ALH from OMAERUV, and AOD and AAOD from AERONET. More description can refer to the overview at the beginning of this document.

- L248-249, sentence "It should be noted . . . SVR". Please discuss why have you chosen to train this ALH-adjusting
420    SVR on the Thomas fire and not on the dataset for the AAOD retrieval SVR.

Similar to the previous response, this step is no longer applicable as we have deleted this part to avoid confusions. More description can refer to the overview at the beginning of this document.

- L260. I don't get what you mean by "We fit the SVR for AAOD prediction to both data sets".

Similar to the previous response, this step is no longer applicable as we have deleted this part to avoid confusions.
425    More description can refer to the overview at the beginning of this document.

- L262-264. I am lost here. Up to this point I was convinced that you trained two SVMs: one to adjust OMI ALH to the TROPOMI value and one to predict AAOD from UVAI, ALH and AOD, and that the goal of the ALH-adjusting SVM was to allow the use of OMI data to train the SVM for TROPOMI. Now I learn that there is a third SVM. It looks to me like this sentence contains new information, so it does not just "summarize the section". Please make sure that this
430    is better explained in the paper, because it makes it really difficult to follow the discussion.

As described at the beginning of this document, there is only one SVR model in the revised manuscript. The step that adjust ALH in OMAERUV to the TROPOMI value is no longer applicable. Although in the previous version manuscript, the adjusted ALH leads to slightly better SSA retrieval, but the retrieval from SVR trained by the original OMAERUV ALH is acceptable enough.

435    The re-written SVR content is in Section 3, and the procedure of SVR is summarized in flow chart Figure.5.

- L273, "the nonlinear transformation" -> "a nonlinear transformation"

The sentence has been changed accordingly.

- L275. Either shed some light on the connection between the concept of kernel and the training of SVMs, or avoid mentioning kernels at all.

440    The kernel function is described in Section 3.1 in the revised manuscript. The kernel function is a property to solve either linear or non-linear problems, depending on the function types.

- L275. You should make it clear that the Mercer theorem sets the conditions for a function to be admissible as a kernel in positive semi-definite a SVM (basically, it says that the function should give rise to a positive-definite kernel matrix).

445    The sentence has been changed into: *where $K(x_i, x)$ is the kernel function that is positive semi-definite in order to satisfy Mercer's theorem. (line 234)*

- L280. At line 276 you start the paragraph with "It is clear that", but actually point 3 is not clear at all from what you say. Nowhere before this line have you introduced the concept of support vector, nor have you explained what you mean by its "influencing area".

450    The introduction on SVR and its relevant concepts, 'influencing area', 'support vector', etc. are in Section 3.1 in the revised manuscript.

- L282. It would be better to move Section B of the supplement to an appendix in the main paper. Supplement should be used for additional figures and data, not for theoretical explanations.

We have moved this content as part of manuscript in Section 3.4 SVR hyper-parameter tuning.

455 - L282-283. Before saying that you are using radial basis function kernels, it may be useful to say that these are among the functions that satisfy Mercer's theorem. You can do this at the end of the previous paragraph (L276). Also, I would advise to write down the expression of the RBF kernel, so that the reader can better appreciate what is the parameter sigma that you mentioned earlier.

The expression of RBF kernel is in Section 3.4 Equation (11).

460 - L328. I get a bit confused by the distinction between the validation pixels and the rest of the plume. Are the validation pixels those in the small horizontal strip near the AERONET site in Fig. 9? You may want to indicate that in the paper.

Yes, the mean values of these pixels are used to compare with AERONET. We have replaced the validation pixels with AERONET-collocated pixels. The collocation is the distance within 50 km and the time difference within 3
465 hours: *Then OMI observations are considered as collocated with an AERONET site if their spatial distance is within 50 km and their temporal difference is within 3 hours. (line 253-255)*

- L352, "trained by the adjusted ALH" -> "trained using the adjusted ALH".

The sentence is no longer applicable.

- L353, "to quantify" -> "of quantifying"

470 The sentence has been changed accordingly.

- L366, "representative" -> "well known"

The sentence has been changed accordingly.

475 - P10, References. The first reference looks incorrectly formatted.

The reference format has been changed.

**Response to reviewer Omar Torres's comments**

480 **Summary**

This manuscript documents a statistics-based approach referred to as SVR (support vector regression) to retrieve single scattering albedo using MODIS retrieved aerosol optical depth (AOD) and TROPOMI UV Aerosol Index (UVAI) and aerosol layer height (ALH) from TROPOMI radiance measurements in the Oxygen-A band.

AERONET ground-based aerosol observations and the 13-year satellite OMI aerosol record (OMAERUV product) are
485 used to build a training data set. The OMAERUV component of the training data set consists of a sub-set of ancillary parameters as well as UVAI and ALH values assumed in OMAERUV for the simultaneous retrieval of AOD and SSA. The resulting training data set includes only UVAI and ALH values associated with high accuracy OMAERUV AOD/SSA retrievals as measured by the difference between collocated AERONET and OMAERUV reported parameters (not larger than 0.03 for SSA and better than 5% for AOD).

490 Two versions of the trained SVR algorithm were used to retrieve the SSA of an aerosol plume over the Pacific Ocean off the coast of Southern California on December 12, 2017. Retrievals were also carried out using a conventional radiative-transfer-based algorithm, referred to as RTM by the authors. Comparison of the three satellite-based retrievals to AERONET Version 2 retrieved SSA at 500 nm (University of California Santa Barbara site) shows that the three space-based inversions agree with the only AERONET ground-based measurement available within
495 AERONET's stated uncertainty (±0.03). On the other hand, the spread of the three satellite based SSA retrievals over the AERONET site is 0.01.

The authors examined the resulting SSA spatial variability over the extent of the plume, and conclude that the results of the SVR retrievals that show higher homogeneity are more convincing than the RTM approach that shows more spatial variability.

500 **Comments**

The authors have not demonstrated that the proposed SVR algorithm performs better than conventional RTM-based approaches. Deriving conclusions on the suitability of a retrieval method based on just one independent measurement is scientifically dubious. The author's accompanying argument that the lower spatial variability of the SVR approach makes the result more convincing is purely subjective and lacks the backing of a rigorous error analysis. It also ignores
505 the radiative and dynamic interaction between the aerosol plume and the atmosphere that could generate SSA heterogeneity over a plume stretching over hundreds of kilometers.

The authors should carefully build an evaluation dataset using as many AERONET observations as possible, to judiciously examine the SVR algorithm performance. The interpretation of spatial variability is certainly not easy. Perhaps, CTM-generated data could also be used for this purpose.

510 I see a problem with the use of the AERONET as both training and evaluation tool. Unlike the AOD, AERONET SSA is not regarded a 'ground truth' measurement. The SSA is the result of an inversion procedure that yields non-unique solutions, and can produce different answers as the inversion algorithm evolves. For instance, for the case study in this paper the AERONET V2 500 nm SSA value used for evaluation of the satellite retrieval was 0.960. In the recently released AERONET V3 data, the reported SSA for the same event is now 0.982. If a SVR operational algorithm is in
515 place, does the algorithm needs to be re-trained every time a new version of the AERONET data becomes available?

Based on the above consideration I do not think this work is publishable in its current form. Additional specific comments follow.

We have added more case applications to present the capability using SVR algorithm to retrieve SSA in Section 3.6 of the revised manuscript, as long as there are collocated TROPOMI, MODIS and AERONET measurement available. In
520 the end there are 9 collocated samples from total 5 cases.

We also have rephrased the discussion on comparison between SSA retrieved from radiative transfer simulations (Section 2.2) and that retrieved from SVR algorithms (Section 3.6) in order to make it more objective.

The CTM data might be an option. We have provided MERRA-2 aerosol reanalysis data (M2T1NXAER) and derived SSA by dividing total scattering AOD with total extinction AOD (averaged between 12:00-15:00 local time) in
525 Appendix C. We also provide the range of retrieved SSA for each case (maximum SSA – minimum SSA) in Table 4. According to MERRA-2, although the plume pattern differences between satellite observations and model simulations exist, the MERRA-2 SSA heterogeneity of plume is at level around 0.1. This value is significantly smaller than the spatial variability of SSA retrieved in Experiment 1, while it is closer to the SSA variability retrieved from SVR.

Your concern on the use of AERONET is true. The supervised machine leaning algorithms have to be re-trained every
530 time if the training data is changed, and the predicted results may also change accordingly. In our case, if the AERONET product is updated, then the SVR algorithm has to be re-trained. It is therefore important to ensure the training data set is of high quality. As your said, the SSA from AERONET inversion product is not 'ground truth', but in most cases, the SSA provided by other sources is compared with that from AERONET. AERONET SSA plays a role as a reference, even though it is not the true value.

535 **Specific comments**

Line 29: Equation (1) is not consistent with equation given in Herman et al [1997]. What is the meaning of $\lambda$ and $\lambda_0$?

It is actually the same equation as the Eq(2) in Herman et al (1997) only with notations changed. We have added description: *where $I_\lambda$ and $I_{\lambda_0}$ are the radiance at wavelength $\lambda$ and $\lambda_0$ (reference wavelength). (line 34)*

Line 31: Such a SSA global long-term record derived from the information content of the UVAI already exists [Torres
540 et al., 2007]. It is produced by inverting OMI observations at 354 and 388 nm (same wavelengths used in the UVAI definition) to simultaneously retrieve aerosol optical depth (AOD) and single scattering albedo (SSA) at 388 nm. The AOD/SSA retrieval approach by the OMAERUV algorithm is fully documented [Torres et al., 2007; Torres et al.,

2013] and SSA retrieval results have been systematically evaluated by comparison to the global AERONET SSA record [Torres et al., 2013; Jethva et al, 2014] and to SKYNET [Jethva and Torres, 2019] and MFRSR [Mok et al.,
545 2016] SSA retrievals. The author's disregard of the 15-year near UV SSA record in the literature review is rather puzzling.

Sorry for the confusion, but we are not saying that there is no method to retrieve SSA from UVAI or there is no long-term record of SSA that derived from UVAI. The message we want to deliver is that the SSA does not have abundant amount of data as UVAI. UVAI data has global coverage for over 4 decades, while SSA data availability is much less
550 than that.

The sentence has been changed into: *It would be beneficial to derive aerosol absorption properties from the long-term global UVAI records, e.g. the single scattering albedo (SSA), which is the ratio of aerosol scattering to aerosol extinction. (line 37-39)*

Line 38: The label 'RTM-based method' is not appropriate. All atmospheric retrievals methods are one way or another
555 based on radiative transfer calculations. The authors are referring to SSA inversions in the UVAI space that infer SSA by 'matching' calculated to observed UVAI. The listed references on this approach are mostly academic exercises, none of which led to algorithm development. While the UVAI parameter contains information on aerosol properties, it is also affected by land surface effects, ocean color, sub-pixel size clouds, gas absorption, etc. Thus, the direct UVAI to SSA conversion techniques is not an optimal way to extract aerosol absorption from the near UV measurements. It
560 is best to use actual radiances.

Truly, all the atmospheric methods are based on radiative transfer calculations. We used to name 'RTM-based method' is to compare with 'SVR-based method', which is only applicable in this paper. In the revised manuscript, we call it as SSA retrieved from radiative transfer simulations.

We admit that fitting the radiances should be better than fitting the UVAI. Nevertheless, the radiance itself is also
565 affected by surface conditions, clouds, atmospheric gases and aerosols. On the other hand, the UVAI only contains the information of aerosol absorption.

Line 47: Please mention the recently developed ALH retrieval capability from EPIC oxygen absorption bands to retrieve ALH of dust layers and carbonaceous aerosol layers over both ocean and land surfaces Xu et al., 2017, 2019]

We have added the work of Xu et al. (2017, 2019). (line 52-53)

570 Line 70: The discussion of SSA retrieval for this event should also include OMAERUV SSA results if available.

We have added the information of OMAERUV SSA if available, which is in Table 2 and Table 4.

The OMAERUV pixels are applied the same collocation method as that of TROPOMI (distance within 50 km and time difference within 3 hours). For the California fire on 2017-12-12, the OMAERUV SSA is $0.92\pm0.01$, which is 0.06 lower than that of AERONET. Considering all case studies, there are total 6 OMAERUV samples collocated with
575 AERONET, with half of them within difference of 0.03 (Figure 15).

Line 111: An UVAI threshold of 1.0 also excludes low altitude absorbing aerosol layers, and low AOD elevated layers.

Yes, indeed. We only focus on the aloft (strong) absorbing aerosol layers in case studies. The low AOD may contains low aerosol signal.

580 Line 114: Sensitivity of results of the assumed 50 hPA pressure thickness assumption should be discussed.

The depth of 50 hPa is used in the TROPOMI ALH retrieval algorithm. We have added explanation in the manuscript: *For the forward radiative transfer calculations, the input aerosol profile is parameterized according to the settings in ALH retrieval algorithm: a one-layered box shape profile, with central layer height derived from TROPOMI and an assumed constant pressure thickness of 50 hPa (Sanders and de Haan, 2016). (line 136-138)*

585 Besides, the effect of aerosol layer depth on retrieved UVAI is minor, please see the sensitivity study in Sun et al., 2018.

Line 130: The Herman et al [1997] assumption (spectrally flat As in the near UV) has been shown to be not generally valid. Is there a reason why the authors do not use the OMI-based Kleipool et al [2008, 2010] databases?

We used surface albedo at 388 nm of the OMI LER data (Kleipool et al., 2008, 2010) and set the same value for surface albedo at 354 nm based on Herman et al. (1997). Since this is proved to be not generally valid, we have re-run the simulation that use 354 and 388 directly based on Kleipool et al. (2008, 2010). Although in the radiative transfer calculation, due to the round-off, the results may not be significantly changed.

Line 131: In the description of the OMAERUV record, the authors list only the UVAI and ALH and omit the fact that both AOD and SSA are reported retrieved parameters. After all, the reason why the ALH is included in the OMAERUV product is that the inversion requires information on ALH. A more candid description of the OMAERUV product should read '....long-term UVAI, AOD and SSA with corresponding ALH....'

The part is moved to Section 3.2. The sentence has changed into: *We choose OMAERUV because it is currently the only product containing a long-term UVAI, AOD, SSA and corresponding ALH (Torres et al., 2007; 2013). (line 243-246)*

Line 142: Discuss the error bars and whiskers on Fig 2, particularly for the SSA. What are the implications of the expected diurnal variability?

This figure is no longer in the manuscript. This figure is used to provide an overview of AERONET version 2 inversion product availability for the first case we chosen. There was only one day in 2017 December that captures the plume generated by the fire event meanwhile there are TROPOMI UVAI and ALH available. But now we find more cases, thus it is no longer necessary to show this plot.

The diurnal variability of SSA may be caused by the changes of aerosol types, meteorological conditions (cloud contamination, wind direction, humidity, etc.), combustion phases (for biomass burning aerosols), as well as the measuring period, etc.

Line 144: The time difference between TROPOMI and AERONET observations on December 12, 2017 is about 2.5 hours. Discuss the implication of that time difference in the context of the AERONET results in Fig 2.

In the manuscript, we always use the time window of 3 hours to collocate AERONET and corresponding satellite measurements, as what is done in Jethva et al., 2014. The time window is used to exclude AERONET measurements during early morning or late afternoon meanwhile ensures there are sufficient records available. All records within the time window are averaged into one value, which implies that we accept the SSA discrepancy due to the time difference as long as the time difference is within 3 hours. Moreover, since there is only one record, it is almost impossible to determine whether this record is the truth or just an outlier.

Nevertheless, we have added description on the effects of the time difference: *Although our retrieved SSA seems better than that provided in OMAERUV, one should keep in mind that there is only one record for this event, the meteorological conditions, combustion phases and even the aerosol compositions may change during the 3-hour time difference. (line 183-185)*

Line 145: Both AERONET SSA and TROPOMI are results of inversions in which multiple solutions are possible. Thus, an inversion cannot be validated with another inversion. Use 'compare' instead of 'validate'. Use 'comparison' instead of 'validation' in all instances in the paper where the word 'validation' is used.

We have changed them accordingly through the manuscript.

Line 146: Use AERONET version 3 data in the construction of the training data set. There are significant differences between version 2 and 3 of the AERONET inversion product.

We have replaced the AERONET inversion product into version 3 for all the places in the revised manuscript where AERONET data is used.

Line 170. Provide the reasoning to conclude that the southern part of the plume is the most absorbing region. All it can be said, is that the largest UVAI is observed in that region, but AOD, ALH and spectral aerosol absorption exponent affect the magnitude of the resulting UVAI.

*The sentence has been changed into: The highest UVAI appeared at the south part of the plume, where both aerosol loading and aerosol layering are relatively high (AOD > 2 and ALH is over 2.5 km). (line 159-160)*

Line 187. Assuming constant refractive index for wavelengths longer than 388 nm is not a reasonable assumption. At longer wavelengths, the role of black carbon is more important. Discuss the implication of this assumption on the reported results.

Because the spectral dependence of the refractive index between UV and visible band is also not well-understand, and we do not know the exact compositions in the smoke plume. We thus assumed it is 'gray' within this range to keep it simple and only investigate the influences due to spectral dependence in the near-UV range. We thought in previous OMAERUV product the biomass burning aerosols are assumed to be spectrally flat (Jethva and Torres, 2011).

We have added the explanation: *As we only investigate the influence due to aerosol absorption spectral dependence in near-UV range in this study, aerosol absorption at wavelengths larger than 388 nm is set equal to that at 388 nm. (line 122-124)*

Line 193: The 'existing' MODIS AOD and TROPOMI ALH retrievals involve assumptions on particle size distribution (PSD) and aerosol single scattering albedo. Are the assumed PSD's in the two algorithms consistent? How about the complex refractive indices assumed in the AOD and ALH retrieval? Please list the values of those parameters and discuss the implication of inconsistencies if any.

The 'existing' measurements here indicates the source for training data set, which consists of OMAERUV UVAI and ALH, and AERONET AOD and AAOD. The MODIS AOD and TROPOMI ALH are only used in the case application phase (prediction phase), rather than in the training phase. We have noticed the inconsistency you mentioned due to the different aerosol models in two independent algorithms. The inconsistency itself is not the most interested to the SVR model, but the retrieval bias caused by the a priori aerosol models matters. As you also suggested in other comment, we have provided the error analysis of retrieved SSA due to the uncertainties of input UVAI, ALH and AOD in Section 3.5.

But for your request, we will answer your question in this document. There is inconsistency in aerosol models between MODIS/MYD04 product and TROPOMI/ALH product. The MODIS aerosol models can refer to Remer et al. (2006) and Levy et al. (2013). The global evaluation shows that over 66% Dark Target retrievals are within uncertainty envelop of $\pm 0.05 + 15\% \ast AOD_{AERONET}$, and the uncertainty of Dark Blue retrievals is $\pm 0.03 + 0.2 \ast AOD_{MODIS}$ (Sayer et al., 2013).

The aerosol models in TROPOMI ALH product can refer to Sanders et al. (2016). Aerosols are characterized by SSA of 0.95 and Henyey-Greenstein phase function with asymmetry factor of 7. These values are the averages of long-term AERONET observations for all aerosol types. Retrieved ALH is insensitive to errors in SSA (even with error as large as 0.2). The bias due to SSA is usually smaller over ocean and lager over land, but generally meets the TROPOMI target accuracy of 50 hPa. The algorithm is robust over dark surface even with incorrect knowledge of the phase function. Over bright surface, however, the ALH bias depends on the aerosol loading. For AOD at 550 nm above 0.4, the bias is typically smaller than 50 hPa.

Moreover, there is a long-term downward trend in the magnitude of TROPOMI irradiance (Rozemeijer and Kleipool, 2018), which results in the degradation in UVAI. The degradation is around 0.2 since from August 2018 to June 2019 (Lambert et al., 2019)

We have added descriptions of the uncertainty of MODIS AOD and TROPOMI ALH: *The error the retrieved SSA due to the input features may come from the observational or retrieval uncertainties in each parameter. In our case, the typical UVAI bias requirement is at magnitude of 1 (Lambert et al., 2019). It is reported TROPOMI UVAI suffers from the long-term downward wavelength-dependent trend in irradiance (Rozemeijer and Kleipool, 2018). The detected degradation in $UVAI_{354,388}$ is around 0.2 since August 2018 (Lambert et al., 2019). The typical accuracy of TROPOMI ALH is 50 hPa, though in some situations the bias may over this value (e.g. low aerosol loading over bright surface) (Sanders et al., 2016). Depending on the retrieval algorithm the uncertainty of MODIS AOD is $\pm 0.05 + 15\% AOD_{AERONET}$ (Dark Target algorithm) (Levy et al., 2010) or $\pm 0.03 + 0.2 AOD_{MODIS}$ (Deep Blue algorithm) (Sayer et al., 2013). (Deep Blue algorithm) (Sayer et al., 2013). (line 318-325)*

Line 206: Please describe in more detail the implied statistical analysis of 13 years of data involving the OMAERUV and AERONET data sets. What are the parameters being examined?

The parameters to be examined are: UVAI, ALH, measurement geometries, surface albedo and surface pressure in OMAERUV, and SSA, AOD and AAOD from AERONET.

In the revised manuscript, they are located at line 252-254: *The parameters in OMAERUV-AERONET joint data set for feature selection consists of UVAI calculated by 354 and 388 nm wavelength pair, satellite geometries, surface conditions and ALH from OMAERUV, and SSA, AOD and AAOD from AERONET.*

Line 208: The 13-year OMAERUV global dataset includes AOD and SSA, a record that the authors claim back on page 31, does not exist.

Sorry we are not clear about this comment as we only have 24 pages in the last version manuscript.

Line 212: I am totally lost here. The statement *'samples are excluded if the SSA difference between OMI and AERONET are larger than 0.03 or the AOD difference between OMI and AERONET is larger than 5%'* is incomprehensible. What OMI SSA/AOD are the authors talking about? Are these OMAERUV-retrieved values? Up to this point in the manuscript, the authors have not acknowledged the existence of such products. If these are indeed the OMAERUV SSA/AOD, then the authors have created a dataset consisting of the best quality OMAERUV AOD and SSA retrievals (as measured by the level of agreement with AERONET) to train the SVR algorithm. It is suggest that the description OMAERUV-SSA and OMAERUV-AOD be used (instead of the generic OMI-SSA or OMI-AOD) to avoid confusing the reader since there is a second OMI aerosol algorithm (OMAERO).
Sorry for the confusion. They indicate OMAERUV-SSA and OMAERUV-AOD. We have changed it accordingly in the revised manuscript.

Line 228: Add Torres et al 2013 reference to the CALIOP ALH climatology.

We have added it accordingly: *The best-guessed ALH in the OMAERUV is a combined information of CALIOP climatology or assumed ALH in the retrieval (if the CALIOP climatology is not available) (Torres et al., 2013). (line 246-247)*

Line 232: The UVAI height dependence was first documented [Herman at al., 1997; Torres et al., 1998] based on analysis of TOMS data.

Due to the content changes, this sentence is no longer applicable in the revised manuscript.

Line 235: If spectrally dependent AOD (354 and 388 nm) and ALH are indeed independently know, one should be able to retrieve the SSA at 354 and 388 nm via a direct RTM inversion of the 354 and 388 nm radiances (not the UVAI). This is the simplest RTM approach that would fully characterize the aerosol plume.

Indeed, the method mentioned here is true. But it is still inevitable to assume aerosol micro-physics (size distribution functions, refractive index) if one uses RTM approach. The reason we propose SVR method is to avoid such kind of assumptions and retrieve SSA from empirical measurements.

Line 241: Fig 7(c) is not mentioned in the discussion. Remove it if not needed. Otherwise, explain, or eliminate, the difference between UVAI OMI and UVAI OMAERUV in the z-axis label of figures 7(b) and 7(c).

This figure is no longer applicable. The purpose of this set of 3D plots is to show the difference between UVAI-AOD-ALH relationship in OMAERUV-AERONET and that in TROPOMI-MODIS (for California fire 2017-12-12 only) data set. But the TROPOMI-MODIS joint data is only for elevated absorbing aerosol layers.

We only present the relationship of parameters in OMAERUV-AERONET joint data sets in Figure 6 in the revised manuscript.

Line 245: As described the ALH adjustment sounds arbitrary. It looks to me the authors are just conveniently making up a convenient dataset. Please provide an understandable rationale for the creation of this ALH dataset.

We used to create an adjusted ALH for better SSA retrieval, as the ALH provided by the OMAERUV product is not retrieved by wither from CALIOP climatology or from a priori assumptions during the AOD retrieval. In the previous version manuscript (when AERONET version 2 inversion was used), the results indeed show a slightly better results compared with the results directly retrieved from OMAERUV-AERONET data set. But both results are within

725 AERONET typical uncertainty (0.03). Besides, you commented that it would be over-interpretation if SSA difference is 0.03.

Consequently, we remove the ALH adjustment part in the manuscript, i.e., we only use OMAERUV-AERONET data set to train the SVR model. Please read the Section 3.2 in the revised manuscript. The SSA retrieval is acceptable, with 6 out of 9 AERONET-collocated samples fall within 0.03 difference and all the samples are within 0.05 difference.
730 (Section 3.6)

Line 250 There is no mention in this work of the Oxygen-A band AOD that is simultaneously retrieved with ALH from TROPOMI observations. Shouldn't it be better to use the TROPOMI AOD rather than the MODISD AOD? That would eliminate possible implicit inconsistencies in aerosol microphysics.

It is planned that AOD is simultaneously retrieved with ALH in TROPOMI, but the AOD product has not been
735 operational yet (thus not available).

Line 259: What does 'rule of thumb ratio 70% versus 30%' really mean? This all sounds arbitrary.

It just means the ratio is chosen based on experience. It is quite arbitrary, but normally the fraction of training data set is 60-90%, and the fraction of test data set is 10-40%.

Line 290: Figures should be described sequentially. From the description of figures 7(a) and (7b), the authors jump to
740 Figure 5, and then back to Figure 7(c).

Figure 7 is no longer necessary thus we have deleted it as explained in the previous response.

Line 301: The MODIS AOD uncertainty needs to be taking into account and propagated in a sensitivity analysis of the SVR application. Over the US west coast, in particular, the AOD is subject to large uncertainty due to surface effects [Jethva et al., 2019].

745 The error analysis has been added in Section 3.5 in the revised manuscript. Figure 9 presents the sensitivity of retrieved SSA to the changes in UVAI, AOD and ALH.

Line 323: The difference of 0.01 between the two SVR applications has not statistical meaning, as they are both within the stated AERONET uncertainty of ± 0.03 for a single measurement. Any over-interpretation is just splitting hairs.

Thank you for the correction. We have modified our interpretation on the retrieved SSA. The comparison between
750 retrieved SSA, AERONET SSA, as well as OMAERUV SSA is summarized in Table 4 and described in Section 3.6. The difference between SVR retrieved SSA and AERONET is always within 0.05, and 6 out of 9 collocated samples are within difference of 0.03. Compared with OMAERUV, the SVR retrieved SSA is in better agreement with that of AERONET.

Line 326: I disagree with this statement. No measurable improvement in performance is apparent from this
755 comparison. The use of the adjusted ALH instead of the original OMAERUV ALH makes no statistically quantifiable difference whatsoever. A more systematic analysis using a large number of independently measured SSA values is required.

Thank you for the correction. But as the AERONET product has been changed into version 3, some of the results and conclusions are also changed accordingly, and more case studies has been involved. For details please see Section 3.6.

760 Line 328: In section 4.2, the authors try interpreting the lower spatial variability over the entire plume of the SVR retrieved SSA with respect to the SSA spatial variability resulting from the RTM-based approach, as an indication of better SVR accuracy. The north-south extent of the plume over the ocean is about 1000 km whereas the east-west dimension varies from about 200 to 700 km. For an aerosol plume this large, it is not unreasonable to expect spatial variability in SSA. The SSA of carbonaceous aerosols from biomass burning or wild fires is lowest near the source
765 areas in the flaming phase of the fires. As the resulting smoke layer is transported downwind, it interacts with the surrounding air. Aerosol SSA may increase due to water uptake by hydrophilic particles. The resulting SSA homogeneity over the plume may therefore depend on the homogeneity of meteorological fields.

I do not think this study conclusively demonstrates that the described SVR technique yields more accurate retrieval than standard well thought out RTM approaches. Undoubtedly, however, the availability of TROPOMI ALH observations will improve the accuracy of retrieved aerosol absorption.

Thank you for the correction. The plume spatial variability of this size may exist, depends on the meteorological conditions, combustion phase, etc. But whether a variation from 0.69 to 0.97 is reasonable needs further investigations.

We have added an independent data from a chemistry transport model (i.e. MERRA-2) as a reference to compare with. The MERRA-2 data for each case study is provided in Appendix C. According to MERRA-2, the SSA spatial variability is at magnitude of around 0.1, which is smaller than that in Experiment 1. On the other hand, the spatial contrast of SVR-retrieved SSA is rather flat, with the largest difference from 0.07 to 0.13 (Table 4), depending on cases.

We have modified the explanation: *Depending on the combustion phase and meteorological conditions, heterogeneity in aerosol properties is expected for plume of this size. Nevertheless, whether such a large SSA difference 0f 0.38 (maximum SSA – minimum SSA, Table 2) is reasonable needs further investigations (discussed in Section 3.6.3). (line 192-195)*

We also have added a section to analyze the spatial variability (Section 3.6.3).

Line 345-347: The statement '*In cloud-free cases, it is expected that micro-physical properties of smoke particles within the plume should be similar over short time periods as they were originated from the same source and generated under the same conditions..*' is not always correct. The variability over a large smoke plume like the one used in this analysis may be important.

Thank you for the correction. We have modified our description into: *Depending on the combustion phase and meteorological conditions, heterogeneity in aerosol properties is expected for plume of this size. Nevertheless, whether such a large SSA difference 0f 0.38 (maximum SSA – minimum SSA, Table 2) is reasonable needs further investigations (discussed in Section 3.6.3). (line 192-195)*

Line 364: The statement that the proposed method based on the correlation between UVAI, AOD and ALH requires no a priori assumptions on aerosol micro-physics is incorrect. Implicit microphysics assumptions are involved in the use of MODIS AOD as well as TROPOMI ALH. The authors have ignored this fact, and treat the AOD and ALH as 'given true values', ignoring the fact that these parameters come out as the result of RTM-based inversions that assume particle size distribution, and complex refractive index over an extended spectral range. The results of a sensitivity analysis that propagates AOD and ALH retrieval uncertainties into the SVR method should be included in the paper.

Sorry for the confusion, when we say, 'no a priori assumptions on aerosol micro-physics', it indicates that we do not have to make such assumptions by ourselves as that in the Experiment 1 (to run radiative simulations. Aerosol models in Experiment 1 (or refer toTable 1) may trigger additional uncertainties in the retrieved SSA. By contrast, the SVR model retrieves SSA based on the given data set without additional assumptions from us.

In the revised manuscript, we have tried to emphasize that the SVR is free from the a priori assumptions made in the radiative transfer simulations in Experiment 1 as clear as possible. For example:

*From our perspective, ML techniques can avoid making assumptions on poorly-understand aerosol micro-physics as that in the first experiment. (line 75-76)*

*An inappropriate assumption may lead to significant bias in retrieved SSA (Fig.3). On the other hand, SVR (and other ML algorithms) is applicable to solve ill-posed inversion problems by learning the underlying behavior of a system from a given data sets without such a priori knowledge on aerosol micro-physics. (line 200-202)*

*In the second part of this paper, we propose a statistical method based on the long-term records of UVAI, AOD, ALH and AAOD using an SVR algorithm, in order to avoid making assumptions on aerosol absorption spectral dependence over near-UV band. (line 407-409* The error the retrieved SSA due to the input features may come from the observational or retrieval uncertainties in each parameter. In our case, the typical UVAI bias requirement is at magnitude of 1 (Lambert et al., 2019). It is reported TROPOMI UVAI suffers from the long-term downward wavelength-dependent trend in irradiance (Rozemeijer and Kleipool, 2018). The detected degradation in $UVAI_{354,388}$ is around 0.2 since August 2018 (Lambert et al., 2019). The typical accuracy of TROPOMI ALH is 50 hPa, though in

815  some situations the bias may over this value (e.g. low aerosol loading over bright surface) (Sanders et al., 2016). Depending on the retrieval algorithm the uncertainty of MODIS AOD is $\pm0.05+15\%AOD_{AERONET}$ (Dark Target algorithm) (Levy et al., 2010) or $\pm0.03+0.2AOD_{MODIS}$ (Deep Blue algorithm) (Sayer et al., 2013).*)*

The error analysis has been added in Section 3.5 in the revised manuscript. Figure 9 presents the sensitivity of retrieved SSA to the changes in UVAI, AOD and ALH.

820  Line 365: The statement '*a priori assumptions on aerosol microphysics is considered one of the major error sources in RTM-based method*' is misleading. I should read instead 'wrong *a priori assumptions ..*'

This content has been moved to Section 2.2. The sentence has been changed into: *The large variability in retrieved SSA (from 0.69±0.13 to 0.94 ±0.03) demonstrates that inappropriate assumptions on the spectral dependence of near-UV aerosol absorption may significantly bias interpretations of smoke aerosol absorption and should be carefully*
825  *handled in forward radiative transfer calculations. (line 165-167)*

Line 368: '*Convincing..*' is not an objective characterization. I was not convinced as stated in this review.

We have avoided using this term in the revised manuscript.

[revised manuscript text omitted]

Apart from pre-assumed aerosol micro-physics, the aerosol loading and the aerosol vertical distribution are two key parameters in forward simulations of UVAI. The former is usually provided in terms of the aerosol optical depth (AOD). There are plentiful AOD products providing wide spatial-temporal coverage with various spectral choices. By contrast, only little information on the aerosol vertical distribution is available. The most well-known aerosol profile product is offered by the Cloud-Aerosol Lidar with Orthogonal Polarization (CALIOP), but the number of measurements is limited because of its narrow tracks (Winker et al., 2009). Passive sensors only measure columnar quantities but, in some cases, also provide the aerosol layer height (ALH), a compact form of aerosol profile information indicating where most aerosols are located. Chimot et al. (2017) present the feasibility of ALH retrieval using the oxygen (O₂-O₂) band at 447 nm of the Ozone Monitoring Instrument (OMI), but so far it has not been run operationally yet. ¶
Recently, a new ALH algorithm based on the near-Infrared (NIR) O₂ A-band has been developed for the TROPOspheric Monitoring Instrument (TROPOMI) on board the Copernicus Sentinel-5 Precursor (S-5P) (Sanders et al., 2015). TROPOMI was launched on 13 October 2017. The instrument is equipped with both the UV-visible (270–500 nm) and the near-infrared (NIR) (675–775 nm) channels (Veefkind et al., 2015), which makes it possible to interpret UVAI using corresponding ALH measurements. Furthermore, TROPOMI has a wide swath of 2600 km, providing daily global coverage with a high spatial resolution of 7×3.5 km² in nadir.¶
The purpose of this paper is to demonstrate the potential of the TROPOMI ALH product for quantifying aerosol absorption.

**Moved up [1]:** As the TROPOMI ALH product is not operationally available yet, we focus on the data of one of the largest wildfires that happened in southern California in 2017, i.e. the Thomas Fire (http://www.fire.ca.gov/current_incidents/incidentdetails/Index/1922 ). Ignited on 4 December 2017, the fire was expanded quickly northwest by the strong and persistent Santa Ana winds and was fully under control on 12 January 2018. The precise cause of the fire remains unknown, but a prolonged period of heat and absence of precipitation definitely contributed to this devastating fire (https://inciweb.nwcg.gov/incident/5670/ ). We selected one day (12 December 2017) for our case study. As shown in Fig.1, 
[revised manuscript text omitted]

**Moved (insertion) [2]**

**Moved up [2]:** UVAI offline data can be accessed via http://doi.org/10.5270/S5P-0wafvaf). The TROPOMI UVAI is calculated for two different wavelength pairs. One uses the conventional 340 and 380 nm to continue the heritage of UVAI records from multiple sensors, and the other uses 354 and 388 nm in order to allow comparison with OMI measurements (D.C. Stein Zweers, 2016). In this study we retrieve the SSA based on the latter pair. Satellite measurement geometries (solar/viewing zenith angle $\theta_0/\theta_v$, solar/viewing azimuth angle $\varphi_0/\varphi_v$) and the surface pressure ($P_s$) included in the UVAI product are input for the radiative transfer calculations. The scene albedo ($A_{sc}$) from the same product is also used in the pre-processing as will be described later.

¶
Data sets listed in this section are either used by the RTM-based method or the SVR-based method, or both. The pre-processing and detailed usage of those data are explained in section 3. ¶
**2.1** TROPOMI satellite data¶
In this study, we employ the TROPOMI Level 2 reprocessed UVAI product to quantify aerosol absorption for the target fire event (TROPOMI UVAI data on 12 December 2017 is only internally available, last access: 19 June 2018. UVAI offline data can be accessed via http://doi.org/10.5270/S5P-0wafvaf). The TROPOMI UVAI is calculated for two different wavelength pairs. One uses the conventional 340 and 380 nm to continue the heritage of UVAI records from multiple sensors, and the other uses 354 and 388 nm in order to allow comparison with OMI measurements (D.C. Stein Zweers, 2016). In this study we retrieve the SSA based on the latter pair. Satellite measurement geometries (solar/viewing zenith angle $\theta_0/\theta_v$, solar/viewing azimuth angle $\varphi_0/\varphi_v$) and the surface pressure ($P_s$) included in the UVAI product are input for the radiative transfer calculations. The scene albedo ($A_{sc}$) from the same product is also used in the pre-processing as will be described later. ¶ ... [3]

**Moved (insertion) [4]**

[revised manuscript text omitted]

The values chosen by the above methods are robust in our case (Fig.B1-B3 in the Supplement, part B), i.e. retaining a relatively low error while preventing overfitting. Table 3 summaries the settings of the SVR models determined by tuning procedure and the evaluation of the algorithm performance. All 3 SVR models present good generalization capabilities as the differences in root mean square error (RMSE) between the training data and the testing data are minor. The accuracy of the SVR model for ALH prediction is 0.26 km. Fig.7c shows the relationship of UVAI, AOD and the SVR predicted ALH. The structure is more similar to that in Fig.7a and |ρ| between UVAI and ALH increases from 0.30 to 0.61, which is sufficient to mitigate the impact of uncertainties of ALH in the OMAERUV product. Note that this value may be overestimated as the SVR for ALH prediction is only trained by a specific case due to the limited availability of TROPOMI ALH, but it is more reasonable compared with the original UVAI and ALH relationship in the OMI-AERONET data. The predicted ALH, together with OMI UVAI, AERONET AOD and AAOD provides a new training data set for AAOD prediction, i.e. the adjusted training data set. The accuracy of SVR models for AAOD prediction trained by the original and the adjusted training data set are 0.01.¶

730 brighter surface may cause higher bias in the input AOD and ALH than cases over dark surfaces (Remer 2005; Sanders and de Haan, 2016).

The retrieved SSA for above events is listed in Table 4. Similar to the California 2017-12-12 case, The SVR fails to retrieve reasonable SSA for pixels if input features outside their corresponding histogram in the OMAERUV-AERONET data (Fig.2B), which may cause overestimations in plume mean SSA. The plume SSA of two California

735 fire events are similar, with values around 0.92-0.93. The retrieved SSA for the Canada fire is relatively higher (0.96) We further plot the SSA retrieved by SVR against collocated AERONET records (Fig.15). Including the first case (California fire on 2017-12-12), there are 9 collocated records obtained. The difference between SVR-retrieved SSA and AERONET are all within difference of $\pm 0.05$, among which 6 of them (66%) fall in AERONET SSA uncertainty range ($\pm 0.03$). We also provide SSA from OMAERUV for these cases (Table 4 and Fig.15). Compared with

740 OMAERUV, the SSA retrieved by SVR shows a better consistency with AERONET, though one should keep in mind that the accuracy of SVR-retrieved SSA is $\pm 0.02$ and the model tends to overestimate the SSA for relatively absorbing cases.

**3.6.3 Spatial variability of retrieved SSA**

Compared with Fig.4b, the spatial variability of SSA retrieved by SVR is less strong (Fig.10-14), whose difference

745 between maximum and minimum SSA falls in range from 0.07 to 0.13 (Table 4). In the first experiment, SSA is determined by UVAI for each pixel individually. In the SVR model, the spatial variability of the intermediate output AAOD depends on the three input features. Furthermore, SVR predicts SSA for each pixel based on the common relationship between UVAI, AOD and ALH in the training data set.

Heterogeneity in aerosol properties is expected for plume of this size, but to what extend needs further investigations.

750 Here we assess the SSA spatial variability of by an independent data set. We employ the SSA calculated by AOD and scattering AOD from MERRA-2 aerosol reanalysis hourly single-level product (https://disc.gsfc.nasa.gov/datacollection/M2T1NXAER_5.12.4.htm last access: 16 July 2019). The AOD and aerosol properties of MERRA-2 are proved to be in good agreement with independent measurements (Buchard et al., 2017; Randles et al., 2017). The MERRA-2 AOD and SSA for these cases are shown in Appendix C. The plume can be

755 detected by the high AOD against its surrounding. Although the plume presented by the satellite observations significantly differs from that of model simulations, the SSA spatial difference within the plume is approximately at magnitude of 0.1. From this aspect, the spatial variability of SSA retrieved by the SVR model is in better agreement with MERRA-2.

**4 Conclusions**

760 The long-term record of global UVAI data is a treasure to derive aerosol optical properties such as SSA, which is important for aerosol radiative forcing assessments. To quantify aerosol absorption from UVAI, the information of AOD and ALH is necessary. There are various AOD products while ALH products are much less accessible. Recently, the TROPOMI oxygen A-band ALH product has been run operationally, using which we demonstrate the role of ALH in quantifying SSA from satellite retrieved UVAI for biomass burning aerosols.

765 In the first experiment, we derive the SSA by forward radiative transfer simulation of UVAI for a fire event in California on 2017-12-12. With the TROPOMI ALH, we are able to quantify the influence of assumed spectral dependence of near-UV aerosol absorption (represented by the relative difference between $\kappa_{354}$ and $\kappa_{388}$) on the

We first analyze the results of the RTM-based method. Fig.8a displays the mean SSA of all plume pixels retrieved by the RTM-based method as a function of $\Delta \kappa$. The retrieved aerosol absorption decreases with $\Delta \kappa$. This finding is in good agreement with Jethva and Torres (2011). 'Gray' aerosols require stronger absorption to reach the same level of UVAI than 'colored' aerosols. This also explains the high SSA standard deviation (filled area) in the cases with little or no spectral dependence in aerosol absorption. The large variability in retrieved SSA (from 0.71 $\pm 0.09$ to 0.94 $\pm 0.03$) demonstrates that assumptions on the spectral dependence of near-UV aerosol absorption may significantly bias our interpretations of smoke aerosol absorption and should be carefully handled in forward radiative transfer calculations. ¶
The retrieved aerosol absorption is evaluated with the nearby AERONET site (UCSB), whose SSA at 500 nm at 19:55:07 UTC is 0.96. There are 24 TROPOMI pixels located within 50 km from the UCSB site. Hereafter we call them validation pixels. As illustrated in Fig.8b, the mean SSA of the validation pixels also increases with $\Delta \kappa$ and eventually levels off at around 0.96. The extremely low SSA (0.53) and high variation ($\pm 0.37$) retrieved for 'gray' aerosols prove that the spectral independence assumption is not recommended for smoke aerosols (at least in this case). The differences between the mean SSA of the validation pixels and the AERONET measurement are shown in Fig.8c. The retrieved SSA falls inside the uncertainty range of AERONET ($\pm 0.03$) (Holben et al., 2006) when $\Delta \kappa$ is larger than 15%. This indicates that the assumption of a 20% $\Delta \kappa$ in the OMAERUV algorithm to describe the spectral dependence of aerosol absorption is adequate. Only a slightly better estimate is found when … [25]

[revised manuscript text omitted]

- **TROPOMI:** $\theta_0, \theta_v, \varphi_0, \varphi_v,$ *Ps, ALH*
- **MODIS:** $AOD_{550}$
- **OMI climatology:** $As_{388}$

**TROPOMI:** $UVAI_{354,388}$

[Figure]

**Figure 5: Procedure of the support vector regression (SVR).**

[Figure]

(a) UVAI$_{354, 388}$

[Figure]

(d) Θ [∘]

Figure 4: Data involved in RTM-based method: (a) TROPOMI UVAI calculated by reflectance at 354 and 388 nm; (b) TROPOMI ALH; (c) MODIS AOD at 550 nm; (d) TROPOMI scattering angle **Θ** (calculated from $\boldsymbol{\theta_0}$, $\boldsymbol{\theta_v}$, $\boldsymbol{\varphi_0}$ and $\boldsymbol{\varphi_v}$); (e) TROPOMI surface pressure $P_s$; (f) OMI $A_s$ climatology at 388 nm. All parameters shown here are projected onto TROPOMI UVAI grid. ¶

Moved (insertion) [3]

[Figure]

(a) SSA$_{500}$ (RTM-based Δκ=25%)

Figure 9: SSA retrieved by the: (a) RTM-based method; (b) SVR-based method with OMAERUV ALH; (c) SVR-based method with [28]

[Figure]

[Figure]

¶

**Figure 6: Spearman's rank correlation coefficient matrix ($\rho$) of parameters in the OMAERUV-AERONET joint data set.**

150

155

160

[Figure]

[Figure]

165

[Figure]

**Figure 7: The performance of SVR model as a function to hyper-parameters (C, $\varepsilon$ and $p$). The cross marker represents the values of C and $\varepsilon$ according to Cherkassky and Ma (2004). $p^2$ equaling 1.67 is sufficient to obtain a relatively high accuracy, meanwhile prevents overfitting on the training data set.**

175

[Figure]

**Figure 8: The accuracy of the trained SVR model: (a) the predicted AAOD at 500 nm against true AAOD at 500 nm. The dashed line is the 1:1 line and the solid line is the linear fitting for the testing data set; (b) the predicted SSA at 500 nm against true SSA at 500 nm. The grey and red color indicates samples in training and testing data set, respectively. The values inside parenthesis is the statistics for samples fall in AERONET uncertainty of 0.03.**

[Figure]

**Figure 9: The sensitivity of the SVR-retrieved SSA: (a) the response of predicted SSA at 500 nm as a function of changes in UVAI and ALH; (b) the response of predicted SSA at 500 nm as a function of changes in UVAI and AOD.**

[Figure]

[Figure]

**Figure10: SVR retrievals for California fire event on 2017-12-12: (a) retrieved AAOD at 500 nm; (b) retrieved SSA at 500 nm.**

[Figure]

[Figure]

[Figure]

[Figure]

**Figure11: SVR retrievals for California fire event on 2018-11-09: (a) TROPOMI UVAI calculated by reflectance at 354 and 388 nm; (b) TROPOMI ALH; (c) MODIS AOD at 550 nm; (d) retrieved AAOD at 500 nm; (e)  retrieved SSA at 500 nm.**

[Figure]

**Figure12: SVR retrievals for California fire event on 2018-11-10: (a) TROPOMI UVAI calculated by reflectance at 354 and 388 nm; (b) TROPOMI ALH; (c) MODIS AOD at 550 nm; (d) retrieved AAOD at 500 nm; (e) retrieved SSA at 500 nm.**

**Figure13: SVR retrievals for California fire event on 2018-11-11: (a) TROPOMI UVAI calculated by reflectance at 354 and 388 nm; (b) TROPOMI ALH; (c) MODIS AOD at 550 nm; (d) retrieved AAOD at 500 nm; (e) retrieved SSA at 500 nm.**

[Figure]

**Figure14: SVR retrievals for Canada fire event on 2019-05-29: (a) TROPOMI UVAI calculated by reflectance at 354 and 388 nm; (b) TROPOMI ALH; (c) MODIS AOD at 550 nm; (d) retrieved AAOD at 500 nm; (e) retrieved SSA at 500 nm.**

[Figure]

220

[Figure]

**Figure15: SVR-retrieved SSA (black cross) and OMAERUV-retrieved SSA (blue circle) against AERONET SSA at 500 nm for all 5 cases in this study.**

Figure 8: SSA retrieved by the RTM-based method as a function of $\Delta\kappa$ (the relative difference between $\kappa_{354}$ and $\kappa_{388}$): (a) SSA mean and standard deviation (filled region) of 4808 plume pixels; (b) SSA mean and standard deviation (filled region) of the 24 validation pixels; (c) absolute difference between the mean SSA of the 24 validation pixels and the AERONET measurement.

¶
¶
¶

225

[Figure]

[Figure]

[Figure]

**Moved up [3]:**
Figure 9: SSA retrieved by the: (a) RTM-based method; (b) SVR-based method with OMAERUV ALH; (c) SVR-based method with adjusted ALH.¶

[revised manuscript text omitted]

Font: 10.5 pt

| Page 28: [7] Formatted | Sun ji | 7/18/19 5:36:00 PM |
|---|---|---|

Font: 10.5 pt

| Page 28: [8] Formatted | Sun ji | 7/18/19 5:36:00 PM |
|---|---|---|

Font: 10.5 pt

| Page 28: [9] Deleted | Sun ji | 7/20/19 11:21:00 AM |
|---|---|---|

| Page 28: [10] Deleted | Sun ji | 7/20/19 12:11:00 PM |
|---|---|---|

| Page 28: [11] Formatted | Sun ji | 7/20/19 12:16:00 PM |
|---|---|---|

Formatted

| Page 28: [12] Deleted | Sun ji | 7/20/19 12:20:00 PM |
|---|---|---|

| Page 28: [13] Formatted | Sun ji | 7/18/19 5:36:00 PM |
|---|---|---|

Font: 10.5 pt

| Page 28: [14] Formatted | Sun ji | 7/18/19 5:36:00 PM |
|---|---|---|

Font: 10.5 pt

| Page 28: [15] Formatted | Sun ji | 7/18/19 5:36:00 PM |
|---|---|---|

Font: 10.5 pt

| Page 28: [16] Formatted | Sun ji | 7/18/19 5:36:00 PM |
|---|---|---|

Font: 10.5 pt

| Page 28: [17] Formatted | Sun ji | 7/18/19 5:36:00 PM |
|---|---|---|

Font: 10.5 pt

| Page 28: [18] Formatted | Sun ji | 7/18/19 5:36:00 PM |
|---|---|---|

Font: 10.5 pt

| Page 28: [19] Formatted | Sun ji | 7/18/19 5:36:00 PM |
|---|---|---|

Font: 10.5 pt

| Page 28: [20] Deleted | Sun ji | 7/20/19 12:32:00 PM |
|---|---|---|

| | | |
|---|---|---|
| **Page 28: [21] Deleted** | **Sun ji** | **7/20/19 12:26:00 PM** |

| | | |
|---|---|---|
| **Page 28: [22] Deleted** | **Sun ji** | **7/18/19 4:17:00 PM** |
| **Page 28: [23] Deleted** | **Sun ji** | **7/18/19 4:17:00 PM** |

| | | |
|---|---|---|
| **Page 28: [24] Formatted** | **Sun ji** | **7/18/19 5:36:00 PM** |

Font: 10.5 pt

| | | |
|---|---|---|
| **Page 31: [25] Deleted** | **Sun ji** | **7/20/19 2:04:00 PM** |

| | | |
|---|---|---|
| **Page 37: [26] Deleted** | **Sun ji** | **7/20/19 11:51:00 AM** |
| **Page 37: [27] Deleted** | **Sun ji** | **7/18/19 5:36:00 PM** |
| **Page 38: [28] Deleted** | **Sun ji** | **7/20/19 11:52:00 AM** |

| | | |
|---|---|---|
| **Page 44: [29] Deleted** | **Sun ji** | **7/22/19 8:23:00 AM** |
| **Page 44: [30] Deleted** | **Sun ji** | **7/20/19 1:18:00 PM** |

Values          0.11          0.0001          1.67

| | | |
|---|---|---|
| **Page 44: [31] Deleted** | **Sun ji** | **7/20/19 1:18:00 PM** |

---

## Referee Report (RR1)

The new version is significantly improved in relation to the original.

The authors seem to use the terms OMI aerosol record and OMAERUV interchangeably. For readers unfamiliar with the OMI aerosol products, there may be some confusion, since officially two OMI aerosol records are available. I would suggest the authors to either stick to the 'OMAERUV record' on every instance they refer to it, or include an initial statement stating that the expressions OMAERUV and OMI aerosol record are used Interchangeably throughout in the manuscript.

A few specific comments on the current version are listed below.

Line 22. Replace 'acceptable' with 'encouraging'

Line 34: Just saying that $\lambda_0$ denotes the reference wavelength does not provide useful information. It should be stated that this is the longer wavelength in the pair, and it is used to calculate an assumed wavelength-independent-scene-reflectivity, needed for calculation of the of the $I_\lambda$ Rayleigh term in Equation (1).

Line 42. In addition to the listed aerosol related parameters, the magnitude of the observed UVAI also depends on non-aerosol-related factors such as spectral dependence of land surfaces and ocean color effects, sun-glint effects, and cloud effects. This clarification should be included.

This issue was raised in the earlier review to which the authors reply with the statement
  '… the radiance itself is also affected by surface conditions, clouds, atmospheric gases and aerosols. On the other hand, the UVAI only contains the information of aerosol absorption…'.
 Their reply reveals a poor understanding of the UVAI concept, since the UVAI is just the measured radiance conveniently packed in a single parameter. It includes any observed spectral dependence other than Rayleigh scattering.

Line 151. An explanation for the selection criteria UVAI > 1 and CF larger than 0.3 should be added.  Is this the MODIS CF? If so, 0.3 represents significant contamination.

Line 183. Replace 'better' with 'closer to AERONET retrieved SSA'. As stated AERONET SSA does not represent the true SSA value.

Line 245. Although the same wavelength pair (354, 388) is used in the OMAERUV UVAI definition, the way the observed radiances are calculated have changed [Torres et al., 2018]. The new definition handles the effect of clouds using of Mie Theory (instead of the Lambertian approximation), and reduces significantly the across-track angular dependence. It also accounts for wavelength dependence of surface reflectance as well as sun glint over the oceans. For cloud-free conditions, the Mie-UVAI is slightly larger than LER-UVAI (~0.3) for nadir-viewing conditions. The original definition (consistent with that in TROPOMI) is still reported in the OMAERUV as a parameter labeled 'RESIDUE'.  Thus, the original definition should be used for algorithm training purposes. This is important because the difference between the LER and Mie definitions increases with viewing sensor solar zenith angle.

Line 246.  In the description of the OMAERUV record, it should be stated that AOD retrievals have been validated using the multi-year AERONET aerosol record [Ahn et al., 2014], and, similarly, the SSA parameter have been evaluated with AERONET Almucantar retrievals [Jethva, et al., 2014].

Line 426 'Acceptable' is a subjective interpretation. Replace 'acceptable (+/- 0.02)' with ' +/- 0.02'

Cited References

Torres, O., Bhartia, P. K., Jethva, H., and Ahn, C.: Impact of the ozone monitoring instrument row anomaly on the long-term record of aerosol products, Atmos. Meas. Tech., 11, 2701-2715, https://doi.org/10.5194/amt-11-2701-2018, 2018.

Jethva, H., O. Torres, and C. Ahn (2014), Global assessment of OMI aerosol single-scattering albedo using ground-based AERONET inversion, J. Geophys. Res. Atmos., 119, doi:10.1002/2014JD021672.

Ahn, C., O. Torres, and H. Jethva (2014), Assessment of OMI near-UV aerosol optical depth over land, J. Geophys. Res. Atmos., 119, 2457–2473, doi:10.1002/2013JD020188.

---

## Author Response (AR3)

This document contains the authors' responses to the Editor's comments. The Editor's comments are in black, followed by the author's responses in blue.

The new version is significantly improved in relation to the original.

- 5 The authors seem to use the terms OMI aerosol record and OMAERUV interchangeably. For readers unfamiliar with the OMI aerosol products, there may be some confusion, since officially two OMI aerosol records are available. I would suggest the authors to either stick to the 'OMAERUV record' on every instance they refer to it, or include an initial statement stating that the expressions OMAERUV and OMI aerosol record are used Interchangeably throughout in the manuscript.
- 10 Thank you for your correction. We have added the 'It is also noted that there are two official OMI aerosol level 2 products though, the OMI measurements in this paper only refers to the OMAERUV product.' (line 255-257)

A few specific comments on the current version are listed below.

Line 22. Replace 'acceptable' with 'encouraging'

15 We have changed it accordingly. 'The results are encouraging, though at the current phase, the model tends to overestimate the SSA for relatively absorbing cases and fails to predict SSA for some extreme situations.' (line 21-23)

Line 34: Just saying that  $\lambda_0$  denotes the reference wavelength does not provide useful information. It should be stated that this is the longer wavelength in the pair, and it is used to calculate an assumed wavelengthindependent-scene-reflectivity, needed for calculation of the of the  $I_{-\lambda}$  Rayleigh term in Equation (1).

We have added more introduction of the wavelength pair of UVAI. 'λ is the wavelength where the radiance difference between a Rayleigh and a measured scene is calculated, and λ0 is the longer wavelength where a 25 spectrally constant scene reflectivity is assumed for the calculation of IRay.' (line 34-36)

Line 42. In addition to the listed aerosol related parameters, the magnitude of the observed UVAI also depends on non-aerosol-related factors such as spectral dependence of land surfaces and ocean color effects, sun-glint effects, and cloud effects. This clarification should be included.

30 This issue was raised in the earlier review to which the authors reply with the statement '... the radiance itself is also affected by surface conditions, clouds, atmospheric gases and aerosols. On the other hand, the UVAI only contains the information of aerosol absorption...'. Their reply reveals a poor understanding of the UVAI concept, since the UVAI is just the measured radiance conveniently packed in a single parameter. It includes any observed spectral dependence other than Rayleigh

35 scattering.

- Thank you for your correction. We have modified our presentation accordingly. 'Although non-aerosol factors exist, such as spectral dependence of the surface, ocean color, sun-glint and cloud contamination, the most dominant are aerosol concentration, aerosol vertical distribution and aerosol optical properties (Wang et al., 2012; Buchard et al., 2017).' (line 44-46)
- 40
  - Line 151. An explanation for the selection criteria UVAI > 1 and CF larger than 0.3 should be added. Is this the MODIS CF? If so, 0.3 represents significant contamination.

Sorry for the confusion. The cloud fraction here is the TROPOMI FRESCO cloud fraction. We may forget to add it from the last version manuscript. We have added its source in the manuscript just after introduction of TROPOMI ALH product (1t the same band, there is TROPOMI FRESCO cloud support product providing).

- 45 TROPOMI ALH product. 'At the same band, there is TROPOMI FRESCO cloud support product providing cloud fraction (CF) for mitigating cloud effects as will be explained later (https://scihub.copernicus.eu last access: 19\_Sept\_2018) (Apituley et al., 2017).' (line 139-141)
- The explanation for the pixel selection criteria has been added also. 'Before implementing radiative transfer calculations, pre-processing excludes pixels with large solar zenith angle ( $\theta_0 \ge \underline{70^\circ}$ ), weak aerosol absorption (UVAI354,388 < 1), insignificant aerosol amount (AOD550 < 0.5) or cloud contamination (CF > 0.3).' (line 153-155)

Line 183. Replace 'better' with 'closer to AERONET retrieved SSA'. As stated AERONET SSA does not 55 represent the true SSA value. We have changed it accordingly. 'Although our retrieved SSA seems closer to AERONET retrieved SSA than that provided in OMAERUV, one should keep in mind that there is only one record for this event, the meteorological conditions, combustion phases and even the aerosol compositions may change during the 3-hour time difference. (line 186-189)

60

Line 245. Although the same wavelength pair (354, 388) is used in the OMAERUV UVAI definition, the way the observed radiances are calculated have changed [Torres et al., 2018]. The new definition handles the effect of clouds using of Mie Theory (instead of the Lambertian approximation), and reduces significantly the across-track angular dependence. It also accounts for wavelength dependence of surface reflectance as well as sun glint over

- 65 the oceans. For cloud-free conditions, the Mie-UVAI is slightly larger than LER-UVAI (~0.3) for nadir-viewing conditions. The original definition (consistent with that in TROPOMI) is still reported in the OMAERUV as a parameter labeled 'RESIDUE'. Thus, the original definition should be used for algorithm training purposes. This is important because the difference between the LER and Mie definitions increases with viewing sensor solar zenith angle.
- 70 Thank you for your correction. We have re-collected the OMAERUV (as we did not extract the residue from the original files at the first place) and re-collocated it to the AERONET data set using the same criteria.

We have added explanations in the manuscript. 'Note that the UVAI used here is the 'residue' field in the original OMAERUV product, where the simulated radiance  $(l_{\lambda}^{Ray} in Eq.(1))$  is calculated by a simple Lambertian 75 approximation that is consistent with TROPOMI UVAI (Torres et al., 2018).' (line 272-274)

The following heatmap gives a quick view of how the UVAI and residue correlates with other parameters. The correlation between the residue and AOD, AAOD and ALH is relatively higher compared with other parameters, but not significantly high (at or less than the level of 0.5) compared with UVAI. As a result, we use a stricter **80** criterion that only keeping OMAERIUN pixels with cloud fraction smaller than 0.1. This enhance the correlation

80 criterion that only keeping OMAERUV pixels with cloud fraction smaller than 0.1. This enhance the correlation between UVAI and AOD, AAOD and ALH (r = 0.66, 0.66 and 0.4, respectively). There is total 5679 samples in the training data set. We have the following changes in the manuscript:

'OMI pixels with  $\theta_0$  larger than 70° or cloud fraction larger than 0.1 are excluded.' (line 260-261) 85

'In total 5679 samples are obtained.' (line 266)

'The empirical ratio between a training and testing data set is 70% versus 30%, thus there are 3975 samples in the training data set and 1704 samples in the testing data set,' (line 289-290)

But be noted that in the case applications the input TROPOMI data is still using CF < 0.3 to ensure there is collocated pixels with AERONET sites. In the heatmap, the residue seems has a higher dependence on the solar zenith angle and the surface albedo, too. But we may only focus on the aerosol related features. We have the following changes in the manuscript:

'The UVAI also has a dependence on  $\theta_0$ , but in this study we only focus on the aerosol related features.' (line 281-282)

100 'Note that the data includes pixels with CF larger than 0.1 in order to ensure there is satellite measurements collocated with the AERONET sites (though CF is no larger than 0.3).' (line 354-356)

'The input features are selected by the *Spearman's rank* correlation coefficients and a priori knowledge on relationship between UVAI and only aerosol related features.' (line 439-440)

105

90

95

'Other non-aerosol features affecting UVAI should be also taken into consideration.' (line 445-446)

| (  | Deleted:                                                                                      |
|----|-----------------------------------------------------------------------------------------------|
| -( | Deleted: 1,                                                                                   |
| Ì  | Deleted: or UVAI354.588 smaller than 0.8, or pixels with extreme high ALH but low UVAI |
| (  | Deleted: 4003                                                                                 |

**Deleted:** The training data set containing the original OMAERUV ALH and the adjusted ALH are referred to as the original and adjusted training data set, respectively. We fit the SVR for AAOD prediction to both training data sets in order to investigate the importance of a reliable ALH input.

- 120 The new training data set leads to small changes in the SVR model hyper-parameter C, which is 0.08 (used to be 0.11). The accuracy of the updated SVR is slightly lower than that of the previous one (with UVAI in the training data set). The updated SVR tends to predict higher SSA than the previous one, which is as expected, because the Mie-UVAI is overall 0.3 larger than the residue. The higher UVAI (if the ALH and AOD keep the same), the lower predicted SSA as we described in the sensitivity study in Section 3.5. From the case applications, 5 out of 9
- 125 of the predicted SSA records is within +/- 0.03 difference compared with the AERONET records (used to be 66%), and not all predictions are within +/- 0.05 difference (88%).

We also have updated the Figure.6-15 and figures in Appendix B, and results in Table 3 and 4 accordingly.

Line 246. In the description of the OMAERUV record, it should be stated that AOD retrievals have been
 validated using the multi-year AERONET aerosol record [Ahn et al., 2014], and, similarly, the SSA parameter
 have been evaluated with AERONET Almucantar retrievals [Jethva, et al., 2014].

We have added them accordingly. 'Its AOD was validated by the multi-year AERONET record (Ahn et al., 2014) and its SSA was evaluated by AERONET Almucantar retrievals (Jethva et al., 2014).' (line 252-254)

- Line 426 'Acceptable' is a subjective interpretation. Replace 'acceptable (+/- 0.02)' with '+/- 0.02' 135 We have changed it accordingly. 'The accuracy of SVR-predicted SSA is  $\pm 0.02$ , with higher tendency to
- overestimate the SSA for relatively absorbing cases.' (line 441-442)

**Cited References**

- https://doi.org/10.5194/amt-11-2701-2018
   Torres, O., Bhartia, P. K., Jethva, H., and Ahn, C.: Impact of the ozone monitoring instrument row anomaly on the long-term record of aerosol products, Atmos. Meas. Tech., 11, 2701-2715, 2018.
  - Jethva, H., O. Torres, and C. Ahn (2014), Global assessment of OMI aerosol single-scattering albedo using ground-based AERONET inversion, J. Geophys. Res. Atmos., 119, doi:10.1002/2014JD021672.

10.1002/2013JD020188Ahn, C., O. Torres, and H. Jethva (2014), Assessment of OMI near-UV aerosol optical depth over land, J. Geophys. Res. Atmos., 119, 2457–2473, doi:.

|  <li>Jyunting Sun1,2, Pepijn Vecfkind1,2, Swadhin Nanda1,2, Peter van Velhoven3, Pieternel Levelt1,2</li> <li>1Department of Satellite Observations, Royal Netherlands Meteorological Institute, De Bilt, 3731 GA, the Netherlands</li> <li>1Department of Geoscience and Remote Sensing (GRS), Civil Engineering and Geosciences, Delft Diversity of Technology, Ddft, 2620 CD, the Netherlands</li> <li>1Department of Weather & Climate Models, Royal Netherlands Meteorological Institute, De Bilt, 3731 GA, the Netherlands</li> <li>1Department of Weather & Climate Models, Royal Netherlands Meteorological Institute, De Bilt, 3731 GA, the Netherlands</li> <li>1Correspondence to: Jyunting Sun (ijyunting sum@knmi.nl)</li> <li>Abstract. The purpose of this study is to demonstrate the role of aerosol layer height (ALH) in quantifying the single scattering albedo (SSA) from ultraviolet satellite observations for biomass burning aerosols. In the first experiment, we retrieve SSA tore maintring the UVAI difference between observed ones and that simulations, a significant gap method to retrieve SSA (0.25) is found between radiative transfer simulations with spectral flat aerosols and strong spectru dependence may cause severe misinterpretations of aerosol absorption neetral data demonstrate the postorial eagend measurements significant gap to extore regression (SVR). This empirical method is free from the uncertainties due to the imperfection of a SSA for some extorem situations. The apatian contrast in SSA retrievely basching cases and fails to prediate SSA and AERONET are generally within ±0.05, with over half of samples are within ±0.03. The results are encouraging, though at the current phase, the model tends to oversetimate the SSA for relatively absorbing cases and fails to prediated Fourit. Dis , Not Retal, ferent absorb in gaves of anotal absorption in TWAI records.</li> <li>1 Introduction</li> <li>2 The concept of the near-ultraviolet (near-UV) absorbing aerosol al</li>                                                                        |   | from ultraviolet satellite observations                                                                                                                             | Formatted: Font: (Default) Times New Roman                                |
|-------------------------------------------------------------------------------------------------------------------------------------------------------------------------------------------------------------------------------------------------------------------------------------------------------------------------------------------------------------------------------------------------------------------------------------------------------------------------------------------------------------------------------------------------------------------------------------------------------------------------------------------------------------------------------------------------------------------------------------------------------------------------------------------------------------------------------------------------------------------------------------------------------------------------------------------------------------------------------------------------------------------------------------------------------------------------------------------------------------------------------------------------------------------------------------------------------------------------------------------------------------------------------------------------------------------------------------------------------------------------------------------------------------------------------------------------------------------------------------------------------------------------------------------------------------------------------------------------------------------------------------------------------------------------------------------------------------------------------------------------------------------------------------------------------------------------------------------------------------------------------------------------------------------------------------------------------------------------------------------------------------------------------------------------------------------------------------------------------------------------------------------------------------------------------------------------------------------------------------------------------------------------------------------------|---|---------------------------------------------------------------------------------------------------------------------------------------------------------------------|---------------------------------------------------------------------------|
|  <li>Department of Satellite Observations, Royal Netherlands Meteorological Institute, Dc Bilt, 3731 GA, the Netherlands</li> <li>Department of Conseince and Remote Seming (GRS), CVII Engineering and Geosciences, Defit University of Technology, Defit, 2628 CD, the Netherlands</li> <li>Department of Weather & Climate Models, Royal Netherlands Meteorological Institute, Dc Bilt, 3731 GA, the Netherlands</li> <li>Correspondence to: Flyunting Sun (ijyunting, sungikami.al)</li> <li>Abstract. The purpose of this study is to demonstrate the role of acrosol layer height (ALH) in quantifying the single scattering albedo (SSA) from ultraviolet staellite observations for biomass burning acrosols. In the first experiment, we retrieve SSA by minimizing the UVAI difference between observed ones and that simulated by a radiative transfer model. With the recently released ALH product 15-5P TROPOMI constraining forward simulations, as significant gap</li> <li>in the retrieved SSA (0.25) is found between radiative transfer simulations as a significant gap</li> <li>priori assumptions on aerosol absorption. In the second part of this paper, we propose an alternative method to retrieve SSA based on long-term record of collocated satellite and ground-based measurements psing the support vector ergression (SVR). This empirical method is free from the uncertainties due to the imperfection of a priori assumptions on aerosol absorption. In the second part of this paper, we propose an alternative matcher into externe situations. The spatial contrast in SSA for relatively absorbing acrosol absorption free WVR. Filter 2000 SR and AERONET are generally within ±0.05, with over half of samples are within ±0.03. The results are encouraging, though at the current phase, the model tuncts to externe situations. The spatial contrast in SSA for model with the ator SVR by severtime situations. The spatial contrast in SSA for model with the orone provectif d he name-ultraviolet (near-UV) absorbing acrosol absorption fre</li>                                                                                                                                                                                |   | Jiyunting Sun 1,2 , Pepijn Veefkind 1,2 , Swadhin Nanda 1,2 , Peter van Velthoven 3 , Pieternel Levelt 1,2   |                                                                           |
|  <li>1 Department of Geoscience and Remote Seming (GRS), Civil Engineering and Geosciences.
Deff University of Technology, Deff, 2628 CD, the Netherlands</li> <li>1 Department of Weather & Climate Models, Royal Netherlands Meteorological Institute, De Bilt, 3731 GA, the Netherlands</li> <li>1 Department of Weather & Climate Models, Royal Netherlands Meteorological Institute, De Bilt, 3731 GA, the Netherlands</li> <li>1 Correspondence to: Jiyunting Sun (jiyunting, sun@kmi.nl)</li> <li>Abstract. The purpose of this study is to demonstrate the role of aerosol layer height (ALH) in quantifying the single scattering albedo (SSA) from ultraviolet satellite observations for biomass burning aerosols. In the first experiment, we retrieve SSA by minimizing the UVAI difference between observed ones and that simulated by a radiative transfer model, With the recently released ALH product of S-SP TROPOMI constraining forward simulations, a significant gap in the retrieved SSA (0.25) is found between radiative transfer isimulations with spectral dependence may cause severe misintepretations of aerosol absorption. In the second part of this paper, we propose an alternative method to retrieve SSA based on long-term record of collocated statellite and ground-based measurements ging the support vector regression (SVR). This empirical method is free from the uncertainties due to the imperfection of a forto solution. The spatial contrast in SSA for relatively absorbing earos and absorption so aerosol absorption and exolutions is significantly.</li> <li>1 bigher than that of SVR, and the latter better agrees with SSA for multify a clonade at encouse. Moreover, the high resolution TROPOMI UVAI and collocated ALH product swill guide us to more reliable training data set and more powerful algorithms in quantify aerosol absorption from UVAI records.</li> <li>1 Introduction</li> <li>1 The concert of the Nimbus 7/Total Ozone Mapping Spectrometers (TOMS). It detects elevated UV-absorbing ae</li>                                                                                                             | 0 | 1 , Department of Satellite Observations, Royal Netherlands Meteorological Institute, De Bilt, 3731 GA, the Netherlands                                  | Formatted: Font: (Default) Times New Roman                                |
|  <li>Delft University of Technology, Delft, 2628 CD, the Netherlands</li> <li>1 Department of Wenther & Climate Models, Royal Netherlands Meteorological Institute, De Bilt, 3731 GA, the Netherlands</li> <li>2 Correspondence to: Jiyuuting Sun (jiyuuting.sun@kmi.il)</li> <li>Abstract. The purpose of this study is to demonstrate the role of aerosol layer height (ALH) in quantifying the single scattering albedo (SSA) from ultraviolet satellite observations for biomass burning aerosols. In the first experiment, we retrieve SSA by minimizing the UVAI difference between observed ones and that simulated by a radiative transfer model, With the recently released ALH product of 5-5P TROPOMI constraining forward simulations, as gainficant gap in the retrieved SSA (0.25) is found between radiative transfer simulations with spectral flat aerosols and strong spectral dependent aerosols, implying that inappropriate assumptions on aerosol absorption. In the second part of this paper, we propose an alternative method to retrieve SSA based on long-term record of collocated stallitie and ground-based measurements jusing the support vector regression (SVR). This empirical method is free from the uncertainties due to the imperfection of a 5P transformed in recent years. For all cases, the difference between SVR-retrieved SSA and ALRONET are generally within ±0.05, with over half of samples are within ±0.05. The results are encouraging, though at the current phase, the model tends to overestimate the SSA for relatively absorbing cases and fails to predict SSA for some extreme situations. The spatial contrast in SSA retrieved by radiative transfer simulations is significantly higher than that of SVR, and the latter better agrees with SSA for melERA-2 reanalysis. In the future, more sophisticated feature selection procedure and kernel functions should be taken into consideration to improve the SVR model accuracy. Marcover, the high resolution TROPOMI UVAI and collocated ALH products will guide us to more reliable training data</li>                                                                                                                                   |   | 2 Department of Geoscience and Remote Sensing (GRS), Civil Engineering and Geosciences,                                                                  | Formatted: Font: (Default) Times New Roman                                |
|  <li>1 Department of Weather & Climate Models, Royal Netherlands Meteorological Institute, De Bilt, 3731 GA, the Netherlands</li> <li>Correspondence to: Jyunting Sun (ijyunting sun@knni.nl)</li> <li>Abstract. The purpose of this study is to demonstrate the role of aerosol layer height (ALH) in quantifying the single scattering albedo (SSA) from ultraviolet satellite observations for biomass burning aerosols. In the first experiment, we retrieved SSA (0.25) is found between role of x-5P TROPOMI constraining forward simulations, a significant gap</li> <li>in the retrieved SSA (0.25) is found between radiative transfer simulations with spectral flat aerosols and strong spectral dependent aerosols, implying that inappropriate assumptions on aerosol absorption spectral dependence may cause severe misinterpretations of aerosol absorption. In the second part of this paper, we propose an alternative method to retrieve SSA based on long-term record of collocated stellite and ground-based measurements gaing the support vector regression (SVR). This empirical method is free from the uncertainties due to the imperfection of a priori assumptions on aerosol absorption. In the second part of this paper, we propose an alternative method to retrieve SSA based on long-term record of collocated stellite and ground-based measurements gaing the support vector regression (SVR). This empirical method is free from the uncertainties due to the imperfection of a priori assumptions on aerosol absorption. The results are encouraging, though at the current base, the model tends to overestimate the SSA for relatively absorbing cases and fails to prefict SSA for some extreme situations. The spatial contrast in SSA retrieved by radiative transfer simulations is significantly high at not of NCR, and the latter better agrees with SSA for MERA-2 renalysis. In the future, more sophisticated faiture selection procedure and kerned 
[revised manuscript text omitted]

Utitie: tota of in a Rayleigh atmosphere (Herman et al., 1997):
Utitie: tota of in a given wavelength pair ( $\lambda$ and $\lambda_o$ )
(1)                                                                                                                                                                                                                     |   | spectral dependent aerosols, implying that inappropriate assumptions on aerosol absorption spectral dependence may                                                  |                                                                           |
| method to retrieve SSA based on long-term record of collocated satellite and ground-based measurements using the
support vector regression (SVR). This empirical method is free from the uncertainties due to the imperfection of a
priori assumptions on aerosol micro-physics as that in the first experiment. We present the potential capabilities of the
SVR by several fire events happened in recent years. For all cases, the difference between SVR-retrieved SSA and
AERONET are generally within ±0.05, with over half of samples are within ±0.03. The results are encouraging,
though at the current phase, the model tends to overestimate the SSA for relatively absorbing cases and fails to predict
SSA for some extreme situations. The spatial contrast in SSA retrieved by radiative transfer simulations is significantly
higher than that of SVR, and the latter better agrees with SSA from MERRA-2 reanalysis. In the future, more
sophisticated feature selection procedure and kernel functions should be taken into consideration to improve the SVR
model accuracy. Moreover, the high resolution TROPOMI UVAI and collocated ALH products will guide us to more
reliable training data set and more powerful algorithms in quantify aerosol absorption from UVAI records.
Introduction
The concept of the near-ultraviolet (near-UV) absorbing aerosol index (UVAI) initially came along with the ozone
product of the Nimbus 7/Total Ozone Mapping Spectrometers (TOMS). It detects elevated UV-absorbing aerosol
layers by measuring the spectral contrast difference between a satellite observed radiance in a real atmosphere and a
model simulated one in a Rayleigh atmosphere (Herman et al., 1997):                                                                                                                                                                                                                                                                                                                                                                                                                                                                                                          |   | cause severe misinterpretations of aerosol absorption. In the second part of this paper, we propose an alternative                                                  |                                                                           |
| support vector regression (SVR). This empirical method is free from the uncertainties due to the imperfection of a priori assumptions on aerosol micro-physics as that in the first experiment. We present the potential capabilities of the SVR by several fire events happened in recent years. For all cases, the difference between SVR-retrieved SSA and AERONET are generally within $\pm 0.05$ , with over half of samples are within $\pm 0.03$ . The results are encouraging, though at the current phase, the model tends to overestimate the SSA for relatively absorbing cases and fails to predict SSA for some extreme situations. The spatial contrast in SSA retrieved by radiative transfer simulations is significantly higher than that of SVR, and the latter better agrees with SSA from MERRA-2 reanalysis. In the future, more sophisticated feature selection procedure and kernel functions should be taken into consideration to improve the SVR model accuracy. Moreover, the high resolution TROPOMI UVAI and collocated ALH products will guide us to more reliable training data set and more powerful algorithms in quantify aerosol absorption from UVAI records.  I Introduction The concept of the near-ultraviolet (near-UV) absorbing aerosol index (UVAI) initially came along with the ozone product of the Nimbus 7/Total Ozone Mapping Spectrometers (TOMS). It detects elevated UV-absorbing aerosol layers by measuring the spectral contrast difference between a satellite observed radiance in a real atmosphere and a model simulated one in a Rayleigh atmosphere [Herman et al., 1997): The conc to $\mu h_{\mu}^{abs}$ , $\mu = \mu h_{\mu}^{abs}$ , $\mu = \frac{h_{\mu}^{abs}}{abs}$ , $\mu = \frac{h_{\mu}^{abs}}{$ |   | method to retrieve SSA based on long-term record of collocated satellite and ground-based measurements using the                                                    | Deleted: and ALH                                                          |
| priori assumptions on aerosol micro-physics as that in the first experiment. We present the potential capabilities of the SVR by several fire events happened in recent years. For all cases, the difference between SVR-retrieved SSA and AERONET are generally within $\pm 0.05$ , with over half of samples are within $\pm 0.03$ . The results are encouraging, though at the current phase, the model tends to overestimate the SSA for relatively absorbing cases and fails to predict SSA for some extreme situations. The spatial contrast in SSA retrieved by radiative transfer simulations is significantly higher than that of SVR, and the latter better agrees with SSA from MERRA-2 reanalysis. In the future, more sophisticated feature selection procedure and kernel functions should be taken into consideration to improve the SVR model accuracy. Moreover, the high resolution TROPOMI UVAI and collocated ALH products will guide us to more reliable training data set and more powerful algorithms in quantify aerosol absorption from UVAI records.  Interduction The concept of the near-ultraviolet (near-UV) absorbing aerosol index (UVAI) initially came along with the ozone product of the Nimbus 7/Total Ozone Mapping Spectrometers (TOMS). It detects elevated UV-absorbing aerosol layers by measuring the spectral contrast difference between a satellite observed radiance in a real atmosphere and a model simulated one in a Rayleigh atmosphere (Herman et al., 1997):  With the tot of the magnet of t                                                                                                                                                             |   | support vector regression (SVR). This empirical method is free from the uncertainties due to the imperfection of a                                                  |                                                                           |
| SVR by several fire events happened in recent years. For all cases, the difference between SVR-retrieved SSA and
AERONET are generally within $\pm 0.05$ , with over half of samples are within $\pm 0.03$ . The results are encouraging,
though at the current phase, the model tends to overestimate the SSA for relatively absorbing cases and fails to predict
SSA for some extreme situations. The spatial contrast in SSA retrieved by radiative transfer simulations is significantly
higher than that of SVR, and the latter better agrees with SSA from MERRA-2 reanalysis. In the future, more
sophisticated feature selection procedure and kernel functions should be taken into consideration to improve the SVR
model accuracy. Moreover, the high resolution TROPOMI UVAI and collocated ALH products will guide us to more
reliable training data set and more powerful algorithms in quantify aerosol absorption from UVAI records.
1 Introduction
The concept of the near-ultraviolet (near-UV) absorbing aerosol index (UVAI) initially came along with the ozone
product of the Nimbus 7/Total Ozone Mapping Spectrometers (TOMS). It detects elevated UV-absorbing aerosol
layers by measuring the spectral contrast difference between a satellite observed radiance in a real atmosphere and a
model simulated one in a Rayleigh atmosphere (Herman et al., 1997):
With $t = top (t_1, t_2)^{abs}$ , $t = top (t_2)^{abs}$                                                                                                 |   | priori assumptions on aerosol micro-physics as that in the first experiment. We present the potential capabilities of the                                           |                                                                           |
| AERONET are generally within ±0.05, with over half of samples are within ±0.03. The results are encouraging,         though at the current phase, the model tends to overestimate the SSA for relatively absorbing cases and fails to predict         SSA for some extreme situations. The spatial contrast in SSA retrieved by radiative transfer simulations is significantly         higher than that of SVR, and the latter better agrees with SSA from MERRA-2 reanalysis. In the future, more         sophisticated feature selection procedure and kernel functions should be taken into consideration to improve the SVR         model accuracy. Moreover, the high resolution TROPOMI UVAI and collocated ALH products will guide us to more         reliable training data set and more powerful algorithms in quantify aerosol absorption from UVAI records.         1       Introduction         Free concept of the near-ultraviolet (near-UV) absorbing aerosol index (UVAI) initially came along with the ozone         product of the Nimbus 7/Total Ozone Mapping Spectrometers (TOMS). It detects elevated UV-absorbing aerosol         layers by measuring the spectral contrast difference between a satellite observed radiance in a real atmosphere and a         model simulated one in a Rayleigh atmosphere (Herman et al., 1997):         Uttrate       top (the spectral contrast difference (the spectr                                                                                                                                                                                                                                                                                                                                                                                                                                                                                                                                           |   | SVR by several fire events happened in recent years. For all cases, the difference between SVR-retrieved SSA and                                                    |                                                                           |
|  <li>though at the current phase, the model tends to overestimate the SSA for relatively absorbing cases and fails to predict SSA for some extreme situations. The spatial contrast in SSA retrieved by radiative transfer simulations is significantly higher than that of SVR, and the latter better agrees with SSA from MERRA-2 reanalysis. In the future, more sophisticated feature selection procedure and kernel functions should be taken into consideration to improve the SVR model accuracy. Moreover, the high resolution TROPOMI UVAI and collocated ALH products will guide us to more reliable training data set and more powerful algorithms in quantify aerosol absorption from UVAI records.</li> <li>1 Introduction</li> <li>The concept of the near-ultraviolet (near-UV) absorbing aerosol index (UVAI) initially came along with the ozone product of the Nimbus 7/Total Ozone Mapping Spectrometers (TOMS). It detects elevated UV-absorbing aerosol layers by measuring the spectral contrast difference between a satellite observed radiance in a real atmosphere and a model simulated one in a Rayleigh atmosphere (Herman et al., 1997):</li> <li>Witting to a time with abs at an with Ray.</li>                                                                                                                                                                                                                                                                                                                                                                                                                                                                                                                                                                                                                                                                                                                                                                                                                                                                                                                                                                                                                           |   | AERONET are generally within $\pm 0.05$ , with over half of samples are within $\pm 0.03$ . The results are encouraging,                                            |                                                                           |
| SSA for some extreme situations. The spatial contrast in SSA retrieved by radiative transfer simulations is significantly
higher than that of SVR, and the latter better agrees with SSA from MERRA-2 reanalysis. In the future, more
sophisticated feature selection procedure and kernel functions should be taken into consideration to improve the SVR
model accuracy. Moreover, the high resolution TROPOMI UVAI and collocated ALH products will guide us to more
reliable training data set and more powerful algorithms in quantify aerosol absorption from UVAI records.
1 Introduction
The concept of the near-ultraviolet (near-UV) absorbing aerosol index (UVAI) initially came along with the ozone
product of the Nimbus 7/Total Ozone Mapping Spectrometers (TOMS). It detects elevated UV-absorbing aerosol
layers by measuring the spectral contrast difference between a satellite observed radiance in a real atmosphere and a
model simulated one in a Rayleigh atmosphere (Herman et al., 1997):
UNITY = tab $(h_{in} e^{h_{in}})$ (1)
Field Code Changed                                                                                                                                                                                                                                                                                                                                                                                                                                                                                                                                                                                                                                                                                                                                                                                                                                                                                                                                                                                                                                                                                                                                                                         |   | though at the current phase, the model tends to overestimate the SSA for relatively absorbing cases and fails to predict                                            |                                                                           |
|  <li>higher than that of SVR, and the latter better agrees with SSA from MERRA-2 reanalysis. In the future, more sophisticated feature selection procedure and kernel functions should be taken into consideration to improve the SVR model accuracy. Moreover, the high resolution TROPOMI UVAI and collocated ALH products will guide us to more reliable training data set and more powerful algorithms in quantify aerosol absorption from UVAI records.</li> <li>I Introduction</li> <li>Formatted: Heading 2, Line spacing: 1.5 lines</li> <li>Deleted: 1 Introduction1</li> <li>Formatted: Formatted: Formatted</li>                                                                                                                                                               |   | SSA for some extreme situations. The spatial contrast in SSA retrieved by radiative transfer simulations is significantly                                           |                                                                           |
| sophisticated feature selection procedure and kernel functions should be taken into consideration to improve the SVR model accuracy. Moreover, the high resolution TROPOMI UVAI and collocated ALH products will guide us to more reliable training data set and more powerful algorithms in quantify aerosol absorption from UVAI records.  I Introduction The concept of the near-ultraviolet (near-UV) absorbing aerosol index (UVAI) initially came along with the ozone product of the Nimbus 7/Total Ozone Mapping Spectrometers (TOMS). It detects elevated UV-absorbing aerosol layers by measuring the spectral contrast difference between a satellite observed radiance in a real atmosphere and a model simulated one in a Rayleigh atmosphere (Herman et al., 1997): UNIT of the output of the code Changed  Field Code Changed  Field Code Changed                                                                                                                                                                                                                                                                                                                                                                                                                                                                                                                                                                                                                                                                                                                                                                                                                                                                                                                                                                                                                                                                                                                                                                                                                                                                                                                                                                                                                                |   | higher than that of SVR, and the latter better agrees with SSA from MERRA-2 reanalysis. In the future, more                                                         |                                                                           |
|  <li>model accuracy. Moreover, the high resolution TROPOMI UVAI and collocated ALH products will guide us to more reliable training data set and more powerful algorithms in quantify aerosol absorption from UVAI records.</li> <li>1 Introduction</li> <li>Formatted: Heading 2, Line spacing: 1.5 lines</li> <li>The concept of the near-ultraviolet (near-UV) absorbing aerosol index (UVAI) initially came along with the ozone product of the Nimbus 7/Total Ozone Mapping Spectrometers (TOMS). It detects elevated UV-absorbing aerosol layers by measuring the spectral contrast difference between a satellite observed radiance in a real atmosphere and a model simulated one in a Rayleigh atmosphere (Herman et al., 1997):</li> <li>With a tot (l = dl), abs = l = dl), Ray.</li>                                                                                                                                                                                                                                                                                                                                                                                                                                                                                                                                                                                                                                                                                                                                                                                                                                                                                                                                                                                                                                                                                                                                                                                                                                                                                                                                                                                                                                                          |   | sophisticated feature selection procedure and kernel functions should be taken into consideration to improve the SVR                                                |                                                                           |
| reliable training data set and more powerful algorithms in quantify aerosol absorption from UVAI records. 1 Introduction Formatted: Heading 2, Line spacing: 1.5 lines         The concept of the near-ultraviolet (near-UV) absorbing aerosol index (UVAI) initially came along with the ozone product of the Nimbus 7/Total Ozone Mapping Spectrometers (TOMS). It detects elevated UV-absorbing aerosol layers by measuring the spectral contrast difference between a satellite observed radiance in a real atmosphere and a model simulated one in a Rayleigh atmosphere (Herman et al., 1997):       Deleted:       Deleted:       Deleted:         Uttate       tato (la - (la), abs )       (la), Bag )       (1)       Field Code Changed                                                                                                                                                                                                                                                                                                                                                                                                                                                                                                                                                                                                                                                                                                                                                                                                                                                                                                                                                                                                                                                                                                                                                                                                                                                                                                                                                                                                                                                                                                                 |   | model accuracy. Moreover, the high resolution TROPOMI UVAI and collocated ALH products will guide us to more                                                        |                                                                           |
| 1 Introduction       Formatted: Heading 2, Line spacing: 1.5 lines         The concept of the near-ultraviolet (near-UV) absorbing aerosol index (UVAI) initially came along with the ozone product of the Nimbus 7/Total Ozone Mapping Spectrometers (TOMS). It detects elevated UV-absorbing aerosol layers by measuring the spectral contrast difference between a satellite observed radiance in a real atmosphere and a model simulated one in a Rayleigh atmosphere (Herman et al., 1997):       Deleted: Introduction*         With a conc (I = - (I), S obs = I = - (I), S Ray )       (1)                                                                                                                                                                                                                                                                                                                                                                                                                                                                                                                                                                                                                                                                                                                                                                                                                                                                                                                                                                                                                                                                                                                                                                                                                                                                                                                                                                                                                                                                                                                                                                                                                                                                        |   | reliable training data set and more powerful algorithms in quantify aerosol absorption from UVAI records.                                                           |                                                                           |
| The concept of the near-ultraviolet (near-UV) absorbing aerosol index (UVAI) initially came along with the ozone
product of the Nimbus 7/Total Ozone Mapping Spectrometers (TOMS). It detects elevated UV-absorbing aerosol
layers by measuring the spectral contrast difference between a satellite observed radiance in a real atmosphere and a
model simulated one in a Rayleigh atmosphere (Herman et al., 1997):
UVAL $= 400 \text{ (} h e^{-f_{1}} e^{\frac{h}{2}} e^{\frac{h}{2}} e^{-f_{1}} e^{\frac{h}{2}} e^{\frac{h}{2}} e^{-f_{1}} e^{\frac{h}{2}} e^{\frac{h}{2}} e^{-f_{1}} e^{\frac{h}{2}} e^{\frac{h}{2}} e^{-f_{1}} e^{\frac{h}{2}} e^{-f_{1}} $                                                                                                    |   | 1 Introduction                                                                                                                                               | Formatted: Heading 2, Line spacing: 1.5 lines                             |
| product of the Nimbus 7/Total Ozone Mapping Spectrometers (TOMS). It detects elevated UV-absorbing aerosol
layers by measuring the spectral contrast difference between a satellite observed radiance in a real atmosphere and a
model simulated one in a Rayleigh atmosphere (Herman et al., 1997):
$\frac{d_{10}}{d_{10}} \frac{d_{10}}{d_{10}} \frac{d_{10}}{d_{10}} \frac{d_{10}}{d_{10}} \frac{Ray}{d_{10}}$ (1) Formatted: Font: 10.5 pt, Not Bold, Font color: Auto, (Chinese (China), (Other) English (US)
Field Code Changed                                                                                                                                                                                                                                                                                                                                                                                                                                                                                                                                                                                                                                                                                                                                                                                                                                                                                                                                                                                                                                                                                                                                                                                                                                                                                                                                                                                                                                                                                                                                                                                                                                                             |   | The concept of the near-ultraviolet (near-UV) absorbing aerosol index (UVAI) initially came along with the ozone                                                    | Deleted: 1 Introduction                                                   |
| layers by measuring the spectral contrast difference between a satellite observed radiance in a real atmosphere and a
model simulated one in a Rayleigh atmosphere (Herman et al., 1997):
$\frac{1}{1}$ (1) (Chinese (China), (Other) English (US) (Deleted: Deleted: Deleted: Deleted: (1) (1) (1) (1) (1) (1) (1) (1) (1) (1)                                                                                                                                                                                                                                                                                                                                                                                                                                                                                                                                                                                                                                                                                                                                                                                                                                                                                                                                                                                                                                                                                                                                                                                                                                                                                                                                                                                                                                                                                                                                                                                                                                                                                                                                                                                                                                                                                                                                                           |   | product of the Nimbus 7/Total Ozone Mapping Spectrometers (TOMS). It detects elevated UV-absorbing aerosol                                                          | Formatted: Font: 10.5 pt, Not Bold, Font color: Auto, (Asia               |
| model simulated one in a Rayleigh atmosphere (Herman et al., 1997):
$\frac{Deleted: \text{ for a given wavelength pair } (\lambda \text{ and } \lambda_0)}{Field Code Changed}$ (1)                                                                                                                                                                                                                                                                                                                                                                                                                                                                                                                                                                                                                                                                                                                                                                                                                                                                                                                                                                                                                                                                                                                                                                                                                                                                                                                                                                                                                                                                                                                                                                                                                                                                                                                                                                                                                                                                                                                                                                                                                                                                                                          |   | layers by measuring the spectral contrast difference, between a satellite observed radiance in a real atmosphere and a                                              | Chinese (China), (Other) English (US)                                     |
| (1) Field Code Changed                                                                                                                                                                                                                                                                                                                                                                                                                                                                                                                                                                                                                                                                                                                                                                                                                                                                                                                                                                                                                                                                                                                                                                                                                                                                                                                                                                                                                                                                                                                                                                                                                                                                                                                                                                                                                                                                                                                                                                                                                                                                                                                                                                                                                                                                          |   | model simulated one in a Rayleigh atmosphere (Herman et al., 1997):                                                                                                 | Deleted: for a given wavelength pair ( $\lambda$ and $\lambda_0$ ) |
| $OVAI = -100 \left( log_{10} \left( \frac{\pi}{l_{\lambda 0}} \right) - log_{10} \left( \frac{\pi}{l_{\lambda 0}} \right) \right)$                                                                                                                                                                                                                                                                                                                                                                                                                                                                                                                                                                                                                                                                                                                                                                                                                                                                                                                                                                                                                                                                                                                                                                                                                                                                                                                                                                                                                                                                                                                                                                                                                                                                                                                                                                                                                                                                                                                                                                                                                                                                                                                                                              |   | $UVAI = -100 \left( log_{10} \left( \frac{I_{\lambda}}{I_{\lambda 0}} \right)^{obs} - log_{10} \left( \frac{I_{\lambda}}{I_{\lambda 0}} \right)^{Ray} \right) $ (1) | Field Code Changed                                                        |

[revised manuscript text omitted]

Field Code Changed Formatted: Font: 10.5 pt

**Field Code Changed**

Moved (insertion) [1]

**Deleted:** As the TROPOMI ALH product is not operationally available yet, we focus on the data of one of the largest wildfires that happened in southern California in 2017, i.e. the Thomas Fire incidents/incider ndex/1922 ). Ignited on 4 December 2017, the fire was expanded quickly northwest by the strong and persistent Santa Ana winds and was fully under control of 12 January 2018. The precise cause of the fire remains unknown, but a prolonged period of heat and absence of precipitation definitely contributed to this devastating fire (https://inciweb.nwcg.gov/incident/5670/ ). We selected one day (12 December 2017) for our case study. As shown in Fig.1, a brow smoke plume produced by the Thomas Fire was blown away from the continent and transported northwards. The major part of the plum was over the ocean with cloud free conditions, which is favorable for space-borne aerosol observations.

**Field Code Changed**

**Deleted:** The most straightforward approach to derive a relationship between UVAI and quantitative aerosol absorption properties like SSA is through forward radiative transfer simulations Lookup tables (LUTs) of simulated UVAI for various measuring geometries, acrosol properties, atmospheric and surface conditions are constructed by radiative transfer models (RTMs). Then SSA is derived by minimizing the difference between pre-calculated UVAI and satellite observed ones (Colarco et al., 2002; Hu et al., 2007; Jeong and Hsu, 2008; Sun et al., 2018). Hereafter, we refer to this method as the RTM-based retrieval.

- 250 optical depth (AAOD) using machine learning (ML) techniques. ML algorithms learn the underlying behavior of a system from a given training data set. They are particularly useful to address ill-defined inversion problems in the field of geosciences and remote sensing, where theoretical understanding is incomplete but there is a significant amount of observations (Lary et al., 2015). We employ ML techniques in order to avoid explicit assumptions on aerosol microphysics as made in the first experiment. By now, the ALH observations are not abundant and we will use the ALH
- accompanied in the AOD retrieval from the OMAERUV product in the training procedure (Torres et al., 2013). Nevertheless, the recent TROPOMI ALH and other future ALH products make such empirical methods of great potentials. Various ML algorithms have been developed to deal with classification or regression problems. In this paper we choose the support vector regression (SVR), a regression variant form of the support vector machines (SVM) (Drucker et al., 1997). Compared with other algorithms (e.g. the Artificial Neural Network), SVR is less sensitive to
- 260 training data size and can successfully work with limited quantity of data (Mountrakis et al., 2011; Shin et al., 2005). We will present the capability to retrieve SSA from UVAI of using this empirical method with multiple case studies. This paper is organized as follows: the first experiment is outlined in section 2, with description on setting radiative transfer simulations, and the analysis of the uncertainty trigger by the assumption on aerosol absorption spectral dependence; Section 3 starts with introduction of SVR, followed by training data set preparation, SVR model hyper-
- 265 parameter tuning, error analysis and case applications. Finally, the major conclusions and implications for future research are summarized in section 4.

**2** Experiment 1: SSA retrieval using radiative transfer simulations**

In this section, we present the first experiment that retrieves SSA by radiative transfer calculations as done in previous studies (Colarco et al., 2002; Hu et al., 2007; Jeong and Hsu, 2008; Sun et al., 2018). Forward radiative transfer

- 270 simulations are realized by the KNMI developed radiative transfer model DISAMAR (Determining Instrument Specifications and Analyzing Methods for Atmospheric Retrieval) (de Haan, 2011). Fig.1 illustrates the model inputs and the procedure. For each pixel, first, aerosol optical properties are computed by Mie theory for various pre-defined aerosol models. Then DISAMAR calculates UVAI using the corresponding satellite information: AOD, ALH, the solar zenith angle ( $\theta_0$ ), the viewing zenith angle ( $\theta_v$ ), the solar azimuth angle ( $\varphi_0$ ), the viewing azimuth angle ( $\varphi_v$ ), surface
- albedo (As) and surface pressure (P€) of the target pixel. The output of the forward simulations is a LUT of UVAI as a function of the input SSA (determined by the pre-defined aerosol models), which is fit by a second order polynomial function. Finally, by specifying the corresponding satellite observed UVAI, the SSA of the target pixel is estimated from the UVAI-SSA relationship. The retrieved SSA is reported at 500 nm in order to compare with the results of the SVR method. Section 2.1 will introduce the input parameters in radiative transfer simulations, followed by retrieval results in section 2.2.

**2.1 Radiative transfer simulation setup**

**2.1.1 Aerosol models**

The aerosol models used for the Mie calculations are a combination of the aerosol models in ESA Aerosol\_cci project (Holzer-Popp et al., 2013) and that in the OMAERUV algorithm (Torres et al., 2007; Torres et al., 2013). We assume a fine mode smoke aerosol type and further divide it into 7 subtypes as listed Table 1. We use the particle size distribution of the fine mode strongly absorbing aerosol of ESA Aerosol\_cci project. The geometric radius (ra) is 0.07

 $\mu m$  (effective radius  $r_{eff}$  of 0.14  $\mu m$ ) and the geometric standard deviation ( $\sigma_q$ ) is 1.7 (logarithm variance  $ln\sigma_q$  of

**Field Code Changed**

Apart from pre-assumed aerosol micro-physics, the aerosol loading and the aerosol vertical distribution are two key parameters in forward simulations of UVAI. The former is usually provided in terms of the aerosol optical depth (AOD). There are plentiful AOD products providing wide spatial-temporal coverage with various spectral choices. By contrast, only little information on the aerosol vertical distribution is available. The most well-known aerosol profile product is offered by the Cloud-Aerosol Lidar with Orthogonal Polarization (CALIOP), but the number of measurements is limited because of its narrow tracks (Winker et al., 2009). Passive sensors only measure columnar quantities but, in some cases, also provide the aerosol layer height (ALH), a compact form of aerosol profile information indicating where most aerosols are located. Chimot et al. (2017) present the feasibility of ALH retrieval using the oxygen (O2-O2) band at 447 nm of the Ozone Monitoring Instrument (OMI), but so far it has not been run operationally vet. ¶

So far it has not been run operationally yet. International (CHR), due so far it has not been run operationally yet. International (CHR), due so far it has been developed for the TROPOSpheric Monitoring Instrument (TROPOMI) on board the Copernicus Sentinel-5 Precursor (S-SP) (Sanders et al., 2015). TROPOMI was launched on 13 October 2017. The instrument is equipped with both the UV-visible (270–500 nm) and the near-infrared (NIR) (675–775 nm) channels (Veefkind et al., 2015), which makes it possible to interpret UVAI using corresponding ALH measurements. Furthermore, TROPOMI has a wide swath of 2600 km, providing daily global coverage with a high spatial resolution of 7×3.5 km2 in nadir.4 The purpose of this paper is to demonstrate the potential of the TROPOMI ALH product for quantifying aerosol absorption.

Moved up [1]: As the TROPOMI ALH product is not operationally available yet, we focus on the data of one of the largest wildfires that happened in southern California in 2017, i.e. the Thomas Fire

(http://www.fire.ca.gov/current\_incidents/incident/details/Index/1922 ). Ignited on 4 December 2017, the fire was expanded quickly northwest by the strong and persistent Stanta Ana winds and was fully under control on 12 January 2018. The precise cause of the fire remains unknown, but a prolonged period of heat and absence of precipitation definitely contributed to this devasating fire (https://inciweb.nwcg.gov/incident/5670/). We selected one day (12 December 2017) for our case study. As shown in Fig.1, a brown smoke plume produced by the Thomas Fire was blown away from the continent and transported northwards. The major part of the plume was over the ocean with cloud free conditions, which is favorable for space-borne aerosol observations.§

**Formatted: Default Paragraph Font, Font: 12 pt, English (UK) Formatted: Default Paragraph Font, Font: 12 pt, English (UK)**

Deleted: We conduct two experiments to investigate the potential of TROPOMI ALH for quantifying aerosol absorption of the smoke plume generated by the Thomas Fire. First, as concluded in our previous study (Sun et al., 2018), the absence of aerosol vertical distribution information and an improper spectral dependence of aerosol absorption in the near-UV region can be responsible for a large difference between estimated and measured SSA. Now with the TROPOMI ALH as a constraint, we are able to quantitatively determine the influences of assumed wavelength dependent aerosol absorption on retrieved SSA. Similar to our previous study (Sun et[1]

**Deleted: introduc**

**Deleted: y**

**Deleted:** the first part of section 3 expresses the setup of the RTMbased method and the first experiment, i.e. a sensitivity study of SSA examining 
[revised manuscript text omitted]